# Quo Vadis, Motion Generation? From Large Language Models to Large Motion Models

## Abstract

Inspired by recent success of LLMs, the field of human motion understanding has increasingly shifted towards the development of large motion models. Despite some progress, current works remain far from achieving truly generalist models, largely due to the lack of large-scale, high-quality motion data. To address this, we present MotionBase, the first million-level motion generation benchmark, offering 15 times the data volume of the previous largest dataset, and featuring multimodal data with hierarchically detailed text descriptions. By leveraging this vast dataset, our large motion model demonstrates strong performance across a broad range of motions, including unseen ones. Through systematic investigation, we underscore the importance of scaling both data and model size, with synthetic data and pseudo labels playing a crucial role in mitigating data acquisition costs. Moreover, our research reveals the limitations of existing evaluation metrics, particularly in handling out-of-domain text instructions — an issue that has long been overlooked. In addition, we introduce a 2D lookup-free approach for motion tokenization, which preserves motion information and expands codebook capacity, further enhancing the representative ability of large motion models. The release of MotionBase and the insights gained from this study are expected to pave the way for the development of more powerful and versatile motion generation models. Our code and database will be released at https://anonymous.4open.science/r/MotionBase.

## 1 Introduction

Motion generation is an emerging field with diverse applications in video games, filmmaking, and robotics animation. At the forefront of this area is text-to-motion generation (T2M) (Ahn et al., 2018; Ahuja & Morency, 2019), which plays a crucial role in translating natural language into human motions. State-of-the-art T2M models typically rely on a combination of the motion quantization methods (e.g., VQ (Van Den Oord et al., 2017)), along with a text encoder (e.g., CLIP (Radford et al., 2021)) and decoder (e.g., GPT-2 (Radford et al., 2019)) to generate motion sequences from detailed textual instructions. Despite the availability of a few high-quality datasets (Guo et al., 2022a; Lin et al., 2024) curated in recent years, their limited size restricts current methods to a narrow range of scenarios, creating performance bottlenecks when addressing diverse or unseen motions, as illustrated in Figure 1 (RIGHT).

The rapid advancement of large language models (LLMs) (Touvron et al., 2023a) in multimodal learning has been significantly bolstered by the availability of vast data resources (Zheng et al., 2024; Xu et al., 2024). In contrast, the volume of motion data remains considerably smaller than that of visual-text data, as illustrated in Figure 1 (LEFT). This disparity primarily arises from the high costs associated with motion data collection, which often requires specialized wearable devices and substantial human labor for annotation. Consequently, developing a state-of-the-art (SoTA) large motion model based on LLMs presents a significant challenge and remains an unresolved issue. While some recent efforts (Jiang et al., 2023) have explored this direction, the effectiveness of large motion models has yet to be fully demonstrated.

In this paper, we aim to address the question: "***Can a large motion model be a promising direction for motion generation?***" To tackle this, we have developed a systematic data collection scheme that led to the creation of MotionBase, the first large-scale dataset containing over one million motion

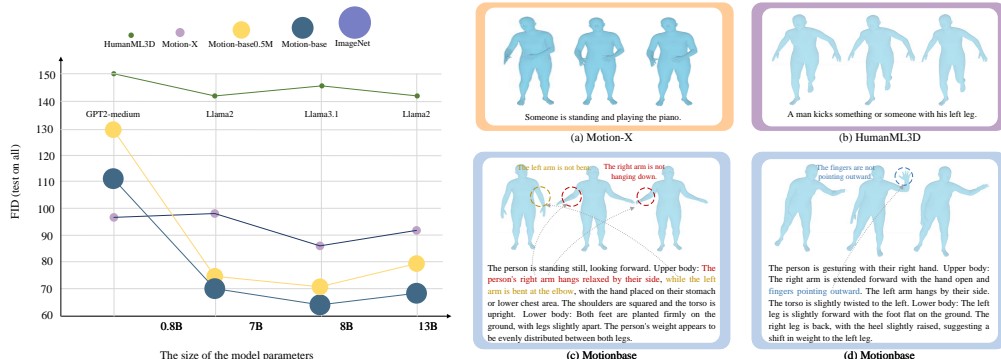

Figure 1: **LEFT**: Curves showing the effects of scaling up large motion models. MotionBase is the first large text-to-motion dataset comparable in scale to visual benchmarks like ImageNet. **RIGHT**: While existing models perform well on constrained datasets like Motion-X and HumanML3D, they struggle with out-of-domain concepts on MotionBase, exhibiting limited generalization.

sequences — 15 times larger than the previous largest dataset. This initiative provides a solid foundation for building robust, universally applicable large motion models and offers a comprehensive testbed for future research.

Building on the solid foundation of MotionBase, we can now conduct a comprehensive investigation into the effectiveness of large motion models. This research aims to firstly identify key factors driving their advancement and offer valuable insights for future model design, including: ❶ scaling both data and model size significantly reduces joint prediction errors on critical metrics while improving generalization to novel motions. ❷ Despite observable domain gaps, synthetic and static data, as well as pseudo motion labels are becoming increasingly essential and effective, especially given the high cost of acquiring ground truth motion data. ❸ Existing metrics show limitations when faced with out-of-domain text instructions. Notably, the widely used metric, FID, fails to accurately capture the alignment between ground truth and generated motions. Our findings highlight the need for a more robust and equitable evaluation framework that enhances open-set generalization.

In addition to these factors, we argue that large motion models are further constrained by inadequate motion representation. Most approaches rely on transforming motion into discrete tokens via vector quantization (VQ), which are then processed by autoregressive models to generate motion sequences. While these methods have produced impressive results, they suffer from two major drawbacks. ❶ **Information loss**: The current VQ process inevitably leads to the loss of critical information. Given a motion clip with $D$-dimensional features $\mathcal{M} = \{m_1, m_2, ..., m_T\}$, where $m_i \in \mathbb{R}^D$, VQ compresses it into a list of 1D embeddings of size $\lfloor T/\alpha \rfloor \times d$, where $\alpha$ is the temporal downsampling ratio and $d$ is the codebook dimension. Unlike images, which consist of uniform RGB pixel values, each motion state $m_i$ contains a set of distinct features (e.g., joint position, velocity, foot-ground contact). Using a single 1D embedding to represent such complex motion states is insufficient. This not only results in the loss of vital information but also limits the model's ability to flexibly generate motion at a part-level. ❷ **Limited Codebook Size:** Existing VQ are limited by a small codebook, meaning that all possible human motions must be selected from these limited options. Consequently, these 1D embeddings fail to capture the vast diversity of human motion.

To address this issue, we propose treating a motion clip as a 2D image with a single channel, represented as $\mathcal{M} \in R^{T \times D \times 1}$. By expanding the dimensionality of the motion clip from 1D to 2D, we enhance the encoder's capacity, improving its ability to represent complex motions while retaining more critical information after tokenization. Although increasing the size of the codebook is a straightforward way to enhance its expressiveness, this approach often leads to "codebook collapse," particularly when training samples are scarce. To mitigate this, we introduce a finite scalar quantizing method inspired by Mentzer et al. (2023), which enables learning a large motion vocabulary without requiring a lookup for corresponding tokens in the codebook for each entry. As a result, we expand the motion codebook by at least two orders of magnitude, boosting its representational capacity while maintaining efficiency.

We summarize our main contributions as follows. **(1) MotionBase**: We introduce MotionBase, the first large-scale motion generation benchmark containing over one million motions with detailed textual descriptions, significantly advancing the capability to effectively train motion generation models. **(2) Key Insights**: Our research identifies critical factors affecting the effectiveness of large motion models, emphasizing the importance of scaling both data and model size. Additionally, we uncover limitations in the current evaluation metrics, particularly when handling diverse and unseen motions. **(3) Novel Motion Quantization**: We propose a novel motion quantization approach that represents motion clips as 2D images and constructs a finite-scale codebook without requiring token lookups. This method retains essential information and expands the capacity of the motion encoder, enhancing the ability of large motion models to leverage large-scale motion data.

## 2 RELATED WORK

### 2.1 LARGE LANGUAGE MODELS AND MULTI-MODALITY

Substantial advancements have been made in enhancing LLMs (Brown et al., 2020; Raffel et al., 2020; Chowdhery et al., 2022) with the ability to understand and respond to human instructions, through a technique known as instruction tuning (Ouyang et al., 2022). Recent research has extended these capabilities to the multimodal domain (Ye et al., 2023; Zheng et al., 2023), with notable work by Liu et al. (2023), who pioneered visual instruction tuning to create a highly adaptable visual assistant. Additionally, Li et al. (2023a) integrated multimodal context directly into instruction data to further enhance model performance. Subsequent studies (Zhang et al., 2023b; Zhao et al., 2023) expanded this research by scaling up instructional datasets and incorporating image-rich text. Notably, Dai et al. (2023) developed InstructBLIP, based on BLIP-2 (Li et al., 2023b), which features an advanced visual feature extraction mechanism to improve performance across vision-language tasks. Despite these breakthroughs, the application of multimodal models to human motion remains less competitive compared to current state-of-the-art (SoTA) methods, although recent initiatives are beginning to explore this domain (Jiang et al., 2023; Zhang et al., 2024b).

### 2.2 VECTOR QUANTIZATION

Vector quantization (VQ) has been highly successful in generating high-quality images (Van Den Oord et al., 2017) and videos (Gupta et al., 2022; Yan et al., 2021). VQ-VAE first converts images into discrete representations and autoregressively models their distribution. Building on this, Lee et al. (2022) introduced residual quantization (RQ), which encodes images into a stacked map of discrete codes, efficiently reducing the spatial resolution of features. You et al. (2022) further developed hierarchical vector quantization (HQ), employing a pyramid scheme with two-level codes for image encoding. Most existing motion generation approaches have adopted VQ or its variants to quantize human motions. However, the small codebook size in traditional VQ methods limits their ability to generalize and accurately represent the diversity of human motions. Although increasing the codebook size can improve representational capacity, it often leads to codebook collapse. Recently, Mentzer et al. (2023) demonstrated that discrete codes can be obtained via scalar quantization, where each scalar entry is independently quantized to the nearest integer through rounding. Similarly, Yu et al. (2023) introduced a lookup-free codebook that maps videos into compact discrete tokens, utilizing all codes without auxiliary losses and expanding the codebook size.

### 2.3 HUMAN MOTION GENERATION

The task of motion generation involves creating human motion based on various inputs, such as text descriptions (Guo et al., 2022b; Petrovich et al., 2022), action labels (Cervantes et al., 2022; Guo et al., 2020) or motion prefixes (Liu et al., 2022; Mao et al., 2019). Among these, text-to-motion (T2M) generation has received the most attention due to the ease and flexibility of using natural language as input. Early approaches (Fragkiadaki et al., 2015; Ghosh et al., 2017; Gopalakrishnan et al., 2019) rely on deterministic motion modeling, which often produce averaged, blurry results. To overcome this, researchers introduce stochastic methods using models like GANs (Cai et al., 2018; Wang et al., 2020) or VAEs (Aliakbarian et al., 2020). For instance, T2M-GPT (Zhang et al., 2023a) extends the temporal VAE to capture the probabilistic relationship between text and motion. Recently, Guo et al. (2024) proposed integrating residual quantization and masked modeling

Table 1: Comparison with existing human motion datasets. More details can be found in our appendix. In the table, B, H, and F refer to body, hand, and face, respectively. "part" indicates that the text captions include fine-grained descriptions of body parts, while "body" means the descriptions are not as detailed. "multi" and "single" specify whether the dataset contains multi-person scenarios or only single-person data. Our MotionBase is the largest motion generation dataset and benchmark to date, featuring at least 15× more data than previous datasets, along with additional modalities.

| | SEQ NUMBER | MOTION | TEXT | RGB | DEPTH | BBOX | PERSON |
|---|---|---|---|---|---|---|---|
| KIT (Plappert et al., 2016) | 5.7K | B | body | ✗ | ✗ | ✗ | single |
| HumanML3D (Guo et al., 2022a) | 29.2K | B | body | ✗ | ✗ | ✗ | single |
| MotionX (Lin et al., 2024) | 81.1K | B,H,F | body | ✓ | ✗ | ✗ | single |
| MotionBase-V1 | >1M | B,H | part | ✓ | ✓ | ✓ | multi |

to improve traditional vector quantization (VQ). Lu et al. (2023) designed a hierarchical VQVAE to separately encode body and hand motions. To better align with a motion auto-encoder, Motion-CLIP (Tevet et al., 2022) incorporates CLIP (Radford et al., 2021) as the text encoder, bringing in more robust text priors. Additionally, Zhang et al. (2024b) and Jiang et al. (2023) explored the development of unified models based on LLMs which accept multimodal conditions (e.g., vision, text, and pose), enabling the generation of subsequent, preceding, or "in-between" motions. Despite leveraging the power of LLMs, these large motion models remain limited to in-domain text instructions and do not yet perform as competitively as existing SoTA methods.

In this work, we aim to bridge the gap between large language models and generalized, reliable large motion models. To achieve this, We begin by introducing MotionBase — a novel, large-scale dataset designed to support extensive pretraining and comprehensive fair evaluation.

## 3 MotionBase Dataset

Data is the foundation of large motion models. With advancements in fields like human pose detection, we are now able to extract high-quality motion sequences from vast amounts of online videos, including datasets like InternViD (Wang et al., 2023) and WebVid (Bain et al., 2021). In its initial public release, our MotionBase contains over one million motion clips, each annotated with fine-grained automatic pseudo labels. A comparison with existing benchmarks is presented in Table 1. Our data collection pipeline involves the following key steps in order.

❶ **Source Video Collection and Cleaning:** We begin by collecting over 20 million videos from publicly available datasets and online platforms such as YouTube. To ensure quality and relevance, we filter out videos that do not contain human figures.

❷ **2D-3D Keypoint Estimation:** Keypoints are essential for capturing the skeletal structure of human motion. Initially, we estimate whole-body 2D keypoints with confidence scores using a pretrained model (Xu et al., 2022). To further enhance motion accuracy, we estimate precise 3D keypoints with another pretrained model (Sárándi et al., 2023) trained on large 3D datasets, Following the method of Lin et al. (2024), we apply temporal smoothing and enforce 3D bone length constraints during triangulation, improving the stability and consistency of the keypoint estimations.

❸ **Incorporating Additional Modalities:** A comprehensive understanding of human motion benefits from the inclusion of diverse modalities such as RGB and depth data. To enrich MotionBase, we provide annotations for these additional modalities. Furthermore, MotionBase includes videos featuring multi-person scenarios, with each motion sequence grounded in its corresponding video through object-level bounding boxes. Although this paper primarily focuses on the text-to-motion task, these additional modalities open avenues for future research in other areas.

❹ **Local-Global Pose Estimation:** We begin by registering the body model SMPL-X (Pavlakos et al., 2019) for each frame in MotionBase, which leverages keypoints based on progressive learning-based mesh fitting method (Lin et al., 2024). Specifically, we predict SMPL-X parameters using a pretrained body mesh recovery method, OSX (Lin et al., 2023), followed by iterative optimization to fit the parameters to the target 2D and 3D joint positions. After fitting, we apply global motion optimization based on Yuan et al. (2022) to refine both global motions and camera poses simulta-

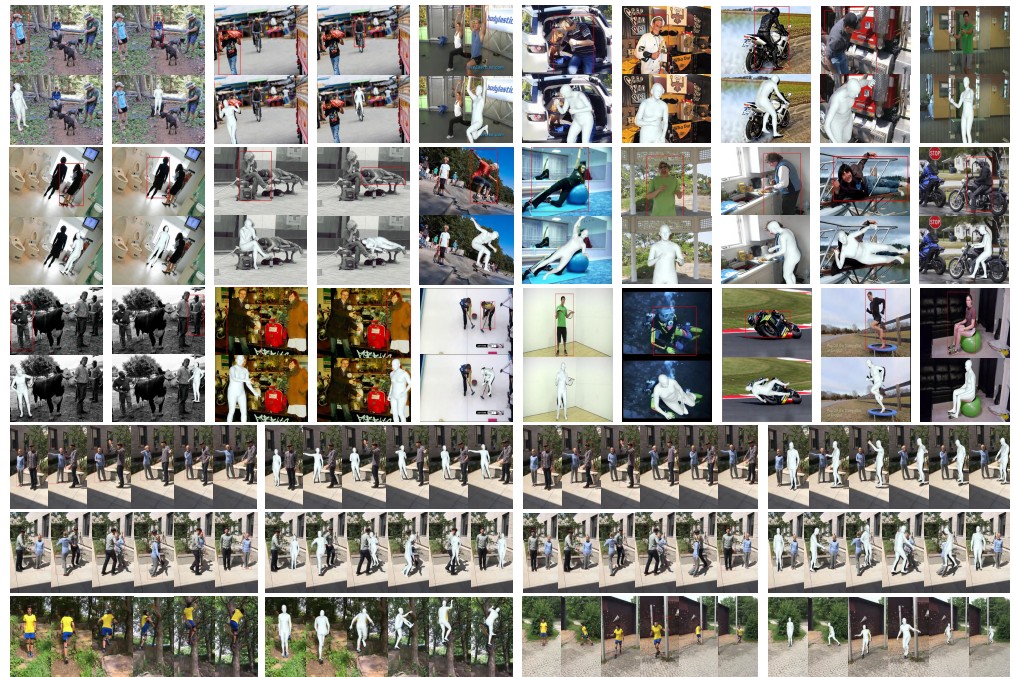

Figure 2: Examples from **MotionBase**, which encompasses a diverse range of human motions, including both long-term clips and static snapshots. It features various scenes, ranging from outdoor environments to indoor settings, and includes both clean, single-person scenarios as well as crowded, multi-person scenes. Additionally, MotionBase comprises a mix of real-world data and synthetic data generated by game engines. For more details about MotionBase, please refer to Appendix A.

neously, ensuring alignment with the video evidence. Finally, for motions with noisy or occluded input data, we reconstruct complete and plausible motions using RoHM (Zhang et al., 2024a).

❺ **Hierarchical Motion Descriptions:** Existing benchmarks face inherent limitations in their text descriptions. Previous studies (Guo et al., 2022a) typically use a single sentence to describe whole-body motions, neglecting finer details of individual body parts, such as the arms or legs. This approach restricts the model's ability to perform more nuanced body comprehension and flexible part-level motion control (e.g., raising only the left arm). Moreover, the richness of text labels often varies across different motions; for example, a large portion of the Motion-X dataset provides only action labels. In contrast, MotionBase offers hierarchical textual annotations for each video inspired by Pi et al. (2023). We carefully design a prompt format and use Gemini-1.5-pro (Reid et al., 2024) to generate detailed descriptions for individual body parts (e.g., left arm, right leg), assigning a dedicated sentence to each. Additionally, we summarize the overall body movement in a paragraph containing 1–3 sentences, providing a more comprehensive description of the motion.

## 4 SCALING UP LARGE MOTION MODEL

### 4.1 OVERALL ARCHITECTURE

Similar to previous LLM-based multimodal models, we treat motion as a foreign language. The overall framework is presented in Figure 11 in Appendix B. Our large motion model, built on a pre-trained LLM, functions as a generative model that connects a motion tokenizer with the LLM backbone $\Theta$. The motion tokenizer encodes raw motion clip features $\mathcal{M}$ into token embeddings $\mathcal{V} = \{v_1, v_2, ..., v_n\} \in \mathbb{R}^{n \times d}$, where $n$ denotes the number of motion tokens and $d$ represents the dimensionality of each token. To integrate motion tokens into the LLM framework, we incorporate $K$ discrete codes in the motion codebook as additional vocabulary for the LLM. Additionally, we introduce two special tokens, <mot> and </mot>, to signify the start and end of motion sequences within the input/output streams. The LLM backbone $\Theta$ is built on a decoder-only architecture using

causal transformers. The model generates outputs $\mathcal{Y} = \{y_1, y_2, ..., y_m\}$ in an auto-regressive manner, where $\mathcal{Y}$ corresponds to the generated motion sequence based on the provided motion-text input tokens. In this work, each motion-text pair in the MotionBase dataset is framed as an instruction-following instance $\{\mathcal{X}_Q, \mathcal{X}_M\}$, representing a question-answer interaction between the user and the motion model. The entire instructional dataset adheres to this unified format. To train our model, we optimize the negative log-likelihood over the predicted tokens which is defined as follows:

$$\mathcal{L}(\Theta) = -\sum_{j=1}^{L} \log P_\Theta(y_j | desc, \hat{y}_{1:j-1}), \tag{1}$$

where $\hat{y}$ and $y$ denote the input and target token sequences, respectively. $\Theta$ represents the model parameters, and $L$ is the length of the target sequence. The input description, $desc$, can be empty depending on the instruction provided.

### 4.2 2D Lookup-free Motion Quantization

Similar to visual tokenization, motion tokenization is a process that compresses motion signals into a series of discrete tokens, typically involving an encoder $\mathbb{E}$, a decoder $\mathbb{D}$ and a codebook $\mathbb{C}$. We propose a 2D lookup-free quantization method as a key component for building large motion models.

**2D Motion Quantization.** Traditional motion quantizers use 1D embeddings to represent motion at each timestamp, which inevitably results in the loss of crucial information. Furthermore, this approach limits the quantizer's ability to generate and interpret part-level motions. To address these limitations, we treat the motion sequence $\mathcal{M} = \{m_1, m_2, ..., m_T\}$ as a single-channel image, representing each motin sequence as $\mathcal{M} \in \mathbb{R}^{T \times D \times 1}$. Each motion embedding $m_i$ is divided into $P$ components, capturing distinct features of motion, such as root orientation, joint rotation and foot contact. Our motion encoder then converts $\mathcal{M}$ into a feature map $\mathbb{E}(\mathcal{M}) \in \mathbb{R}^{\lfloor T/\alpha \rfloor \times P \times d}$, where $\alpha$ denotes the temporal downsampling ratio. This approach ensures that each body part is tokenized separately, allowing for more granular, part-level motion encoding and decoding.

**Lookup-Free Quantization.** Traditional motion quantizers are often constrained by small codebook sizes, restricting their ability to capture the full diversity of human motion. A common approach is to expand the motion vocabulary. However, excessively enlarging the codebook can result in "codebook collapse", where only a small subset of tokens in the codebook is used, offering minimal performance improvements. In some cases, an overly large vocabulary can even degrade the model's overall performance. To address this, a more effective way is to reduce the dimensionality of code embeddings (Mentzer et al., 2023), which limits the representational capacity of individual tokens and encourages more efficient learning across a larger vocabulary. Similar to Yu et al. (2023), we reduce the embedding dimension of the codebook to zero by replacing the codebook $\mathbb{C} \in \mathcal{R}^{K \times d}$ with an integer set $\mathbb{C}$ with $|\mathbb{C}| = K$. Specifically, $\mathbb{C}$ is the Cartesian product of single-dimensional variables $\mathbb{C} = \times_{i=1}^{d} C_i$, where $C_i = \{-1, 1\}$ and $d$ is equal to $\log_2 K$. Given a feature vector $z \in \mathbb{R}^d$, our quantizer $Q(\cdot)$ converts each dimension of the quantized representation into:

$$Q(z_i) = \arg\min_{c_{ik}} ||z_i - c_{ik}|| = -\mathbb{1}\{z_i \leq 0\} + \mathbb{1}\{z_i > 0\}, \tag{2}$$

where $c_{ij}$ denotes the $j$-th value of $C_i$. The token index is computed as $Index(z) = \sum_{i=1}^{d} 2^{i-1} \mathbb{1}\{z_i > 0\}$. To train the tokenizer, we employ a standard combination of reconstruction, perceptual, and commitment losses, along with an entropy penalty to promote better codebook utilization (Yu et al., 2023). Importantly, we exclude the use of GAN loss, as it was found to negatively impact training stability.

## 5 Experiments

### 5.1 Experimental Setup

**Datasets.** Our investigation first is conducted on the following text-to-motion datasets: HumanML3D (Guo et al., 2022a) and Motion-X (Lin et al., 2024). HumanML3D comprises 14,616 motion clips sourced from the AMASS dataset (Mahmood et al., 2019), paired with 44,970 textual descriptions. Motion-X, a more recent dataset, includes approximately 81,000 motion clips. To

validate our conclusions on larger-scale data, we also carry out experiments on the proposed MotionBase dataset with two variants: MotionBase-0.5 and MotionBase-1.0. MotionBase-0.5 contains 500,000 clips, while MotionBase-1.0 encompasses the full scope of our collected data, with over 1 million clips. Following standard practice, each dataset is split into training, validation, and test sets in proportions of 85%, 5%, and 15%, respectively.

**Evaluation Metrics.** For the motion generation task, we employ the following metrics in our experiments following Guo et al. (2022a). (1) Frechet Inception Distance (FID): This metric assesses overall motion quality by measuring the distributional difference between the high-level features of generated motions and real motions. (2) Motion-retrieval Precision (R-Precision) and Multimodal Distance (MMDist): These metrics evaluate the semantic alignment between the textual input and generated motions. R-Precision measures the top-1/2/3 retrieval accuracy, while MMDist computes the distance between matched text and motion pairs. Additionally, we validate our motion tokenizer by conducting experiments on the motion reconstruction task. This is measured using both Mean Per Joint Position Error (MPJPE) and FID. MPJPE quantifies the average distance (in millimeters) between the predicted joint positions and the ground truth positions across all joints in the skeleton.

**Implementation Details.** For the motion tokenizer, we implement a VQ codebook $\mathbb{C} \in \mathbb{R}^{1024 \times 512}$ with an embedding dimensionality of $d = 512$, and the resulting discrete codes are incorporated as additional vocabulary for the LLM. In comparison, our lookup-free codebook has a size of $2^{16} = 16384$, where the least frequently used tokens from the LLM's codebook are mapped to represent motion codes. The motion encoder $\mathbb{E}$ operates with a temporal downsampling rate of $\alpha = 4$. We experiment with four LLM architectures to build our large motion model: GPT2-medium (Radford et al., 2019), Llama-2-7b, Llama-2-13b (Touvron et al., 2023b), and Llama3.1-8b (Dubey et al., 2024). The motion tokenizer is trained with a learning rate of 1e-4 and a batch size of 256 over 300K iterations. For training the large motion model, full parameter tuning is performed on 8×A800 GPUs, with a batch size of 1024, over 300 epochs. The learning rate is set to 2e-4 for GPT2-medium and 2e-5 for the Llama models. Further details are provided in the appendix due to space limitation.

Table 2: Comparisons under different model and data sizes. All experiments are conducted using the same pretrained VQ model for consistency. Additionally, we re-train the motion autoencoder and text encoder (Guo et al., 2022a) separately on the Motion-X and MotionBase datasets, using their respective data to train the motion autoencoder for each dataset's evaluation.

| Decoder | #Inst. | #Param. | Motion-X | | | MotionBase | | |
|---------|--------|---------|----------|----------|--------|------------|----------|--------|
| | | | R@1 ↑ | R@3 ↑ | FID ↓ | R@1 ↑ | R@3 ↑ | FID ↓ |
| Real | - | - | 0.496 | 0.821 | 0.038 | 0.290 | 0.563 | 0.011 |
| GPT-2 | 0.02M | 355M | 0.206 | 0.402 | 54.017 | 0.037 | 0.109 | 125.824 |
| GPT-2 | 0.08M | 355M | 0.468 | 0.791 | 0.096 | 0.055 | 0.155 | 124.230 |
| GPT-2 | 0.5M | 355M | 0.358 | 0.618 | 4.852 | 0.252 | 0.533 | 0.636 |
| GPT-2 | 1M | 355M | 0.357 | 0.614 | 5.083 | 0.264 | 0.542 | 0.516 |
| LLaMA-2 | 0.02M | 7B | 0.207 | 0.405 | 53.354 | 0.041 | 0.109 | 113.189 |
| LLaMA-2 | 0.08M | 7B | 0.471 | 0.794 | 0.159 | 0.074 | 0.185 | 127.664 |
| LLaMA-2 | 0.5M | 7B | 0.372 | 0.627 | 4.908 | 0.256 | 0.522 | 1.084 |
| LLaMA-2 | 1.0M | 7B | 0.351 | 0.602 | 5.582 | 0.263 | 0.536 | 0.545 |
| LLaMA-3 | 0.02M | 8B | 0.217 | 0.418 | 54.004 | 0.039 | 0.102 | 117.561 |
| LLaMA-3 | 0.08M | 8B | 0.483 | 0.802 | **0.103** | 0.071 | 0.183 | 125.310 |
| LLaMA-3 | 0.5M | 8B | 0.363 | 0.625 | 4.798 | 0.256 | 0.533 | 0.512 |
| LLaMA-3 | 1M | 8B | 0.354 | 0.611 | 5.100 | 0.266 | 0.557 | **0.394** |
| LLaMA-2 | 0.02M | 13B | 0.225 | 0.436 | 53.447 | 0.040 | 0.107 | 117.594 |
| LLaMA-2 | 0.08M | 13B | **0.486** | **0.805** | 0.132 | 0.074 | 0.186 | 126.999 |
| LLaMA-2 | 0.5M | 13B | 0.375 | 0.636 | 4.792 | 0.259 | 0.520 | 0.511 |
| LLaMA-2 | 1.0M | 13B | 0.359 | 0.612 | 5.370 | **0.298** | **0.599** | 0.595 |

## 5.2 DISCUSSION OF SCALING UP MOTION GENERATION

In this section, we investigate the impact of model size and data scale on motion generation performance. We utilize the motion autoencoder (Guo et al., 2022a) retrained on Motion-X and MotionBase datasets to evaluate performance on their respective test sets. We categorize our training data

into four scales: 0.02M (HumanML3D only), 0.08M (Motion-X only), 0.5M (MotionBase-0.5), and 1M (MotionBase-1.0). To ensure fair comparison, we employ the same VQ as the motion tokenizer, maintaining consistency across experiments to validate our conclusions.

**Does increasing model size benefit motion generation?** Yes. As shown in Table 2, our results demonstrate that increasing model size leads to significant performance improvements when provided with the same amount of training data. Specifically, Llama2-13b outperforms Llama2-7b, which in turn surpasses GPT2-medium, illustrating a clear trend of performance gains as model capacity increases. This suggests that models with larger size are better equipped to capture diverse, complex patterns and relationships within human motions.

**Does increasing data scale benefit motion generation?** Yes. In Table 2, when using the same foundation model, increasing the scale of training data leads to substantial improvement on MotionBase test set, aligning with our expected scaling laws. This improvement is particularly pronounced in the R-precision metric, emphasizing the critical role of data scale in enhancing semantic alignment between generated motions and text prompts. However, contrary to our expectations, we observe a noticeable performance decline on Motion-X test set if not trained on Motion-X (0.08M). We attribute this to the limitations of the retrieval-based evaluation model, as discussed in Section 5.4.

**Does the large motion model perform SoTA competitively?** We evaluate our large motion model on the widely adopted HumanML3D benchmark. We compare its performance against a variety of SoTA approaches. This includes diffusion-based methods such as MLD (Chen et al., 2023) and Motion-Diffuse (Zhang et al., 2022), as well as the GPT-based T2M-GPT (Zhang et al., 2023a). We also compare against LLM fine-tuning methods like MotionGPT (Jiang et al., 2023; Zhang et al., 2024b), MotionLLM (Wu et al., 2024), and AvatarGPT (Zhou et al., 2024). As shown in Table 3, our model, which utilizes Llama-2-13B as the decoder and calculates the loss over the entire concatenated sequence of input

Table 3: Comparison with existing SoTA methods on the HumanML3D benchmark. Results marked with ∗ represent values reproduced using the officially released code, while unmarked results are taken from the original papers.

| | Decoder | R@1↑ | R@3↑ | FID↓ | MMDist↓ |
|---|---|---|---|---|---|
| Real | - | 0.511 | 0.797 | 0.002 | 2.974 |
| MLD | - | 0.481 | 0.772 | 0.473 | 3.196 |
| MotionDiffuse | - | 0.491 | 0.782 | 0.630 | 3.113 |
| T2M-GPT | GPT-2 | 0.492 | 0.775 | **0.141** | 3.121 |
| MotionGPT[1],∗ | T5 | 0.409 | 0.667 | 0.162 | 3.992 |
| MotionGPT[1] | T5 | 0.492 | 0.778 | 0.232 | 3.096 |
| MotionGPT[2],∗ | Llama-2-13B | 0.367 | 0.654 | 0.571 | 3.981 |
| MotionGPT[2],∗ | Llama-1-13B | 0.363 | 0.633 | 0.592 | 4.029 |
| MotionGPT[2] | Llama-1-13B | 0.411 | 0.696 | 0.542 | 3.584 |
| MotionLLM | Gemma-2b | 0.482 | 0.770 | 0.491 | 3.138 |
| AvatarGPT | Llama-1-13B | 0.389 | 0.623 | 0.567 | - |
| Ours | Llama-2-13B | **0.519** | **0.803** | 0.166 | **2.964** |

text, achieves SOTA performance. Our large motion model significantly outperforms other LLM-based methods such as MotionGPT and AvatarGPT, as well as the earlier T2M-GPT. In particular, we observe substantial improvements in key metrics such as R@1, R@3, and MMDist, highlighting our model's ability to generate motion sequences that are better aligned with text descriptions and of higher quality.

**Slow convergence of large motion models.** To evaluate the convergence speed of large motion models, we train GPT-2, Llama2-7b, and Llama3.1-8b for 300 epochs on Motion-X. The training curve of with R@1 performance is illustrated in Figure 3. We obverse that all large motion models nearly converge by 200 epochs, with larger models converging faster. Initializing these models with pre-trained weights proves beneficial for speeding up convergence. Compared to large multimodal models like LLaVA (Liu et al., 2023), large motion models require more epochs to capture the complex representations of motion sequences. We attribute the slow convergence of these models to the limited representation capacity of the motion tokenizer, which contains only 512 motion tokens. This suggests the need to optimize the motion tokenizer and expand its rep-

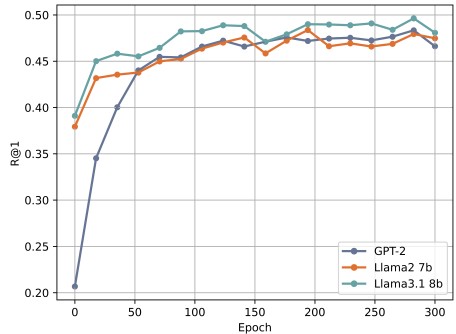

Figure 3: Training curves with Y-axis denoting R@1 retrieval accuracy. All these models are trained for 300 epochs at most and are evaluated every 1000 steps.

resentation space. To address this, we explore 2D-LFQ quantization method as a promising alternative.

**Does Static and Synthetic Data help?** Yes, the addition of static image data and synthesized data both contribute to improvements, as illustrated in Table 4, more analysis can be found in Appendix C.1.

**Do large motion models outperform in out-of-distribution setup?** Yes. We present the results in Table 5. This ablation is essential for further validating the generalization capabilities of large motion models, as the improvements observed in Table 2 may stem from the inclusion of additional in-domain data from Motion-X. In this setup, we select

Table 4: Ablation of the effectiveness of synthetic and static data, which takes about 28% and 44% of all data, respectively.

| TRAIN SET | R@1 ↑ | R@3 ↑ | FID ↓ |
|---|---|---|---|
| Real | 0.290 | 0.563 | 0.011 |
| w/o static & syn | 0.111 | 0.248 | 57.719 |
| w/o static | 0.120 | 0.252 | 55.983 |
| MotionBase | **0.264** | **0.542** | **0.516** |

four subsets from MotionBase, comprising 90K samples (UNSEEN-90K), for evaluation, while the remaining 38 subsets are used for training. This ensures that the test set consists entirely of out-of-domain (OOD) samples. We compare the performance of models trained on HumanML3D, MotionX, and Motion-#38, all utilizing the GPT2-medium architecture, where $\#N$ denotes the number of training subsets. All models are trained using the GPT2-medium. The results on the OOD test set clearly demonstrate that the model trained on MotionBase significantly outperforms those trained on HumanML3D and MotionX, particularly in terms of R@1 and R@3 metrics. These findings strongly highlight the superior generalization ability of large motion models when handling unseen OOD data, especially when trained on diverse, large-scale datasets. However, we once again observe unexpected results with the FID metric, which will be discussed further in Section 5.4.

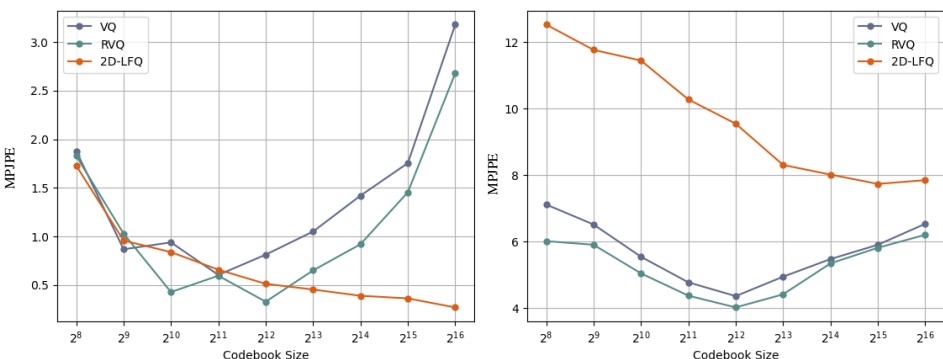

Figure 4: Comparison with different motion quantization on Motion-X (**left**) and MotionBase (**right**). Note that we only show MPJPE (↓) results here. FID results is shown in Appendix C.9.

## 5.3 DISCUSSION OF MOTION QUANTIZATION

In this section, we investigate the impact of different motion quantization methods. We compare our proposed 2D lookup-free quantization (2D-LFQ) against two commonly used approaches: residual vector quantization (RVQ) and vector quantization (VQ), across various codebook sizes ranging from $2^8$ to $2^{16}$. The number of parameters for RVQ/VQ and 2D-LFQ are 19.43M and 108.35M, respectively. As shown in Figure 4, 2D-LFQ demonstrates significant improvements over both RVQ and VQ. Notably, as the codebook

Table 5: Ablation of out-of-domain evaluation on UNSEEN-90K dataset, where $\#N$ denotes we use $N$ subsets of MotionBase for training.

| TRAIN SET | R@1 ↑ | R@3 ↑ | FID ↓ |
|---|---|---|---|
| Real | 0.147 | 0.349 | 0.005 |
| HumanML3D | 0.032 | 0.101 | 204.833 |
| MotionX | 0.042 | 0.119 | 178.368 |
| MotionBase-#38 | **0.136** | **0.321** | **10.613** |

size increases, 2D-LFQ continues to enhance performance, while RVQ and VQ experience diminishing returns or performance degradation with larger codebooks. Our deeper analysis attributes these gains to better codebook utilization by 2D-LFQ. Figure 5 illustrates that the utilization rates

for VQ and RVQ begin to decline once the codebook size exceeds $2^{10}$, which corresponds to the peak performance for these methods, whereas the utilization of 2D-LFQ continues to increase with larger codebooks. Additionally, we conduct further experiments to validate the benefits of 2D motion encoding in Appendix C.9.

## 5.4 LIMITATION OF AUTOMATED METRIC

As mentioned earlier, the FID scores in Table 2 and Table 5 yield unexpected results. Specifically, when evaluating on Motion-X and UNSEEN-90K, FID achieves its best performance when trained on Motion-X, significantly outperforming both the smaller HumanML3D and the larger-scale Motion-Base. In this section, we aim to investigate this anomaly. FID, a standard metric widely used for generation tasks, is typically measured by a pre-trained evaluator. In traditional image generation, FID is calculated using a well-trained, robust visual encoder like InceptionNet (Szegedy et al., 2015), which is trained on millions of images. However, the evaluator currently used to compute FID for motion generation is a simple motion autoencoder with a very small parameter scale (Guo et al., 2022a). Since

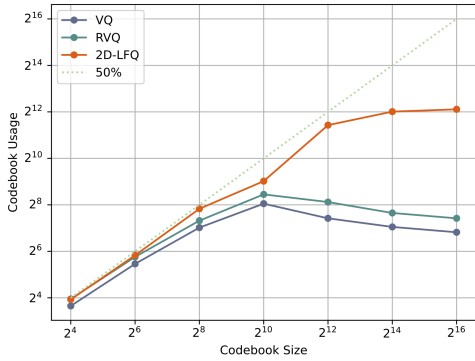

Figure 5: Comparison of codebook utilization for different motion quantization.

this motion autoencoder is trained on limited data consisting of only 20K motions, we argue that it may lack the generalization needed for robust performance, leading to difficulties in reliably capturing the complex semantic alignment between text and motion.Similar unexpected results occur in motion reconstruction as well. As show in Table 6, the FID score on HumanML3D is two orders of magnitude higher when comparing 2D-LFQ and VQ-VAE, despite the former achieving a much lower MPJPE. When tested on MotionBase, 2D-LFQ obtains the highest FID score even while achieving the best MPJPE. We observe the same issue with other metrics like MMDist, as discussed in Appendix C.1. Notably, Voas et al. (2023) have mentioned that existing metrics are sensitive to the quality of the embedding space and do not always align with human perception. These findings highlight the need for a more robust and fair metric for large motion models moving forward.

Table 6: Robustness investigation of the evaluation metrics on the motion reconstruction task.

| Tokenizer | #Num. | #Param. | HumanML3D | | Motion-X | | MotionBase | |
| | | | FID↓ | MPJPE↓ | FID | MPJPE | FID | MPJPE |
|---|---|---|---|---|---|---|---|---|
| VQ-VAE | 512 | 19.43M | 0.078 | 69.2 | 0.852 | 106.4 | 4.366 | 123.6 |
| RQ-VAE | 512 | 19.43M | **0.05** | **37.5** | 0.568 | 56.9 | **4.026** | 78.2 |
| 2D-LFQ | 16384 | 108.35M | 1.769 | 45.6 | **0.295** | **54.1** | 7.853 | **64.1** |

## 6 CONCLUSION

In this paper, we explore how to advance the field of large-scale motion generation. To this end, we introduce a large-scale motion dataset named MotionBase, which includes detailed text descriptions and rich modality annotations, providing a strong foundation for effectively training large motion models. Our research highlights key findings, such as the impact of scaling both data and model size. Additionally, we identify potential limitations in the current evaluation metrics, particularly when assessing diverse and unseen motions. To enhances the benefits large motion models can derive from extensive motion data, we propose a novel motion quantization approach that treats motion clips as 2D images and constructs a finite-scale codebook, eliminating the need for token lookups. We hope that this research offers valuable direction for future work in large-scale motion generation.

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

# Appendices

## A ADDITIONAL DETAILS OF MOSEBASE

In this section, we provide more details about **Motionbase** that are not included in the main paper due to spatial limitations.

### A.1 STATISTIC ANALYSES

MotionBase contains over 1 million motion sequences from 42 different public datasets and web videos on the Internet. Subsets of MotionX, including Animation, Perform, Dance, Aist, Kungfu, GRAB (Taheri et al., 2020), Music, Idea400 (Lin et al., 2024), HAA500 (Chung et al., 2021), Game Motion, and Fitness, are included in MotionBase. Recognizing the high cost of collecting and annotating videos, we also see the untapped potential of images for motion understanding. Consequently, MotionBase incorporates image data by repeating each image across 64 frames and treating it as a motion sequence. For the datasets with long-range videos, such as MPI-INF-3DHP (Mehta et al., 2017), we segment the footage into sub-clips with random durations ranging from 10 seconds to one minute. Figure 6 and Figure 7 illustrate the scale and length distributions of MotionBase.

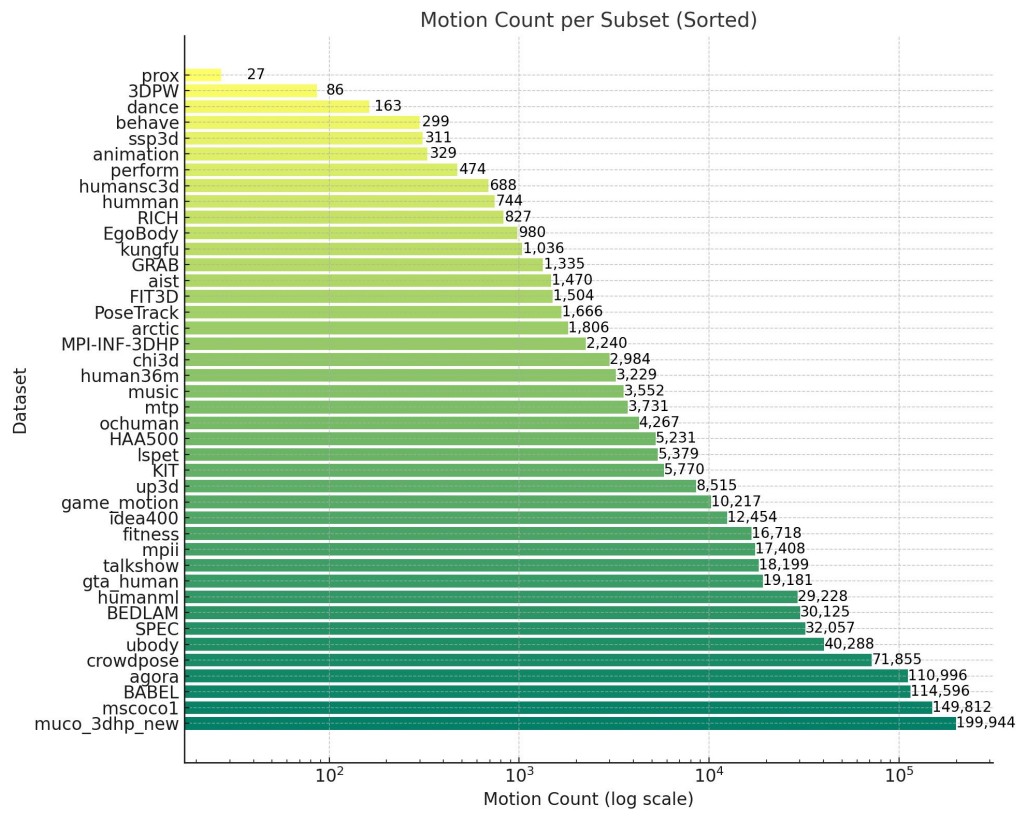

Figure 6: The scale distribution of motion sequences across subsets of MotionBase.

### A.2 PROMPT OF MOTION DESCRIPTION

In this paper, we use Gemini-1.5-pro (Reid et al., 2024) and GPT-4o-mini (OpenAI, 2024) as large multimodal models (LMM) to generate textual annotations for video and image data, respectively. For each person-centric sample, we first crop and track the person's body using the corresponding bounding box(es). The LMM is then tasked with focusing on the person's physical movements and positions in the global space to generate detailed descriptions. Unlike previous datasets, we provide

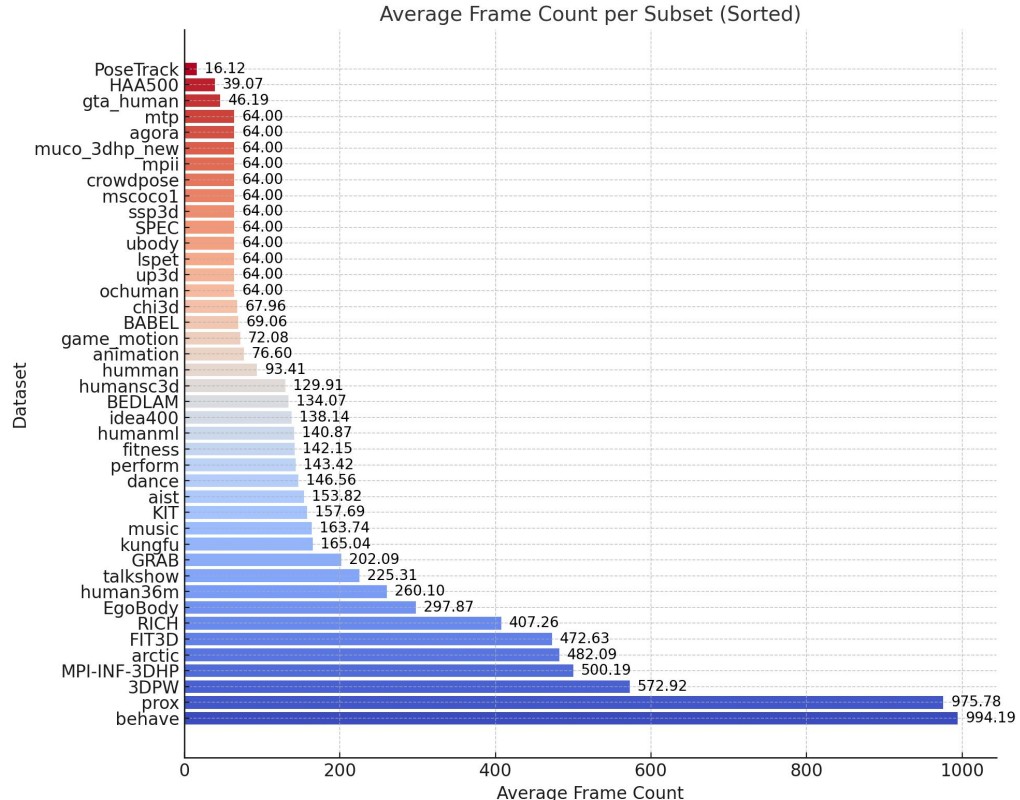

Figure 7: The length distribution across different subsets of MotionBase

more granular motion descriptions by dividing the body into upper and lower sections, prompting the LMM to generate part-specific descriptions ("part-level"). Additionally, an overall summary of the entire body's movement ("whole-body") is also produced. Figure 8 illustrates the prompt used to caption human motion sequences in MotionBase.

### A.3 WORD DISTRIBUTION ANALYSIS

To further explore the annotated motion text, we generate word clouds from the entire text corpus in MotionBase. Since the annotations in MotionBase consist of both whole-body and part-level descriptions, we create separate word clouds for general labels and more detailed annotations, as shown in Figure 9 and Figure 10, respectively. In Figure 9, we observe that the whole-body annotations primarily highlight high-level motion activities, such as standing, sitting, and walking. In contrast, Figure 10 shows that part-level annotations focus more on specific body movements, including the torso, shoulders, legs, and arms. We believe that this hierarchical structure of annotations will enhance the understanding of motion.

## B ADDITIONAL OVERVIEW OF MODEL ARCHITECTURE

Due to space limitations in the main paper, we provide the overview of our model architecture in Figure 11 in this appendix. Following most LMMs, our large motion model consists of two stages: pre-training and fine-tuning. During the pre-training stage, we train a motion encoder, a motion decoder, and a motion codebook to represent motions using discrete tokens. With this motion tokenizer, we fine-tune an autoregressive language model to predict motion tokens. In the inference stage, the input text is processed by the language model to generate motion tokens in an autoregressive manner, which are then decoded into natural motion by the pre-trained motion decoder.

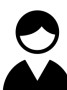

Begin by providing a general overview of the person's current action (e.g., walking, sitting, interacting) within the BBOX area. Then, proceed with a detailed breakdown, focusing exclusively on the physical movements and positions of the person within the BBOX. For the upper body, describe the position and movement of the arms, hands, shoulders, and torso. For the lower body, detail the position and movement of the legs, feet, and overall balance. Ensure the description strictly covers physical actions without mentioning facial expressions, clothing, or environmental elements outside the BBOX.

**Example:**

The person is standing still, observing something in front of them.

- **Upper body：** Their arms hang relaxed by their sides, with the shoulders slightly back and the chest open. The torso is upright, with minimal movement, indicating a calm, neutral stance.

- **Lower body：** Both feet are planted firmly on the ground, shoulder-width apart. The knees are slightly bent, and their weight is evenly distributed between both legs.

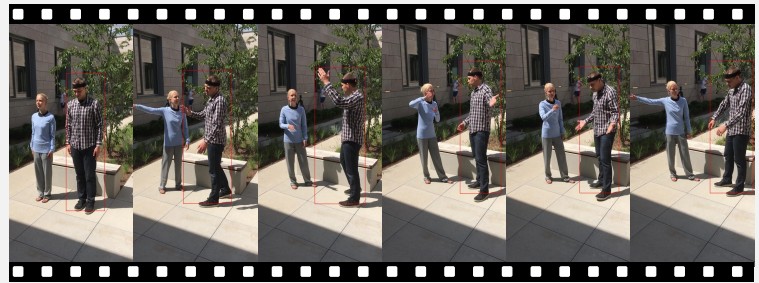

The person is standing within the designated area, engaging in a conversation seemingly directed toward someone positioned off-camera to the left.

**Upper Body:**

* **Arms:** Initially held loosely at the sides, the arms transition to various positions throughout the interaction. At times, they rise to chest level with palms open, suggesting an explanatory gesture. Occasionally, one or both arms extend outwards, indicating direction or emphasis.
* **Hands:** Hand movements correspond with arm gestures. Palms face upwards and outwards during open-handed motions, then relax to a neutral position when the arms are at rest.
* **Shoulders:** Shoulders remain relatively relaxed throughout, with subtle shifts in position reflecting the arm movements. They don't appear tense or raised, implying a generally comfortable stance.
* **Torso:** The torso largely remains stationary, facing forward, with slight turns coinciding with the shifting weight distribution of the lower body.

**Lower Body:**

* **Legs:** Legs maintain a comfortable stance, slightly apart, with the weight appearing balanced. There's a subtle shift in weight distribution as they adjust their stance.
* **Feet:** Feet remain planted on the ground, primarily shoulder-width apart. The positioning suggests a grounded and stable stance.
* **Overall Balance:** The individual appears balanced and at ease throughout the interaction, with movements suggesting engagement in the conversation rather than discomfort or restlessness.

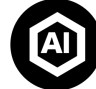

Figure 8: Prompt examples to label human motions in the video. We use Gemini-1.5-pro and GPT-4o-mini to generate motion descriptions for the video and image data, respectively. We provide "whole-body" (UP) and "part-level" (DOWN) labels for each sample in the dataset.

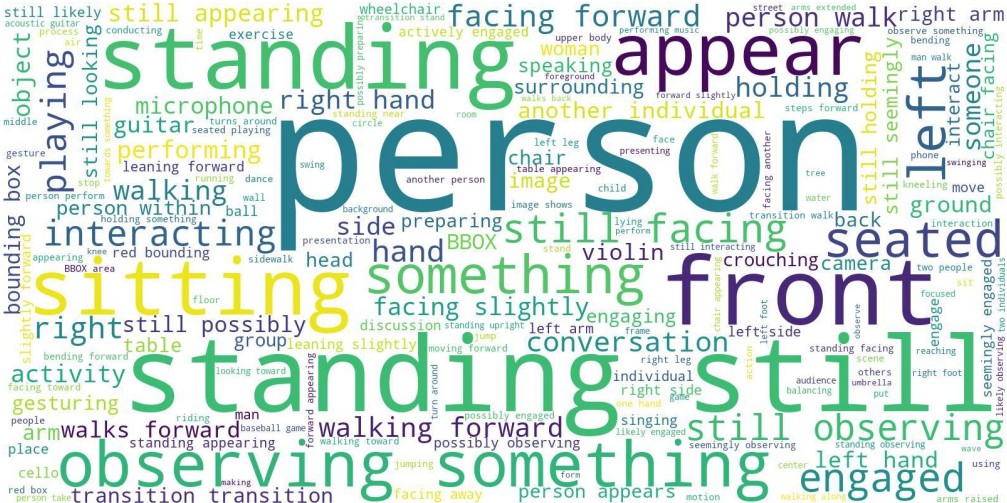

Figure 9: Word cloud of whole-body textual annotation in MotionBase.

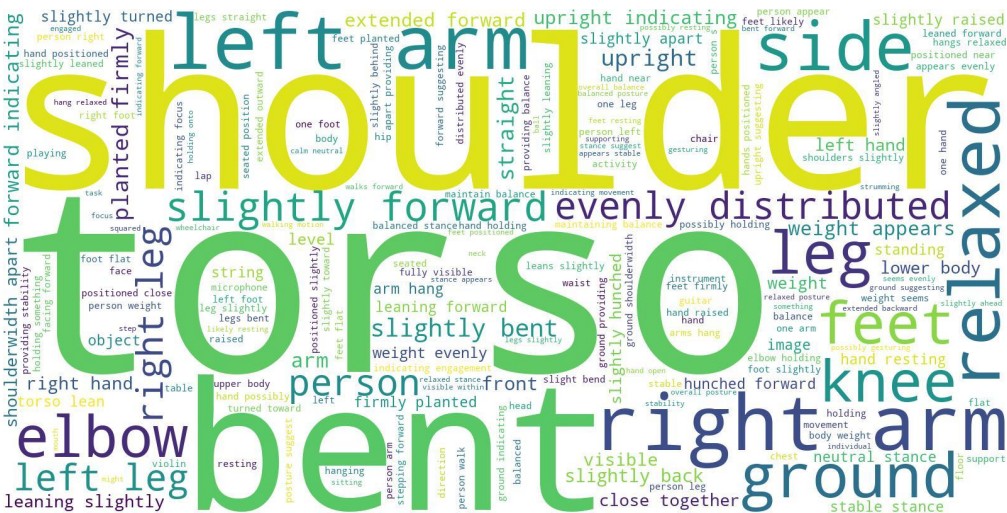

Figure 10: Word cloud of part-level textual annotation in MotionBase.

## C  ADDITIONAL EXPERIMENTAL RESULTS

In this section, we provide more experimental analysis which can not be presented in our main paper due to space limitation.

Table 7: Ablation of the effectiveness of synthetic data and static data.

| TRAIN SET | R@1↑ | R@3↑ | FID↓ | MMDist↓ |
|---|---|---|---|---|
| Real | 0.290 | 0.563 | 0.011 | 3.480 |
| w/o static & syn | 0.111 | 0.248 | 57.719 | 8.412 |
| w/o static | 0.120 | 0.252 | 55.983 | 8.175 |
| MotionBase | **0.264** | **0.542** | **0.516** | **4.007** |

Figure 11: Overview of the large motion model, which can be divided into two stages. In the first stage(**left**), we pre-train a motion VQ-VAE to quantify motion sequences into tokens. In the second stage(**right**), we fine-tune an autoregressive language model to predict motion tokens.

Table 8: Results on the test set with synthetic and static data filtered out.

| TRAIN SET | R@1 ↑ | R@3 ↑ | FID ↓ | MMDist ↓ |
|---|---|---|---|---|
| Real | 0.196 | 0.474 | 0.006 | 1.647 |
| w/o static & syn | 0.167 | 0.396 | 1.740 | 2.323 |
| w/o static | 0.166 | 0.393 | 1.780 | 2.356 |
| MotionBase | **0.168** | **0.399** | **1.614** | **2.300** |

## C.1 ABLATION OF SYNTHESIS AND STATIC DATA

For handling static data, our core strategy is to introduce specific language prompts during training. Specifically, by adding language markers such as "keep the action still," we explicitly guide the model to understand the distinction between static and dynamic actions. Prompt-based methods can effectively differentiate between different motion distributions. To validate this approach, we conduct a series of ablation experiments. We train GPT2-medium on three variations of MotionBase: without synthetic data, without image data, and without both synthetic data and image data. The model is trained for 300 epochs with a learning rate of 2e-4. Using the VQ-VAE and retrieval model trained on MotionBase, we test on the MotionBase test set and a subset of the test set where static and synthetic data are filtered out. The results are shown in Table 7 and Table 8. Our findings indicate that incorporating both static data (i.e., image data) and synthetic data leads to performance improvements in terms of R-Precision.

Table 9: Comparison of evaluations using different encoder models.

| Decoder | #Inst. | #Param. | EM_Humanml3d | | | EM_Motion-X | | |
|---|---|---|---|---|---|---|---|---|
| | | | R@1 ↑ | R@3 ↑ | FID ↓ | R@1 ↑ | R@3 ↑ | FID ↓ |
| Real | - | - | 0.511 | 0.797 | 0.002 | 0.496 | 0.821 | 0.038 |
| GPT-2 | 0.02M | 355M | 0.466 | 0.752 | **0.101** | 0.358 | 0.651 | **0.050** |
| GPT-2 | 0.08M | 355M | 0.462 | 0.744 | 0.208 | 0.362 | 0.656 | 0.754 |
| LLaMA-2 | 0.02M | 7B | 0.497 | 0.778 | 0.214 | 0.378 | 0.671 | 0.122 |
| LLaMA-2 | 0.08M | 7B | 0.474 | 0.758 | 0.452 | 0.376 | 0.673 | 0.518 |
| LLaMA-3 | 0.02M | 8B | 0.500 | 0.783 | 0.173 | 0.380 | 0.675 | 0.094 |
| LLaMA-3 | 0.08M | 8B | 0.499 | 0.786 | 0.264 | 0.393 | 0.696 | 0.591 |
| LLaMA-2 | 0.02M | 13B | **0.519** | **0.803** | 0.166 | 0.395 | 0.695 | 0.105 |
| LLaMA-2 | 0.08M | 13B | 0.504 | 0.790 | 0.393 | **0.400** | **0.700** | 0.637 |

## C.2 ABLATION OF DIFFERENT ENCODER MODELS

Table 9 presents the evaluation results on the HumanML3D test set using different encoder models (EM). We employ the same dual-encoder architecture (Guo et al., 2022a) but trained it on two

Table 10: Comparison between fine-tuning and learning from scratch on the Motion-X test set.

| #Inst | From Sctrach | R@1 ↑ | R@3 ↑ | FID ↓ | MMDist ↓ |
|-------|--------------|-------|-------|-------|----------|
| Real | - | 0.496 | 0.821 | 0.038 | 2.438 |
| 0.02M | Yes | 0.035 | 0.103 | 16.904 | 9.280 |
| 0.02M | No | 0.206 | 0.402 | 54.017 | 8.218 |
| 0.08M | Yes | 0.460 | 0.782 | 0.113 | 2.862 |
| 0.08M | No | 0.468 | 0.791 | 0.096 | 2.798 |

Table 11: Results of different loss calculation methods on the HumanML3D test set.

| Loss Calculation | R@1 ↑ | R@3 ↑ | FID ↓ | MMDist ↓ |
|------------------|-------|-------|-------|----------|
| Real | 0.511 | 0.797 | 0.002 | 2.974 |
| Motion Seq Loss | 0.388 | 0.650 | 0.680 | 3.919 |
| Whole Seq Loss | 0.466 | 0.752 | 0.101 | 3.234 |

distinct datasets: HumanML3D and Motion-X, where HumanML3D is a subset of Motion-X. The results highlight the limited generalization ability of the encoder model. When using the model trained on the larger Motion-X dataset, performance metrics on HumanML3D decrease. This suggests that training on the broader Motion-X dataset negatively impacts R-Precision performance on the HumanML3D subset. Furthermore, when the encoder model is trained on Motion-X, increasing the training data size for the text-to-motion model leads to significant performance gains. Conversely, when using the encoder model trained on HumanML3D, the performance of the text-to-motion model degrades as the training data size increases. This might be attributed to inherent limitations in the encoder model itself.

## C.3 ABLATION OF LEARNING FROM SCRATCH VS. FINE-TUNING

We compare the performance of fine-tuning GPT-2 against training it from scratch (random initialization). As shown in Table 10, fine-tuned models consistently outperform those trained from scratch, particularly when trained on HumanML3D and evaluated on MotionX. The improvement of pretrained LLM highlights the importance of text pre-training in enhancing the model's understanding of text descriptions and improving its generalization capabilities.

## C.4 ABLATION OF DIFFERENT LOSS CALCULATION STRATEGIES

We also investigate the impact of different loss calculation strategies on model performance: We compare two strategies: 1) calculating the loss solely on the output motion tokens, and 2) calculating the loss on both the input text and the output motion tokens. As shown in Table 11, our results indicate that the second strategy yields better performance. This improvement compared to the first alternative is likely due to the strategy's ability to prevent catastrophic forgetting of text understanding. Additionally, it helps mitigate overfitting to motion patterns in the training data, thereby enhancing the model's generalization ability.

## C.5 ABLATION STUDY ON HIERARCHICAL TEXT AND BASIC TEXT

To investigate the effectiveness of hierarchical text representation, we conduct a series of ablation experiments. As shown in Table 12, we compare the training results using hierarchical text with both basic and detailed descriptions, against the results using only basic descriptions. The experimental results demonstrate that hierarchical text can effectively enhance the model's semantic understanding, thereby improving the semantic matching of generated motions.

It is worth noting that the evaluation results for hierarchical text are sometimes overestimated, even surpassing the ground truth. We hypothesize that this is because the evaluator itself is a network

Table 12: Results of Hierarchical Text and Basic Text on MotionBase.

| Training text | R@1 ↑ | R@3 ↑ | FID ↓ | MMDist ↓ |
|---|---|---|---|---|
| Real | 0.290 | 0.563 | 0.011 | 3.480 |
| Basic text | 0.264 | 0.542 | 0.516 | 4.007 |
| Hierarchical text | 0.302 | 0.603 | 0.521 | 3.523 |

Table 13: Results of LoRA and full parameter fine-tuning on MotionBase.

| Training method | R@1 ↑ | R@3 ↑ | FID ↓ | MMDist ↓ |
|---|---|---|---|---|
| Real | 0.290 | 0.563 | 0.011 | 3.480 |
| LoRA | 0.249 | 0.520 | 1.896 | 3.869 |
| Full Param | 0.264 | 0.542 | 0.516 | 4.007 |

model trained on the training set to fit its distribution, and may exhibit bias on the test set. If the generated text-motion data aligns better with the training set distribution, the evaluation metrics might even outperform the ground truth on the test set. Therefore, how to quantitatively evaluate motion generation performance remains an interesting research topic worthy of further exploration.

### C.6 ABLATION STUDY ON LORA AND FULL PARAMETER FINE-TUNING

We conduct an ablation study comparing LoRA and full parameter fine-tuning. As shown in Table 13, LoRA fine-tuning struggles to achieve competitive results. We attribute this limitation to the introduction of new motion tokens, which necessitate substantial parameter adjustments for the language model to comprehend these additional tokens. The constrained nature of LoRA fine-tuning appears insufficient to effectively address these demands.

### C.7 EXPERIMENTAL COMPARISON WITH T2M-GPT ON MOTIONBASE

We train the T2M-GPT model on the MotionBase dataset and compare it with a model based on GPT-2 medium. As shown in Table 14, despite comparable parameter counts, the T2M-GPT method struggles to produce competitive results. Because of the inherent limitations of CLIP's text encoding capabilities, models trained this way struggle to understand a wider range of motion-related language. We believe that large motion models based on decoder-only LLMs, which jointly train text tokens and motion tokens, achieve better text-motion semantic alignment and stronger motion generation capabilities.

### C.8 ABLATION OF MOTION GENERATION BASED ON LFQ

To validate the applicability of the LFQ quantization method for motion generation, we conducted experiments summarized in Table 15. These experiments include data scaling with GPT-2 and parameter scaling using 0.02M training samples. The results are consistent with our initial conclusions, confirming robust performance across scaling scenarios. Furthermore, LFQ demonstrates a slight

Table 14: Results of T2M-GPT and GPT-2 on MotionBase.

| Model | #Param. | R@1 ↑ | R@3 ↑ | FID ↓ | MMDist ↓ |
|---|---|---|---|---|---|
| Real | - | 0.290 | 0.563 | 0.011 | 3.480 |
| T2M-GPT | 380M | 0.243 | 0.504 | 1.909 | 4.593 |
| GPT-2 Medium | 355M | 0.264 | 0.542 | 0.516 | 4.007 |

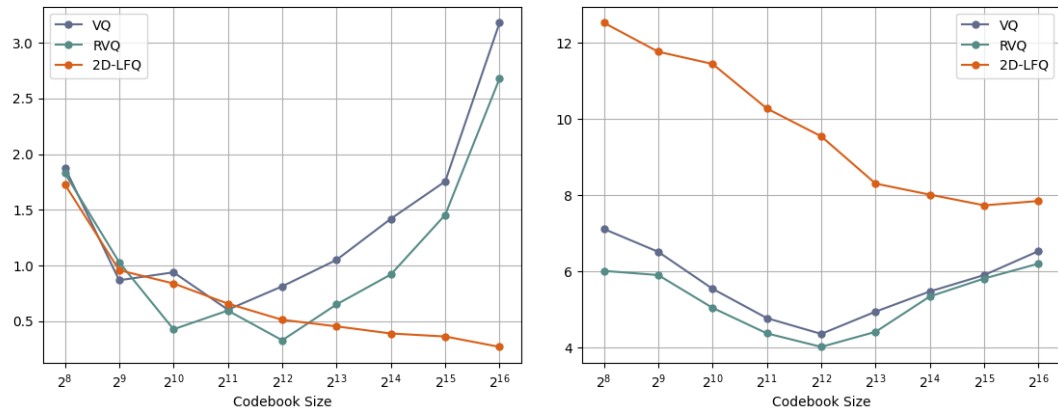

Figure 12: Comparison with different motion quantization on the Motion-X (**left**) and MotionBase dataset (**right**). The Y-axis denotes FID (↓).

performance advantage over VQ when evaluated with GPT-2. Given that LFQ utilizes a significantly larger codebook, which increases training difficulty, we anticipate that further improvements could be achieved by scaling both model parameters and training data.

Table 15: Ablation of motion generation using LFQ and VQ under different setups.

| Decoder | #Inst. | #Param. | Motion-X | | | MotionBase | | |
|---------|--------|---------|---------|---------|--------|-----------|---------|--------|
| | | | R@1 ↑ | R@3 ↑ | FID ↓ | R@1 ↑ | R@3 ↑ | FID ↓ |
| GPT-2-VQ | 1M | 355M | 0.357 | 0.614 | 5.083 | 0.264 | 0.542 | 0.516 |
| GPT-2-LFQ | 0.02M | 355M | 0.166 | 0.341 | 76.214 | 0.042 | 0.085 | 136.254 |
| GPT-2-LFQ | 0.08M | 355M | 0.332 | 0.558 | 6.245 | 0.062 | 0.144 | 128.071 |
| GPT-2-LFQ | 1M | 355M | 0.394 | 0.628 | 4.275 | 0.326 | 0.607 | 0.452 |
| GPT-2-LFQ | 0.02M | 355M | 0.166 | 0.341 | 76.214 | 0.042 | 0.085 | 136.254 |
| LLaMA-2-LFQ | 0.02M | 7B | 0.225 | 0.383 | 68.542 | 0.062 | 0.140 | 125.082 |
| LLaMA-2-LFQ | 0.02M | 13B | 0.206 | 0.351 | 71.238 | 0.085 | 0.184 | 119.036 |

## C.9    ABLATION OF MOTION QUANTIZATION

First, we provide additional FID results on Motion-X in Figure 12. It is worth noting that while our motion quantizer performs worse than RQ-VAE on the smaller HumanML3D dataset, it surpasses both VQ and RQ when evaluated on the larger Motion-X and MotionBase benchmarks, as can be seen in Table 6. This suggests that our approach offers a greater advantage when applied to larger datasets, highlighting its improved generalization compared to previous methods.

To further validate the effectiveness of our 2D quantization strategy, we compare the 2D-LFQ method with its 1D counterpart (which is identical to VQ except for the quantization strategy). The results, shown in Table 16, demonstrate that 2D quantization in LFQ significantly outperforms the 1D version. This highlights the superior ability of 2D quantization to enhance the representational capacity of the motion tokenizer.

Table 16: Ablation of 2D motion quantization vs. its 1D version.

| Tokenizer | #Num. | #Param. | HumanML3D | | Motion-X | | MotionBase | |
|-----------|-------|---------|-----------|---------|----------|---------|-----------|---------|
| | | | FID ↓ | MPJPE ↓ | FID | MPJPE | FID | MPJPE |
| 1D-LFQ | 16384 | 19.43M | 3.85 | 52.5 | 2.783 | 78.9 | 10.358 | 80.1 |
| 2D-LFQ | 16384 | 108.35M | **1.769** | **45.6** | **0.295** | **54.1** | **7.853** | **64.1** |

## D    DATASET CONSTRUCTION PIPELINE

Our data collection pipeline is a multi-stage process designed to curate a large-scale, high-quality, and richly annotated multimodal motion dataset. The detailed steps are outlined below:

**Video Data Collection and Cleaning:** We amass over 20 million videos from publicly available datasets like InternVid and WebVid, as well as online platforms such as YouTube. To maintain data relevance and quality, we employ a pretrained human detection model to filter out videos lacking human presence.

**2D and 3D Keypoint Estimation:** We estimate 2D human keypoints and their corresponding confidence scores using the pretrained VitPose model (Xu et al., 2022). To further refine motion information, we leverage a pretrained 3D keypoint estimation model (Sárándi et al., 2023) trained on extensive 3D datasets. Following the methodology of Lin et al. (2024), we apply temporal smoothing and 3D bone length constraints during triangulation to enhance the stability and consistency of the keypoint estimations.

**Multimodal Information Integration:** For a more comprehensive understanding of human motion, MotionBase incorporates RGB, depth data, and annotations for multi-person scenarios. In multi-person sequences, each motion is grounded to its respective video via object-level bounding boxes. While this work primarily focuses on text-to-motion tasks, these additional modalities pave the way for future research in related areas.

**Local-Global Pose Estimation:** We fit the SMPL-X body model (Pavlakos et al., 2019) to each frame in MotionBase using a progressive learning-based mesh fitting approach (Lin et al., 2024). Specifically, we predict SMPL-X parameters using the pretrained OSX method (Lin et al., 2023), followed by iterative optimization to align the parameters with the target 2D and 3D joint positions. Subsequently, we apply a global motion optimization technique based on Yuan et al. (2022) to refine both global motions and camera poses, ensuring consistency with the video evidence. Finally, for motion sequences with noisy or occluded input data, we employ RoHM (Zhang et al., 2024a) to reconstruct complete and plausible motions.

**Single-Frame Pose Expansion:** To enhance dataset diversity and scale, we expand single-frame pose data into multi-frame sequences. We achieve this using the PHC (Luo et al., 2023) strategy and the pre-trained motion completion model MotionGPT (Jiang et al., 2023). The PHC strategy ensures the physical plausibility of the generated motion sequences, while MotionGPT provides motion priors to enhance naturalness and fluidity.

**Hierarchical Motion Descriptions:** MotionBase features hierarchical text annotations to address limitations in existing dataset descriptions. Leveraging the Gemini-1.5-pro large language model (Reid et al., 2024) and a carefully crafted prompt format, we generate detailed descriptions for individual body parts (e.g., left arm, right leg), dedicating a sentence to each. Furthermore, we summarize the overall body movement with 1-3 sentences, providing a more holistic motion description.

## E    DATASET QUALITY EVALUATION

### E.1    MOTION DATA QUALITY

To ensure dataset quality, we conduct multifaceted evaluations of the motion data.

**Refinement using a Reinforcement Learning-based Strategy:** We use PHC to train a reinforcement learning-based policy model that refines the raw motion data, ensuring conformity to physical laws and enhancing realism. This policy takes raw motion sequences as input, treats them as target poses, and generates new motion sequences satisfying physical laws in a simulated environment, thereby eliminating issues such as jitter and foot sliding. While this strategy may encounter challenges with drastic movements, it effectively improves data quality for most motion sequences.

**Data Diversity:** A key advantage of the MotionBase dataset is its scale and diversity. We collect over one million motion sequences from multiple sources (including InternVid and internet videos), encompassing a wide range of motion types. This diversity supports the training of more generalizable motion models.

| Text Prompt | Generated Motion Sequences |
|---|---|

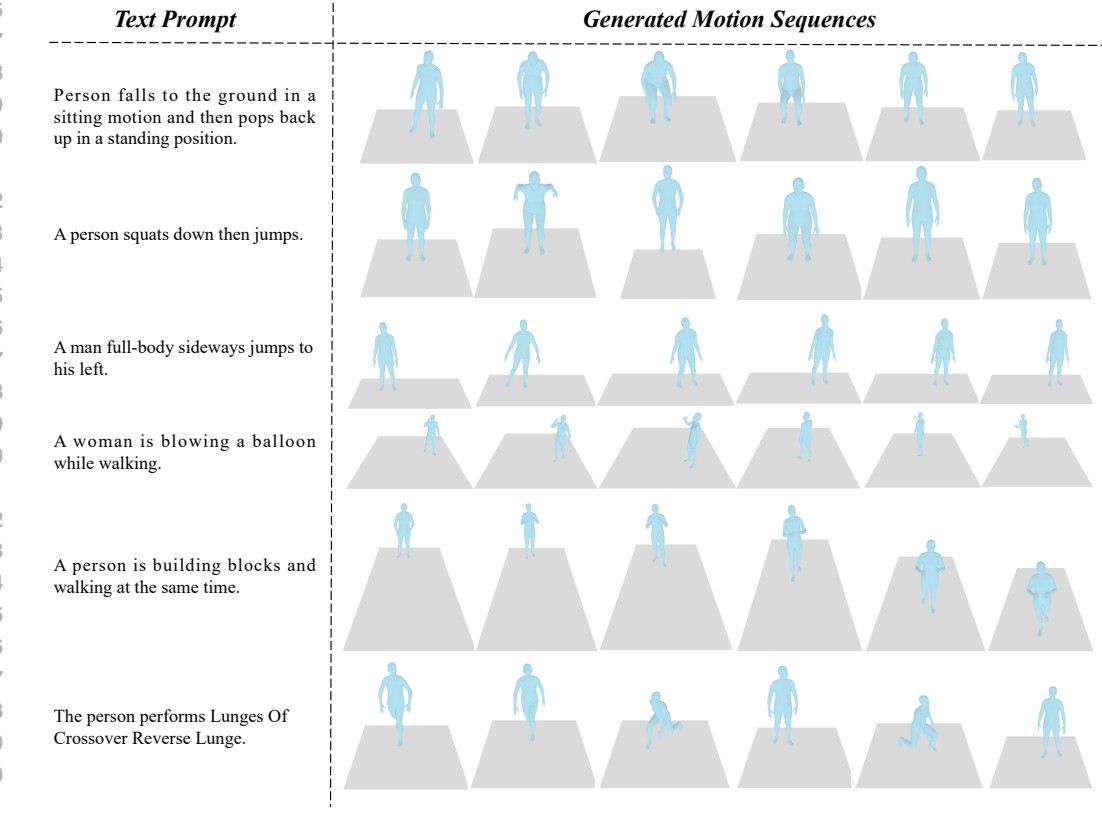

Person falls to the ground in a sitting motion and then pops back up in a standing position.

A person squats down then jumps.

A man full-body sideways jumps to his left.

A woman is blowing a balloon while walking.

A person is building blocks and walking at the same time.

The person performs Lunges Of Crossover Reverse Lunge.

Figure 13: Quantitative examples of motions generated by our large motion model.

### E.2 TEXT DESCRIPTION QUALITY

To ensure text description quality, we employ a multi-level evaluation approach.

**Automatic Evaluation based on Large Language Models:** We automatically evaluate text descriptions in MotionBase using large language models such as Gemini-1.5-pro (Reid et al., 2024) and GPT-4. We use a 1-to-5 scoring system based on these criteria:

- 1 point (Very Poor): The description is vague, irrelevant to the motion content, or contains severe grammatical errors.
- 2 points (Poor): The description lacks specifics, detail, or contains obvious errors.
- 3 points (Fair): The description basically reflects the motion content but lacks detail and may contain minor errors.
- 4 points (Good): The description is accurate and detailed, clearly expressing the motion process.
- 5 points (Excellent): The description is precise, comprehensive, and fluent, providing an in-depth analysis of motion details.

We use GPT-4o to score text descriptions from MotionBase, MotionX, and HumanML3D. MotionBase achieves an average score of 3.837, while MotionX and HumanML3D score 1.386 and 1.703, respectively, indicating higher quality in MotionBase's text descriptions. To further evaluate consistency between text descriptions and motion content, we also input text descriptions and corresponding rendered motion videos into the Gemini-Pro model for scoring. MotionBase achieves an average score of 3.82, while MotionX and HumanML3D score 2.25 and 3.08, respectively, again confirming the quality advantage of MotionBase's text descriptions.

**Consistency Check of Hierarchical Descriptions:** MotionBase provides hierarchical text descriptions, including overall, local detail, and rule-based descriptions. We use GPT-4 and manual checks

to ensure consistency across different levels, guaranteeing logical coherence and informational completeness.

# F ADDITIONAL QUALITATIVE RESULTS

We provide some examples to visualize the human motions predicted by our large motion model trained on MotionBase, as illustrated in Figure 13. As can be seen, our large motion model is capable of generating motion sequences that align well with the input texts, demonstrating the effectiveness of the MotionBase dataset.

