# OpenReview forum: "Quo Vadis, Motion Generation? From Large Language Models to Large Motion Models"
_ICLR.cc/2025/Conference — Submitted to ICLR 2025_

### Official Review · Reviewer_WfZ5 · 2024-10-16

**Soundness:** 1
**Presentation:** 2
**Contribution:** 1
**Rating:** 5
**Confidence:** 5

**Summary:**

This paper claims to propose a large motion model with a very large motion database. However, the motion quality is not well evaluated. Besides, the authors propose a motion quantization method, which is borrowed from LFQ (Mentzer et al., 2023). The authors claim a good generation quality of the generated results, which is not provided in the demo.

**Strengths:**

- The writing of this work is a bit fancy.

- The statistics of the dataset are clear.

**Weaknesses:**

There are several fundamental concerns about this work. Each of these is fatal.

1. **The motion collection process.** This process contains several issues.
    - This work does not evaluate the quality of the video mocap data quality. To my knowledge, even the quality of the latest Motion-X++ suffers significant jittering and foot sliding. How can your method escape from this? **(my main concern)**
    - If the quality of the ground truth is not good enough, how can you generate good motion? Therefore, the result in L395-402 is not solid and convincing. **(my main concern)** I suggest authors read the blog [1] written by a well-known graphics scientist, Daniel Holden.
    - The limited contribution of the dataset. The video data comes from InternViD and WebVid and the data collection process is from motion-x and other methods. The dataset contribution is limited.

2. **The text annotation.**
    - The annotation quality of the text by Gemini-1.5-pro is not well evaluated. In my practice, it always contains some answers like "sorry...". The results should be corrected by researchers one by one. Has the >1M data been checked?
    - The proposed contribution of hierarchical text is not discussed well. Has it been used in the model training? If I miss, please point it out. If this annotation is not used, what is the motivation for this hierarchical text contribution? Will it make the result more fine-grained? It is quite unclear. **(my main concern)**

3. **Limited technical/evaluation contribution.** The LFQ is proposed by the original paper. The authors did not have a new understanding over this. Besides, the H2VQ proposed in Humantomato (ICML-24) is also missing for discussion or comparison.

4. This work does not include any demo video, which is unacceptable in the animation community. The FID in Table 5 is extremely large, which strengthens my concerns about the motion quality.

5. **Motivation.** The motivation for introducing LLM is not clear. The method misses a basic baseline of a transformer (like in T2M-GPT, CVPR-23) for comparison. Besides, it is also not clear whether the usage of pre-trained parameters of LLMs or not. Whether the fine-tuning method is LoRA or not is also not well discussed. Therefore, it is not technically sound. **This is my strong concern.**

[1]: Daniel Holden, https://theorangeduck.com/page/animation-quality.

**Questions:**

- The vocabulary of the LLM and motion codebooks are different. How do authors handle this issue? What is the efficiency of the LLM-based motion generation method? Please compare with the fastest motion generation method, MotionLCM (ECCV-24).

- **I would like to know why authors should cite [1].**

[1]: Zheng et al., Steve-eye: Equipping llm-based embodied agents with visual perception in open worlds.

**Details Of Ethics Concerns:**

This paper contains a dataset of human subjects. The human RGB videos are also included. Besides, some of the data comes from other datasets, which should be clear on the usage of commercial or not.

---

> ### Author Response · Authors · 2024-11-22
> **Official Comment to Reviewer WfZ5's weakness 1**
>
> We greatly appreciate the time and effort you invested in providing these detailed observations, questions, and comments. We have carefully considered your comments and have outlined our responses and proposed changes below. We hope these adjustments and explanations address your concerns and further enhance the manuscript.
>
> Before reading our feedback, we introduce a visualization website to validate the quality of our dataset at http://www.motionbase3d.com. (**Please note: this link may be unstable, and refreshing the page 10~20 times may be required to view the content**). In case the reviewer is unable to access the site, we have also provided an anonymous cloud storage link containing all the examples featured on the website: [click this link](https://www.dropbox.com/scl/fo/6w7yuhun8mpuz9yuaz5ej/ALUo_dOLYIyNgOZ3Ll85JCI?rlkey=yecydfnno43b5o602ae2fmr0w&st=rq97kdcw&dl=0). The platform includes visualized samples from our dataset, physically grounded examples of refined motions, rendered results generated by different motion models, and live demonstrations in both simulated environments and real-world settings (using H1 and G1 UNITREE humanoid robots). We hope this platform addresses the data quality concern reviewers may have. If the reviewer requires additional information or examples, we are more than happy to upload further relevant results upon request. Additionally, our dataset is highly scalable due to the robust data collection pipeline. As of the start of the rebuttal period, we have collected over 1.5 million motion trajectories and 4 million motion-text pairs, which are 1.5X compared to our first submission. These datasets have undergone three rounds of rigorous data filtering to ensure their quality.
>
>
> **W1. [The motion collection process] How do you evaluate and make sure the quality of collected motion data, avoid issues like jittering and foot sliding**
>
> We have implemented several strategies to evaluate and refine the motions generated in MotionBase.
> First, thanks to our constructed visualized platform, we can efficiently sample and assess a large amount of motions from MotionBase.
> Then, to ensure motion quality, we adopt the following steps:
> 1. We first train an RL-based policy $ \pi_{\text{refine}} $ to elaborate and refine raw motions, ensuring they adhere to physical laws (e.g., maintaining balance) and appear more realistic. Specifically, this policy takes raw motion sequences as input, treating them as target poses, and generates new motion sequences that satisfy physical laws in a simulated environment, eliminating issues like jittering and foot-sliding.
> 2. While effective, the RL-based policy $ \pi_{\text{refine}} $ may struggle with drastic movements in target poses, leading to slipping within the simulation. For such cases, we adopt the following considerations: If a refined motion maintains balance in the simulated environment for a specific duration, it can be regarded as high-quality with no severe issues like jittering or sliding. If a motion fails to maintain balance from the start, it is considered a bad sample and discarded.
> 3. For motions that pass **STEP 2**, we utilize a pretrained motion model like RoHM[1] for further refinement. Additionally, we experiment with more powerful motion reconstruction models such as WHAM[2]. Based on our experience, existing motion models have been able to effectively enhance the quality of raw motions.
>
> [1] RoHM: Robust Human Motion Reconstruction via Diffusion
>
> [2] WHAM: Reconstructing World-grounded Humans with Accurate 3D Motion
>
> **W1. [The motion collection process] If the quality of the ground truth is not good enough, how can you generate good motion? Therefore, the result in L395-402 is not solid and convincing.**
>
> We understand your concerns regarding the impact of motion quality and the results discussed in lines L395–402. First, we believe the poor FID results in L395–402 primarily stem from the limitations of the retrieval model used in the ICLR version.  The model [1], initially trained on HumanML3D/MotionX, was applied to evaluate MotionBase, which introduced a mismatch. To address this, we have adopted a more robust retrieval model specifically trained on MotionBase. With this improved model, we observed significant performance gains, particularly in the FID metric.
> Second, as previously detailed, we have developed a functional toolbox to refine both motion and text data through various strategies. MotionBase has undergone three rounds of rigorous data cleaning and verification, and we plan to implement additional filtering processes to further enhance its quality. For better visualization, we also showcase our data on a website and hope these enhancements address your concerns and bolster confidence in our results.
>
> [1] Generating Diverse and Natural 3D Human Motions From Text

---

> ### Author Response · Authors · 2024-11-22
> **Official Comment to Reviewer WfZ5's weakness 1**
>
> **W1. [The motion collection process] The limited contribution of the dataset. The video data comes from InternViD and WebVid and the data collection process is from motion-x and other methods. The dataset contribution is limited.**
>
> It is important to note that we do not directly use videos on the Internet, but have undertaken extensive efforts and developed a strategic combination to collect useful motion data from the Internet. Without these efforts, raw video data would remain a noisy, unusable collection. In addition to the strategies we introduced to improve the quality of motion and text descriptions, we provide more details of the data construction process.
>
> 1. **Video Collection and Selection:** It is worth noting that MotionX data constitutes only 5% of the entire dataset. Meanwhile, rather than relying solely on open-source datasets like InternVid, we also extract motion data from self-collected videos sourced from the Internet. For these videos, we use a pretrained 2D human keypoint detection model to filter out those without visible human activity. Additionally, rule-based methods are applied to ensure that the human bounding box occupies a significant portion of the frame, making human movement clearly visible. Videos with only partially visible humans are removed to maintain the quality of the potential motion data. Through these methods, we ensure the extracted motion data is of high quality.
>
> 2. **Short Boundary Detection:** Web videos are generally lengthy and feature varied camera shots. To address this challenge, we adopt the following steps:
>
>     (1) First, for videos shorter than 30 seconds or those with explicit temporal boundaries, we directly use the video clip or the provided boundaries to segment the video into shorter clips.
>
>     (2) For videos longer than 30 seconds, we employ a scene detection model to roughly divide the video into smaller segments.
>
>     (3) For each segment, we further slice it into shorter clips using the following process:
>     - At the beginning, the human with the largest bounding box is selected as the anchor, and their trajectory is tracked throughout the segment.
>     - When the trajectory is interrupted, the start and end times of the interruption are marked as the boundaries of a new clip.
>     - The process repeats by identifying the next largest visible human in subsequent frames and tracking their trajectory.
>     - This process continues until no humans are visible in the video.
>     - Clips without visible humans are filtered out.
>
>     (4) After these steps, if a clip is still longer than 60 seconds, we randomly slice it into several sub-clips, ensuring that each sub-clip is shorter than one minute.
>
> 3. **Removing Occlusion and blur:** Occlusion and motion blur are common issues in human-related videos. To address these problems, we adopt the following steps:
>
>     (1) First, we sample key frames from each video and use a pretrained 2D keypoint detector to extract skeleton keypoints for each human in the key frames. If a significant portion of the keypoints has predicted confidence scores below a specific threshold, we consider the human motion to be occluded and exclude it from further processing.
>
>     (2) We then use a visual foundation model, such as Segment Anything, to generate segmentation masks for each frame. If a large object is detected in front of the human, indicating occlusion, we filter out the corresponding motion data.
>
>     (3) To address motion blur, we track the trajectory of each human whose motion data needs to be extracted. For timestamps with low-confidence keypoint scores, we smooth the trajectory using adjacent detection results to ensure continuity and accuracy.
>
> 4. **Single-frame Motion Processing:** A substantial portion of our data consists of single-frame motions, many of which can be transformed into multi-frame sequences to enhance data diversity. To achieve this, we train a RL-based policy $\pi_{\rm multi\\_frame}$  using the AMASS dataset. This policy generates physically plausible motion sequences within a simulation environment, using the single-frame motion as the target pose. However, due to potential instability caused by drastic lower-body movements, some generated motions may fail to maintain balance. For single-frame motions that cannot be successfully converted, we use another pretrained, target-conditioned motion generator based on existing high-quality motion data. This generator uses the single-frame motion as the target pose and generates the preceding motion, effectively producing a complete sequence. While these generated motions are not fully constrained by physical laws, resulting in less consistent quality compared to those generated by the RL-based policy, they still provide an effective solution for motion conversion.

---

> ### Author Response · Authors · 2024-11-22
> **Official Comment to Reviewer WfZ5's weakness 2**
>
> **W2. [The text annotation] The annotation quality of the text by Gemini-1.5-pro is not well evaluated. In my practice, it always contains some answers like "sorry...". The results should be corrected by researchers one by one. Has the >1M data been checked?**
>
> The quality of generated texts is highly related to the prompt design. We introduce two approaches used in our work to evaluate the text description quality.
>
> 1. Firstly, we sample 10,000 motion descriptions from our MotionBase generated by Gemini, along with 10,000 descriptions each from MotionX and HumanML3D. These descriptions are scored using GPT-4o, which evaluates each description on a scale of 1 to 5 based on predefined criteria focused on clarity, detail, and accuracy. The scoring criteria are as follows.
>
>     - Score 1: The description is vague, lacks specific details about body movement, and contains confusing or unclear expressions.
>     - Score 2: The description covers some movement and posture content but lacks sufficient detail or accuracy and includes unclear expressions.
>     - Score 3: The description clearly outlines movement and posture, providing basic details but lacking in-depth analysis.
>     - Score 4: The description is accurate and detailed, effectively conveying the movement process and changes in body posture, with some analytical depth.
>     - Score 5: The description is precise, comprehensive, and fluent, offering in-depth analysis of every detail of the movement and posture, demonstrating a high level of professionalism and clarity.
>
>   We then calculate the average scores for each dataset: MotionBase (3.837) and MotionX (1.386) and HumanML3D (1.703). These scores suggest that MotionBase descriptions are generally more detailed and accurate compared to MotionX and HumanML3D.
>
> 2. To further evaluate the quality of the generated texts for vision-based motions, we prompt Gemini-pro with text descriptions and corresponding rendered motions. Our primary focus is on the accuracy with which the text descriptions reflect the content of the visual cues. To assess this, we present 500 rendered samples with their corresponding text descriptions from each dataset to Gemini, requesting a score based on the criteria we established earlier. The evaluation results provide valuable insights. The texts of MotionX and HumanML3D receives an average score of 2.25 and 3.08, respectively. Notably, MotionBase achieves a significantly higher average score of 3.82, outperforming the other two datasets.
>
> **W2. [The text annotation] The proposed contribution of hierarchical text is not discussed well. Has it been used in the model training? If I miss, please point it out. If this annotation is not used, what is the motivation for this hierarchical text contribution? Will it make the result more fine-grained? It is quite unclear.**
>
> | Training text | R@1 ↑ | R@3 ↑ | FID ↓ | MMDist ↓ |
> |-----------------|-------|-------|-------|-----------|
> | Real            | 0.290 | 0.563 | 0.011 | 3.480     |
> | Basic           | 0.264 | 0.542 | 0.516 | 4.007     |
> | Hierarchical    | 0.302 | 0.603 | 0.521 | 3.523     |
>
> YES, the hierarchical text is used during pretraining to improve the diversity of text corpus and make the description finer-grained. The table compares experimental results using hierarchical text versus using only basic text. The results show that hierarchical text can effectively enhance the model's semantic understanding and thereby improve the semantic matching of generated motions.
>
> In addition, the hierarchical texts also ensure the quality of motion description generated by Gemini-1.5 or GPT-4o. Specifically, our latest motion descriptions are structured into three hierarchical and complementary levels: an overall description of the entire body, detailed part-level description for each body part, rule-based description of each joint's relative movement derived from joint positions (e.g., "The left hand is wide apart from the right hand. It is on the ground, the left upper arm, both legs and the left forearm are aligned horizontally......"). To enhance the reliability and quality of the motion descriptions, we condition GPT-4o with two levels of description while using the remaining level as the evaluation target.  GPT-4o then refines the textual content. By doing this, each level of description can provide complementary details and correction for the other two levels, enabling the generation of more precise and reliable motion descriptions.

---

> ### Author Response · Authors · 2024-11-22
> **Official Comment to Reviewer WfZ5's weakness 3-4**
>
> **W3. The H2VQ proposed in Humantomato (ICML-24) is missing for discussion or comparison. And the technical and evaluation contribution of motion VQ is limited.**
>
> To the best of our knowledge, Humantomato has not released their code, and no successful re-implementation has been found. As a result, it is currently impossible to conduct a quantitative comparison with the H2VQ method proposed in Humantomato. Additionally, we re-implemented RVQ and achieved better reconstruction performance (56.9 MPJPE) compared to H2VQ (62.34 MPJPE reported) on MotionX. Given this, we have chosen not to include a comparison with H2VQ in Table 6. We would be glad to perform such a comparison once Humantomato releases their code.
> However, to improve the reference discussion, we also include this reference to our related work.
>
> The primary contribution of our motion tokenization technique lies in the exploration of tokenization methods specifically designed for motion generation. While previous works have used 1D discrete vectors to represent the body, we highlight the information loss that this approach may cause. In this paper, we propose a novel direction for encoding motions by treating the motion sequence as a 2D image. It's important to note that this approach has not been extensively discussed in prior work, and we validate its potential and effectiveness through our experiments. Therefore, the inclusion of LFQ in our study serves as a preliminary exploration to assess the potential of alternative tokenization methods for motion representation. We intend to conduct a more thorough investigation of this approach in future work.
>
> **W4. This work does not include any demo video, which is unacceptable in the animation community.**
>
> Thank you for your suggestion! To address your concern about the visualization of demo videos, we have created a website showcasing a variety of examples, including samples from our datasets and generated results from our models. Additionally, we present physically grounded examples, as well as examples deployed in both simulated and real-world environments ---- an aspect that has not been highlighted in previous works!
>
> **W4. The FID in Table 5 is extremely large, which strengthens my concerns about the motion quality.**
>
> We figure out that the high FID values reported in Table 5 are primarily due to the limited encoding capability of the retrieval model used for evaluation.
>
> The model originally used in our ICLR submission was trained on HumanML3D and was unable to fully capture the characteristics of the motion data, leading to abnormally high FID scores. To address this, we train a more powerful and robust encoding model on our MotionBase dataset. After re-evaluating with this improved model, we observe a significant reduction in FID values, bringing them into a reasonable and acceptable range.  We include these updated evaluation results in the revised version to ensure clarity. We also plan to use more recently proposed text-motion retrieval models like TMR[1] to obtain a more accurate evaluation of model performance.
>
> [1] TMR: Text-to-Motion Retrieval Using Contrastive 3D Human Motion Synthesis.

---

> ### Author Response · Authors · 2024-11-22
> **Official Comment to Reviewer WfZ5's weakness 5**
>
> **W5. Motivation. The motivation for introducing LLM is not clear. The method misses a basic baseline of a transformer (like in T2M-GPT, CVPR-23) for comparison. Besides, it is also not clear whether the usage of pre-trained parameters of LLMs or not. Whether the fine-tuning method is LoRA or not is also not well discussed. Therefore, it is not technically sound.**
>
> We incorporate LLM to enhance language understanding, a critical aspect of text-to-motion generation, which relies on strong comprehension of textual nuances. LLMs, with their proven capabilities in natural language processing, enable our model to better capture the context and subtleties of text descriptions. As a result, this leads to the generation of more accurate and contextually appropriate motion sequences. The table compares our method to baseline T2M-GPT (CVPR 2023) on HumanML3D. Our results show significant improvements. These results highlight the advantages of leveraging LLMs. We use the pre-trained LLM parameters, which offer greater capacity and superior language understanding so that our model is able to learn more complex mappings between language and motion. In Table 9 of Appendix C.3, we compare experiments with and without pre-trained parameters, showing that fine-tuned models using pre-trained parameters consistently outperform models trained from scratch. During training, we utilize full fine-tuning. While we also experimented with LoRA, it struggled to achieve competitive results. We attribute this limitation to the introduction of new motion tokens, which demand substantial parameter adjustments. LoRA, with its constrained fine-tuning approach, appears less equipped to handle these requirements effectively.
>
> | Training method | R@1 ↑ | R@3 ↑ | FID ↓ | MMDist ↓ |
> |-----------------|------:|------:|------:|----------:|
> | Real            | 0.290 | 0.563 | 0.011 | 3.480     |
> | LoRA            | 0.249 | 0.520 | 1.896 | 3.869     |
> | Full Param      | 0.264 | 0.542 | 0.516 | 4.007     |

---

> ### Author Response · Authors · 2024-11-22
> **Official Comment to Reviewer WfZ5's question**
>
> **Q1. The vocabulary of the LLM and motion codebooks are different. How do authors handle this issue? What is the efficiency of the LLM-based motion generation method? Please compare with the fastest motion generation method, MotionLCM**
>
> To address varying vocabularies, we extend the LLM's vocabulary by incorporating motion codebook tokens as additional entries. The LLM is then trained directly on text-motion paired data, enabling it to effectively learn the associations between textual descriptions and motion tokens.
>
> We test our 7M LLM-based model on RTX-4090 GPU for inference, the token generation speed is 23 token/sec. Since each token represents 4-frame motion, the generation speed of our model is around 90 FPS, denoting a real-time inference spped which can be deployed in reality, although still can not beat the speed of fastest methods like MotionLCM. However, our focus is exploring the capabilities of large models for text-to-motion generation. We are not prioritizing speed optimization in this work. Speed improvements, such as using Flash Attention, are orthogonal to our current research goals and represent an important area for future work.
>
> **Q2. Why authors should cite [1]?**
>
> The references cited here primarily aim to demonstrate that the development of multimodal large language models depends heavily on the availability of extensive data across various domains, including digital environments like games, egocentric scenarios, and others.
>
> We hope our clarifications address your questions, and we kindly ask if there are additional concerns we can address to further improve your support for the paper. Thank you again for your valuable suggestions.

---

### Official Review · Reviewer_qpUP · 2024-11-01

**Soundness:** 2
**Presentation:** 1
**Contribution:** 2
**Rating:** 5
**Confidence:** 5

**Summary:**

This paper collects a new large-scale text-motion dataset called MotionBase and then finetunes LLMs with different sizes. Additionally, for better scaling, the authors follow the video domain to train a new LFQ tokenizer with a large vocab size.

**Strengths:**

1. The authors try to scale the tokenizer vocab size and the model size.
2. The authors collect a new large-scale text-motion dataset.

**Weaknesses:**

1. The biggest server weakness is containing the one-frame pose data into the database. The Agora, mscoco, muco_3dhp, and other more datasets are used for 3d pose estimation, and they even occupy a large portion of the whole database, which may lead to static motion generation.
2. The motion quality has not been validated. Neither the estimated motions nor the texts generated by LLM have been checked manually or by any algorithm. The video collection process is not clarified clearly. A lot of web videos are long and contain various camera shots. Which film shot boundary detection algorithm are you using? And how many frames do you insert into LLM to get the text? More details need to be added.
3. The experiments with static data ablation study are not fair. Does the validation set contain static data and synthetic data?

**Questions:**

1. The FID in Table 6 is so wired. The FID of reconstruction is 1.76 while the generation FID in Table 3 is 0.166. This is impossible from my understanding. I suspect that the reconstruction result is not good enough. The original MPJPE calculation will subtract the root movement. If you calculate MPJPE similarly, the high reconstruction FID means the translations are not accurate.
2. What do the authors get from scaling experiments? Did the author see any hope for emerging? The shown examples are common cases, that can be also observed in other motion generation work.
3. Did the supervised label contain only motion tokens or both text and motion tokens?
4. Did the author try zero-shot text testing? For example, could the largest model do some texts like "The old man with a broken leg is walking forward slowly with a crane"?

---

> ### Author Response · Authors · 2024-11-22
> **Official Comment to Reviewer qpUP's weakness 1**
>
> We greatly appreciate the time and effort you invested in providing these detailed observations, questions, and comments. We have carefully considered your comments and have outlined our responses and proposed changes below. We hope these adjustments and explanations address your concerns and further enhance the manuscript.
>
> Before reading our feedback, we introduce a visualization website to validate the quality of our dataset at http://www.motionbase3d.com. (**Please note: this link may be unstable, and refreshing the page 10~20 times may be required to view the content**). In case the reviewer is unable to access the site, we have also provided an anonymous cloud storage link containing all the examples featured on the website: [click this link](https://www.dropbox.com/scl/fo/6w7yuhun8mpuz9yuaz5ej/ALUo_dOLYIyNgOZ3Ll85JCI?rlkey=yecydfnno43b5o602ae2fmr0w&st=rq97kdcw&dl=0).
> The platform includes visualized samples from our dataset, physically grounded examples of refined motions, rendered results generated by different motion models, and live demonstrations in both simulated environments and real-world settings (using H1 and G1 UNITREE humanoid robots). We hope this visualization platform addresses the data quality concern reviewers may have. If the reviewer requires additional information or examples, we are more than happy to upload further relevant results upon request. Additionally, our dataset is highly scalable due to the robust data collection pipeline. As of the start of the rebuttal period, we have collected over 1.5 million motion trajectories and 4 million motion-text pairs, which are 1.5 times compared to our first submission. These datasets have undergone three rounds of rigorous data filtering to ensure their quality.
>
> Here are our responses to the weaknesses.
>
> **W1. Does the large proportion of single-frame motion data in the dataset contribute to static motion generation?**
>
> No.
> 1. First, our dataset is designed to be highly scalable, with the latest version now including a reduced share (44%) of one-frame motion data and over 1.5 million motions ---- a significant expansion over the ICLR version. Notably, this update incorporates **more than 50% additional multi-frame motions** extracted from open-source human behavior datasets (e.g., NTU120, Kinetics-700) and publicly available web videos. We plan to further increase the proportion of multi-frame motions in future versions.
> 2. Second, a substantial portion of one-frame motions can be transformed into multi-frame sequences. To achieve this, we train a RL-based policy $\pi_{\rm multi-frame}$ using the AMASS dataset. This policy generates physically plausible motion sequences within a simulation environment by using the single-frame motion as the target pose. Due to potential instability caused by drastic lower-body movements, some generated motions may fail to maintain balance.
> 3. For single-frame motions that fail conversion in **STEP 2**, we employ a pretrained, target-conditioned motion generator based on existing high-quality motion data. This generator uses the single-frame motion as the target pose and generates its preceding motion, effectively producing the entire motion sequences. Compared to motions converted through **STEP 2**, these generated motions may not fully adhere to physical laws.
> 4. To further avoid static motion generation, we provide distinct prompts for single-frame and multi-frame motions during the LLM's fine-tuning, ensuring that the model is capable of generating dynamic motion in response to specific commands.

---

> > ### Author Response · Authors · 2024-11-22
> > **Official Comment to Reviewer qpUP's weakness 2**
> >
> > **W2. The motion quality has not been validated. Neither the estimated motions nor the texts generated by LLM have been checked manually or by any algorithm.**
> >
> > In fact, we adopt several strategies to evaluate (**and keep improving**) both the motions and the texts of MotionBase.
> >
> > For the motions, our visualization platform allows us to efficiently sample and manually check a large number of samples.
> >
> > In addition, we employ the following steps to ensure the quality of collected motions:
> > 1.  RL-based Refinement. We first train a RL-based policy $\pi_{\rm refine}$ to elaborate and refine raw motions, ensuring they adhere to physical laws (e.g., maintaining balance) and appear more realistic. Specifically, this policy takes raw motion sequences as input, treating them as target poses, and generates new motion sequences that satisfy physical laws in a simulated environment, eliminating issues like jittering and foot-sliding.
> > 2. Empirical Assessment. While effective, the RL-based policy $\pi_{\rm refine}$ may struggle with drastic movements in target poses, leading to slipping within the simulation. For such cases, we adopt the following empirical considerations: If a refined motion maintains balance in the simulated environment for a specific duration, it can be generally regarded as high-quality with no significant jittering or sliding. If a motion fails to maintain balance from the start, it is considered a bad sample and discarded.
> >  3. Motion Model Refinement. For motions that pass in the last step, we still utilize a well-pretrained motion model like RoHM[1] for further refinement. Additionally, we experiment with more powerful motion reconstruction models such as WHAM[2]. Based on our experience, existing motion models now have been able to effectively enhance the quality of raw motions.
> >
> > For the texts, we apply two approaches to evaluate the quality of textual motion descriptions:
> > 1. we sample 10,000 motion descriptions from our MotionBase generated by Gemini, along with 10,000 descriptions each from MotionX and HumanML3D. These descriptions are scored using GPT-4o, which evaluates each description on a scale of 1 to 5 based on predefined criteria focused on clarity, detail, and accuracy. The scoring criteria are as follows:
> >     - Score 1: The description is vague, lacks specific details about body movement, and contains confusing or unclear expressions.
> >     - Score 2: The description covers some movement and posture content but lacks sufficient detail or accuracy and includes unclear expressions.
> >     - Score 3: The description clearly outlines movement and posture, providing basic details but lacking in-depth analysis.
> >     - Score 4: The description is accurate and detailed, effectively conveying the movement process and changes in body posture, with some analytical depth.
> >     - Score 5: The description is precise, comprehensive, and fluent, offering in-depth analysis of every detail of the movement and posture, demonstrating a high level of professionalism and clarity.
> > We then calculate the average scores for each dataset: MotionBase (3.837) and MotionX (1.386) and HumanML3D (1.703). These scores suggest that the text descriptions of MotionBase are generally more detailed and accurate compared to MotionX and HumanML3D.
> > 2. To further evaluate the quality of the generated texts for vision-based motions, we prompt Gemini-pro with text descriptions and corresponding rendered motions. Our primary focus is on the accuracy with which the text descriptions reflect the content of the visual cues. To assess this, we present 500 rendered samples with their corresponding text descriptions from each dataset to Gemini, requesting a score based on the criteria we established earlier. The evaluation results provide valuable insights. The texts of MotionX and HumanML3D receives an average score of 2.25 and 3.08, respectively. Notably, MotionBase achieves a significantly higher average score of 3.82, outperforming the other two datasets.
> >
> > To improve the quality of text descriptions, we adopt the following method. Specifically, our latest motion descriptions are structured into three hierarchical and complementary levels: an overall description of the entire body, detailed part-level description for each body part, rule-based description of each joint's relative movement derived from joint positions (e.g., "The left hand is wide apart from the right hand. It is on the ground, the left upper arm, both legs and the left forearm are aligned horizontally ..."). We condition GPT-4o with two levels of text description while using the remaining level as the target to be refined. GPT-4o then assesses and refines the textual content of the target level, enabling the generation of more precise and reliable text descriptions.
> >
> > [1] RoHM: Robust Human Motion Reconstruction via Diffusion
> >
> > [2] WHAM: Reconstructing World-grounded Humans with Accurate 3D Motion

---

> > > ### Author Response · Authors · 2024-11-22
> > > **Official Comment to Reviewer qpUP's weakness 2-3**
> > >
> > > **W2. Could you clarify the video collection process? Many web videos are lengthy and include various camera shots. Which film shot boundary detection algorithm are you using? Additionally, how many frames are input into the LLM to generate the text? Providing more details on these aspects would be helpful.**
> > >
> > > Sure. Due to space limitations, we could not provide a detailed explanation of our data curation process for web videos before. Our videos are sourced from two main resources: open-source datasets like InternVid and self-collected web videos. For these videos, We use a pretrained 2D human keypoint detection model to filter out videos without visible human activities. Additionally, rule-based methods are applied to ensure the human bounding box occupies a significant portion of the frame, making the human movement clearly visible. In addition, videos containing only partially visible humans are removed to ensure the quality of potential motion data extracted from the videos. By doing this, we ensure the quality of remaining human-related videos.
> > >
> > > For the shot boundary problem you concern, here is a brief introduction of our solution.
> > > 1. For videos under 30 seconds or those with explicit temporal boundaries, we directly use the video clip or provided boundaries to segment the video into shorter clips.
> > > 2. For videos longer than 30 seconds, we use a scene detection model to roughly divide the video into smaller segments.
> > > 3. For each segment, we adopt the following steps to further slice it into shorter clips:
> > >     - At the beginning, the human with the largest bounding box is selected as the anchor, and we track their trajectory throughout the segment.
> > >     - When the trajectory is interrupted, the start and end times are marked as the boundaries of a new clip.
> > >     - The process repeats by identifying the next largest visible human in subsequent frames and tracking their trajectory.
> > >     - This recycling process continues until no humans are visible in the video.
> > >     - Clips without any visible humans are filtered out.
> > > 4. After these steps, if a clip is still longer than 60 seconds, we randomly slice it into several sub-clips, ensuring each of them is shorter than one minute.
> > > For you last question, we use 4 frame to represent one second following most previous works, the input length of the LLM is varied to increase the training performance.
> > >
> > > **W3. The experiments with static data ablation study are not fair. Does the validation set contain static data and synthetic data?**
> > >
> > > To address your concerns, we conducted experiments on a test set filtered to exclude static and synthetic data. The results in Table 2 indicate that static and synthetic data still provide significant value. By incorporating additional static semantic prompts during model training, we enable the model to effectively distinguish between dynamic and static actions. This allows us to leverage a larger volume of data to establish a stronger model prior, ultimately enhancing overall model performance.
> > > | Train Set         | R@1   | R@3   | FID    | MMDist  |
> > > |-------------------|-------|-------|--------|---------|
> > > | Real              | 0.196 | 0.474 | 0.006  | 1.647   |
> > > | w/o static & syn  | 0.167 | 0.396 | 1.740  | 2.323   |
> > > | w/o static        | 0.166 | 0.393 | 1.780  | 2.356   |
> > > | MotionBase        | **0.168** | **0.399** | **1.614** | **2.300** |

---

> > > > ### Author Response · Authors · 2024-11-22
> > > > **Official Comment to Reviewer qpUP's questions**
> > > >
> > > > **Q1. The FID in Table 6 is so wired. The FID of reconstruction is 1.76 while the generation FID in Table 3 is 0.166.**
> > > >
> > > > There might be a misunderstanding regarding the results. The FID of 1.76 in Table 6 corresponds to using LFQ as the motion tokenizer, whereas in Table 3, we use VQ as the tokenizer, as explained in the implementation details. To clarify further, as shown in the first row of Table 1, VQ achieves an FID of 0.078 on HumanML3D, which aligns with the results in Table 3. Due to time constraints and limited resources, we were unable to fully validate the effectiveness of combining LFQ and LLM. The inclusion of LFQ in our study is mainly intended as a preliminary exploration to observe the potential of alternative tokenization methods for motion representation. Even though, we plan to conduct a more comprehensive investigation into this direction in future work.
> > > >
> > > > **Q2. What do the authors get from scaling experiments? Did the author see any hope for emerging? The shown examples are common cases, that can be also observed in other motion generation work.**
> > > >
> > > > Our experiments reveal a clear trend: larger motion models and more extensive training data consistently improve motion generation performance. This finding contrasts with some previous works, such as MotionGPT [1], which reports performance drops with larger T5 models. Furthermore, due to limitations in data scale, no prior research has demonstrated a scaling law in human motion understanding, where expanding the dataset with a massive amount of motion data leads to consistent improvements. In contrast, our results show the hope that both increasing model parameters and expanding the dataset contribute to enhanced generalization. For example, models trained on MotionBase-1.0 (with 1 million samples) outperform those trained on MotionBase-0.5 (with 500,000 samples). Additionally, larger motion models built on more powerful language models (e.g., Llama2-13B vs. GPT2-medium) consistently achieve higher R-precision and lower FID scores. Beyond the common results, our model performs well on **out-of-distribution** and unseen commands, highlighting the value of scalable data and model architectures. We also present motion examples that are ready for deployment in both **simulated and real-world** environments.
> > > >
> > > > [1] MotionGPT: Human Motion as a Foreign Language
> > > >
> > > > **Q3. Do the supervised labels include only motion tokens, or does it contain both text and motion tokens?**
> > > >
> > > > The supervised labels include both text and motion tokens. As shown in Table 10 in Appendix C.4, we compare the performance of two supervised training methods: one using only motion tokens and the other incorporating both text and motion tokens. The results demonstrate that incorporating both significantly enhances performance. We hypothesize that this improvement is due to the potential issue of catastrophic forgetting in the language model when supervision is limited to motion tokens alone. Including text tokens helps maintain the language model's capabilities, leading to better overall performance.
> > > >
> > > > **Q4. Did the authors conduct zero-shot text testing? For instance, could the largest model generate motions for descriptions like, "The old man with a broken leg is walking forward slowly with a cane"?**
> > > >
> > > > Yes, we have conducted **out-of-domain evaluations** on 90K unseen motions to assess the zero-shot capabilities of our model. As shown in Table 5, our model, powered by the large-scale MotionBase dataset, demonstrates significantly better performance compared to models trained on smaller datasets, such as HumanML3D. Additionally, we provide a set of generation examples on our visualization platform to illustrate the model's capabilities. However, the specific example you mentioned may not be directly applicable to current motion models. This is because existing motion data predominantly represents individuals with two healthy feet. In this work, motion data is further refined using a physical-law-based RL policy to ensure balance and realism in the generated motions. Therefore, it's merely possible to generate motions of ``an old man with a broken leg''.
> > > >
> > > > We hope our clarifications address your questions, and we kindly ask if there are additional concerns we can address to further improve your support for the paper. Thank you again for your valuable suggestions.

---

> ### Comment · Reviewer_qpUP · 2024-11-26
> **Reply to rebuttal**
>
> Thanks for the rebuttal. However, I still have the remaining questions.
>
> 1. The content of dataset collections and processes, including the training of policies and motion generators, are not included in the main scripts and appendix. In addition, they lack the quantitative experiments to evaluate the policy.
>
> 2. Furthermore, the motion qualities have not been verified through quantitative experiments. I suggest the author carefully compare their methods on some benchmarks, like RICH, EMDB2, etc. The Gemini can somehow evaluate the alignment between the motion and the text. However, it is improper to evaluate the motion qualities.
>
> 3. LFQ is one of the main contributions of the paper, however, the authors scale the model with VQ instead of LFQ. I think this should not be regarded as a good thing in ICLR.
>
> Overall, I decided to keep my score now.

---

> > ### Author Response · Authors · 2024-11-28
> >
> > Dear reviewer, thank you for your feedback.
> >
> > 1. Thanks for your notice. We have included these contents in our new appendix version. For policy training, we use the official implementation from PHC[1] repository, which achieves a 97.1% success rate in tracking AMASS motions. For the motion completion model, we use the official model from MotionGPT[2].
> >
> > [1]Perpetual Humanoid Control for Real-time Simulated Avatars
> >
> > [2]MotionGPT: Human Motion as a Foreign Language
> >
> > 2. The RICH and EMDB2 benchmarks are primarily designed for pose estimation evaluation, which differs from our verification objectives. Quantitative evaluation of motion quality in large-scale video data is challenging due to the lack of standard ground truth. In our motion estimation pipeline, we use officially pre-trained models that have been validated on their respective benchmarks, maintaining consistent performance with the original models. Considering the limitations of Gemini evaluation, we employ multiple methods to ensure data quality, including physical constraints and empirical filtering.
> >
> > 3. The primary contributions of our paper are the introduction of a large-scale dataset and the validation of the scaling law. We chose VQ as the primary method to ensure a fair comparison with existing baselines, as most current text-to-motion approaches are based on VQ. However, we understand your concerns and have conducted additional experiments to highlight the capabilities of LFQ. Specifically, we present data scaling experiments using GPT-2 in the first block and parameter scaling experiments with 0.02M training samples in the second block. These results are consistent with our initial conclusions, demonstrating robustness across scaling scenarios. Moreover, LFQ shows a slight performance improvement over VQ when evaluated with GPT-2. Due to time and GPU resource limitations, we were unable to conduct further experiments on 7B/13B models with 1M data at this time, but we plan to include them in future revisions. The current additional results have been included in the appendix for reference.
> >
> > | Model | #Inst | MotionX |  |  | MotionBase |  |  |
> > |-------|-------|---------|---------|---------|------------|---------|---------|
> > |       |       | R@1 | R@3 | FID | R@1 | R@3 | FID |
> > | GPT-2-VQ | 1M | 0.36 | 0.62 | 5.08 | 0.26 | 0.54 | 0.52 |
> > | GPT-2-LFQ | 0.02M | 0.16 | 0.34 | 76.21 | 0.04 | 0.08 | 136.25 |
> > | GPT-2-LFQ | 0.08M | 0.33 | 0.55 | 6.24 | 0.06 | 0.14 | 128.07 |
> > | GPT-2-LFQ | 1M | 0.39 | 0.62 | 4.28 | 0.32 | 0.60 | 0.45 |
> > | GPT-2-LFQ | 0.02M | 0.16 | 0.34 | 76.21 | 0.04 | 0.08 | 136.25 |
> > | LLaMA-2-7B-LFQ | 0.02M | 0.22 | 0.38 | 68.54 | 0.06 | 0.15 | 125.08 |
> > | LLaMA-2-13B-LFQ | 0.02M | 0.20 | 0.35 | 71.23 | 0.08 | 0.18 | 119.03 |
> >
> > Thank you again for your valuable review. If you have any unresolved concerns, please feel free to let us know! Or if our responses have addressed your questions, we would be grateful if you could consider adjusting your score accordingly.
> >
> > Best regards,
> >
> > Authors

---

### Official Review · Reviewer_pT95 · 2024-11-03

**Soundness:** 3
**Presentation:** 3
**Contribution:** 4
**Rating:** 6
**Confidence:** 3

**Summary:**

The paper is to answer the research question of "can a large motion model be a promising direction for motion generation?", and the designed a data colletion pipeline which collects multi-modal information including RGB, depth and bounding box with multi-person.
In addition the paper introduces a method to expand the codebook capacity, named lookup-free approach for motion tokenization, for better motion representation.

**Strengths:**

This paper presents the first large-scale dataset specifically designed for motion generation, featuring richly multi-modal data accompanied by hierarchical text descriptions. MotionBase, the dataset introduced, is expected to be highly beneficial for the advancement of future research in motion generation and to serve as a valuable resource for the computer vision community. The dataset offers researchers access to an extensive collection of motion data, enabling more robust analysis and development of large motion model.

**Weaknesses:**

I have minor concerns on this paper.

The layout of the paper is somewhat challenging for readers. It contains numerous messages and analyses, requiring readers to scroll up and down frequently to locate referenced tables. Additionally, due to page limitations, many explanations are placed in the Appendix. Tables and figures are positioned mid-page without aligning well with the paragraph height, disrupting the flow.

This paper was the first to introduce the concepts of partitioning body parts and 2D quantization, making it a valuable reference. (Pi, H., Peng, S., Yang, M., Zhou, X., & Bao, H. (2023). Hierarchical generation of human-object interactions with diffusion probabilistic models. In Proceedings of the IEEE/CVF International Conference on Computer Vision, pp. 15061-15073.)

Minor Issues and Typos:
Appendix D: "quantitative results" should be "qualitative results."
Figure 4: It may improve clarity to add a y-axis label.

**Questions:**

In Table 4, what is the ratio between synthetic, static, and real data? It can be brefiely explained in table caption.

I have the concern of the quality of occlusion cases or blurred images. How the authors recognize the motion is blurred or occluded?
In multi-person settings, the occlusion might be very common.

Since this is a dataset paper, I expect the more detailed explanation and instructions of the benchmark will be released once the paper upon the paper acceptance.

---

> ### Author Response · Authors · 2024-11-22
> **Official Comment to Reviewer pT95's weakness**
>
> We appreciate the reviewer’s careful assessment and acknowledgment of our paper's clarity, motivation, and contributions.
>
> Before reading our feedback, we introduce a visualization website to validate the quality of our dataset at http://www.motionbase3d.com. (Please note: this link may be unstable, and refreshing the page 10~20 times may be required to view the content). In case the reviewer is unable to access the site, we have also provided an anonymous cloud storage link containing all the examples featured on the website: [click this link](https://www.dropbox.com/scl/fo/6w7yuhun8mpuz9yuaz5ej/ALUo_dOLYIyNgOZ3Ll85JCI?rlkey=yecydfnno43b5o602ae2fmr0w&st=rq97kdcw&dl=0). The platform includes visualized samples from our dataset, physically grounded examples of refined motions, rendered results generated by different motion models, and live demonstrations in both simulated environments and real-world settings (using H1 and G1 UNITREE humanoid robots). We hope this platform addresses the data quality concern reviewers may have. If the reviewer requires additional information or examples, we are more than happy to upload further relevant results upon request. Additionally, our dataset is highly scalable due to the robust data collection pipeline. As of the start of the rebuttal period, we have collected over 1.5 million motion trajectories and 4 million motion-text pairs, which are 1.5X compared to our first submission. These datasets have undergone three rounds of rigorous data filtering to ensure their quality.
>
> Here are our responses to weaknesses and questions.
>
> **W1. The layout of the paper is somewhat challenging for readers. It contains numerous messages and analyses, requiring readers to scroll up and down frequently to locate referenced tables. Additionally, due to page limitations, many explanations are placed in the Appendix. Tables and figures are positioned mid-page without aligning well with the paragraph height, disrupting the flow.**
>
> Thank you for your comments on the paper's layout. We want to present as much content as possible therefore the layout could be not well-aligned. In the revised manuscript, we will: (1) Optimize Figure/Table Placement: Align figures and tables with the paragraph height and place them as close as possible to the relevant text to enhance readability. (2) Improve Referencing: Ensure clear and unambiguous references to figures and tables, reducing the need for excessive scrolling. (3) Streamline Appendix: Integrate critical explanations and analyses into the main text where appropriate, while reorganizing the remaining appendix content for better structure and clarity.
>
> **W2 & W3. Important reference and minor issues and typos.**
> Thank you for your notice. We have cited this important reference and corrected the issues and typos you pointed out in the revised paper.

---

> ### Author Response · Authors · 2024-11-22
> **Official Comment to Reviewer pT95's question**
>
> **Q1. What is the ratio between synthetic, static, and real data? in Table 4**
>
> In the latest version, synthetic data accounts for approximately 28% of the total dataset, while real data makes up about 72%. Additionally, static data constitutes about 44% of all data. The proportion of synthetic and static data will continue to decrease as MotionBase expands. We have included these proportion numbers in the revised version of Table 4.
>
> **Q2. The quality of occlusion cases or blurred images. How do the authors recognize that the motion is blurred or occluded? In multi-person settings, occlusion is expected to be very common.**
>
> Occlusion and motion blur are indeed very common in human-related videos. To avoid these issues, we adopt the following steps:
>
>   - 2D Keypoint Detection Fittering. We first sample key frames from each video. A pretrained 2D keypoint detector is then used to extract skeleton keypoints for each human in the key frames. If a significant portion of keypoints has predicted confidence scores below a specific threshold, we consider the human motion to be occluded and exclude it from further processing.
>
>   - Segment Filtering. We utilize a visual foundation model, such as Segment Anything, to generate segmentation masks for each frame. If a large object is detected in front of the human, indicating occlusion, we filter out the corresponding motion data.
>
>   - Adjacent Frame Smoothing. To handle motion blur, we track the trajectory of each human whose motion needs to be extracted from the video. For timestamps with low-confidence keypoint scores, we smooth the trajectory using adjacent detection results to ensure continuity and accuracy.
>
> In addition to occlusion, we apply many additional post-processing techniques to enhance the  quality of the dataset. If the reviewer has any further questions about the process, please feel free to ask. We are willing to provide more details.
>
> We hope our clarifications address your questions, and we kindly ask if there are additional concerns we can address to further improve your support for the paper. Thank you again for your valuable suggestions.

---

> > ### Comment · Reviewer_pT95 · 2024-11-24
> >
> > Thank you for taking the time to address my questions and concerns. I appreciate the effort the authors have put into this work, and I continue to believe that the dataset has the potential to be a valuable resource for the community. However, I also share the concern raised by other reviewers about the high proportion of static data (44%), which could potentially limit the dataset's utility for certain applications.
> > Given the complexity of the topic and the specific concerns raised by Reviewer WfZ5, I feel it would be most appropriate to reserve my final evaluation until the ongoing discussion between the authors and Reviewer WfZ5 concludes.

---

> > > ### Author Response · Authors · 2024-11-25
> > >
> > > Thank you for your response! We appreciate your concern and would like to clarify that, even excluding static data, our dataset contains 800K dynamic motion sequences—ten times larger than existing benchmarks. We believe that this volume of data, which is also scalable, substantiates the contribution we have claimed. Furthermore, the effectiveness of static data has been demonstrated in our ablation study (Table 4). We hope these points address your concern to some extent.
> > >
> > > If there are any remaining issues or concerns that were not addressed in our previous discussion, we would sincerely appreciate it if you could point them out. We are more than happy to provide further clarifications or answer any additional questions!

---

> > > > ### Comment · Reviewer_pT95 · 2024-11-27
> > > > **Final recommendation**
> > > >
> > > > While I am favorable to the paper because I believe that the dataset proposed in the work will be helpful for the community, I am not fully confident in my evaluation of the paper outside the human motion generation aspect, and it seems that other reviewers still have some concerns. Therefore, I will position myself more conservatively by reducing my influence on the final decision.
> > > >
> > > > **Final Score**
> > > > **6: marginally below the acceptance threshold**

---

> > > > > ### Comment · Reviewer_pT95 · 2024-11-27
> > > > >
> > > > > I apologize for the confusion; I meant **6: marginally above the acceptance threshold**.
> > > > >
> > > > > I believe it would be reasonable to determine my score after the discussion between the authors and WfZ5 concludes, however, I am finalizing my score now as the deadline is only a few hours away. Hence, I have taken a conservative position on the unresolved discussion and leave it for the Area Chair to make the final decision.
> > > > >
> > > > > This score adjustment reflects that I am not entirely confident in my evaluation, while still believing that the dataset would be very helpful for the community. I am confident in my evaluation of the various meaningful messages derived from the research question, *"Can a large motion model be a promising direction for motion generation?"* assuming that the experimental results are valid. These contributions are significant, and I believe that the progression from LLM to LMM represents the right future direction.
> > > > >
> > > > > While I value the direction the paper proposes, I find myself limited in my ability to fully evaluate the validity of the data collection pipeline with LLM due to my lack of expertise in LLM domain. Recognizing the potential issues raised by other reviewers, such as the effectiveness of static motions and hierarchical text evaluation. I cannot maintain my score without a stronger basis for my evaluation. As a result, I believe it is appropriate to adopt a more conservative stance.

---

> > > > > > ### Author Response · Authors · 2024-11-29
> > > > > >
> > > > > > Dear reviewer,
> > > > > >
> > > > > > As we near the conclusion of the discussion phase, we would like to inquire if our response has effectively addressed your inquiries. In the rebuttal, we have provided explanations for your questions and concerns. Should our explanations have met your expectations, we kindly request your consideration in revising the score accordingly.
> > > > > >
> > > > > > Should you have any additional comments or concerns, we are more than willing to address them promptly during the response period. We deeply appreciate your constructive and insightful feedback, which has greatly contributed to the refinement of our work.
> > > > > >
> > > > > > Best regards,
> > > > > >
> > > > > > Authors

---

> > > > > > ### Author Response · Authors · 2024-11-30
> > > > > >
> > > > > > Dear reviewer,
> > > > > >
> > > > > > Thank you again for your valuable review. We have responded to your question and concern. We hope to hear back from you! If you have any unresolved concerns, please feel free to let us know! Or if our responses have addressed your questions, we would be grateful if you could consider adjusting your score accordingly.
> > > > > >
> > > > > > Best,
> > > > > >
> > > > > > Authors

---

> > > > > > ### Author Response · Authors · 2024-11-30
> > > > > >
> > > > > > Dear Reviewer pT95,
> > > > > >
> > > > > > We hope this message finds you well. We think it would be necessary to remind you that your prompt decision to lower the score may have been perceived by reviewers who invited you to further discussion as potentially careless and subjective, even considering to reduce the weight of your rating. Despite this, we deeply appreciate your positive perspective and the effort you have invested in engaging with our rebuttal. We believe your insights are critical to ensuring a constructive and well-rounded final discussion.
> > > > > >
> > > > > > **Are you still willing to participate in the ongoing discussion with WfZ5 and us?** If so, we would greatly appreciate it if you could provide additional details about your decision or share any unresolved questions or concerns. Should there be specific points we have not addressed to your satisfaction, please let us know—we would be more than happy to provide a prompt and thorough response.
> > > > > >
> > > > > > Best,
> > > > > >
> > > > > > Authors

---

> ### Author Response · Authors · 2024-11-27
>
> Dear reviewer pT95.
>
> Sincerely thank you for your reviewering time. We would like to kindly remind you that **the rebuttal period has been extended to December 2nd, giving us an additional week to address your concerns.** Please feel free to reach out if you have any further questions, and **we guarantee a prompt response.**
>
> Considering your mentioned the concerns about static data, we attach this part in the bottom. If you have any additional question, please let us know. We have provided over 20 pages of rebuttal material, which has been a time-intensive effort. We greatly value this opportunity to discuss and refine our work, and we sincerely hope we are moving in the right direction.
>
> Best regards,
> Authors
>
>
> **Responses to static data concern:**
> - Most importantly, our dataset comprises over 800K dynamic motion sequences, making it at least 10 times larger than existing benchmarks. This scale ensures that even if all static poses were removed from MotionBase, the dataset's overall contribution would remain largely unaffected.
> - We argue that static poses provide valuable additional knowledge about human activity, similar to how static images contribute to video understanding or single-frame object bounding boxes aid video tracking. The effectiveness of static data is validated in the ablation studies presented in Table 4 and Table 8 in the appendix, where we test on datasets without static data. In fact, we suppose static poses are particularly beneficial for our LLM-based decoder, as they enhance its understanding of positional and rotational relationships among joints. This is especially important when motion vocabulary is learned from scratch during pretraining.
> - The static data will not result in "over smoothing" because:
>     - During fine-tuning, we design specific prompts to guide the model's generation. For static poses, we include additional language prompts, such as 'keep the action still,' to help the model distinguish between generating a static pose and a dynamic motion.This approach leverages the well-known ability of large generation models based on LLMs [1,2,3,4,5] to produce diverse outputs through carefully crafted prompts. In addition, no negative effects are observed when combining images and videos, or bounding boxes and segmentation masks, or 3D skeleton poses and sequences, or vision and audios during training. We believe this principle should extend to motion as well.
>     - Visualization. To better prove our conclusion, in the visualization appendix, we present generation results from both models trained with and without additional language prompts. The clear differences demonstrate that **with additional language prompts, the model can distinguish between different motion distributions, thus not affecting the generation of dynamic motions.**
>     - Many common training strategies, like weighted sampling, weighted gradient, progressively increasing the ratio of dynamic motion, all these strategies can improve the stability of training, ensuring the model to avoid "over smoothing".

---

> ### Author Response · Authors · 2024-11-28
>
> Dear reviewer pT95.
>
> Thank you again for your valuable review. We have responded to your concern about static data quality. We hope to hear back from you. **If you have any unresolved concerns, please feel free to let us know! Or if our responses have addressed your questions, we would be grateful if you could consider adjusting your score accordingly.**
>
> Best regards,
>
> Authors

---

### Official Review · Reviewer_pqNv · 2024-11-04

**Soundness:** 3
**Presentation:** 3
**Contribution:** 3
**Rating:** 6
**Confidence:** 4

**Summary:**

In this paper, the authors introduce MotionBase, a large-scale human motion generation benchmark featuring over one million motion sequences, a fifteen-fold increase over previous datasets, with multimodal data and detailed text descriptions. The authors demonstrate that scaling both data and model size significantly improves motion model performance, particularly with synthetic data and pseudo labels to reduce data acquisition costs. The authors also propose a novel 2D lookup-free motion quantization approach to enhance motion information retention and expand codebook capacity. Experimental results on various datasets validate the efficacy of their approach, with notable performance on out-of-domain data.

**Strengths:**

1. The paper introduces MotionBase, a large-scale dataset comprising over one million human motion sequences, designed to support more comprehensive training and evaluation of motion generation models.

2. The paper identifies key factors influencing the effectiveness of large motion models, underscoring the importance of scaling both data and model size.

3. This paper proposes a 2D lookup-free motion quantization method that enhances motion representation while retaining essential information, thereby contributing to improved model performance.

**Weaknesses:**

1. While MotionBase is introduced as a benchmark with the potential to enhance motion model performance, the paper lacks a thorough comparative analysis across varied methods to demonstrate MotionBase's influence on model efficacy. Additional baselines and a broader selection of models trained on MotionBase would more robustly substantiate its claimed advantages.

2. The paper does not include visual comparisons of motions generated by models trained on the baseline Motion-X dataset versus those trained on the proposed MotionBase dataset.

3. Including the ground truth R-Precision and FID scores in relevant tables would strengthen the presentation and transparency of the results.

4. The paper would benefit from dynamic visualizations within the qualitative analysis of the motions in the proposed datasets, which could provide a clearer and more engaging illustration of the dataset's scope and quality.

**Questions:**

1. It will be interesting to see the scalability of different architectures. Have the authors explored fine-tuning existing methods, such as MotionGPT[1] or MoMask[2], with larger parameter settings on the MotionBase dataset?

2. Regarding the automated evaluation metrics referenced by the authors, it is also noteworthy that the R-precision scores are relatively low on the proposed large-scale MotionBase dataset, potentially weakening the benchmarking results. Implementing text-motion retrieval models like TMR[3] may provide a more accurate evaluation of model performance.

[1] MotionGPT: Human Motion as a Foreign Language.

[2] MoMask: Generative Masked Modeling of 3D Human Motions.

[3] TMR: Text-to-Motion Retrieval Using Contrastive 3D Human Motion Synthesis.

---

> ### Author Response · Authors · 2024-11-22
> **Official Comment to Reviewer phNv's weaknesses**
>
> Thank you for your appreciation of our work, and proposing thoughtful feedback which helps us further improve our work.
>
> Before reading our feedback, we introduce a visualization website to validate the quality of our dataset at http://www.motionbase3d.com. (**Please note: this link may be unstable, and refreshing the page 10~20 times may be required to view the content**). In case the reviewer is unable to access the site, we have also provided an anonymous cloud storage link containing all the examples featured on the website: [click this link](https://www.dropbox.com/scl/fo/6w7yuhun8mpuz9yuaz5ej/ALUo_dOLYIyNgOZ3Ll85JCI?rlkey=yecydfnno43b5o602ae2fmr0w&st=rq97kdcw&dl=0).
> The platform includes visualized samples from our dataset, physically grounded examples of refined motions, rendered results generated by different motion models, and live demonstrations in both simulated environments and real-world settings (using H1 and G1 UNITREE humanoid robots). We hope this visualization platform addresses the data quality concern reviewers may have. If the reviewer requires additional information or examples, we are more than happy to upload further relevant results upon request. Additionally, our dataset is highly scalable due to the robust data collection pipeline. As of the start of the rebuttal period, we have collected over 1.5 million motion trajectories and 4 million motion-text pairs, which are 1.5 times compared to our first submission. These datasets have undergone three rounds of rigorous data filtering to ensure their quality.
>
> Here are our responses to weaknesses.
>
> **W1 & Q1. The paper lacks a thorough comparative analysis across varied methods to demonstrate MotionBase's influence on model efficacy. Additional baselines and a broader selection of models trained on MotionBase would more robustly substantiate its claimed advantages.**
>
> In this work, our primary focus was on constructing the high-quality MotionBase dataset and investigating the scaling law of model size and dataset scale. Due to time limitation, we were unable to perform a comprehensive comparison among all existing methods. We acknowledge this as an important area for improvement and plan to include additional baseline models and experimental results to provide a more thorough analysis of our findings.
>
> **W2.  The paper does not include visual comparisons of motions generated by models trained on the baseline Motion-X dataset versus those trained on the proposed MotionBase dataset.**
>
> We showcase the generated results from different models on the model generation result page of our visualization platform, which allows you to directly compare the motions generated by models trained on Motion-X and MotionBase. We believe this visual comparison will provide a clearer and more intuitive demonstration of the improvements MotionBase brings to model training.
>
> **W3. Including the ground truth R-Precision and FID scores in relevant tables would strengthen the presentation and transparency of the results.**
>
> We have added the ground truth R-Precision and FID scores to Table.2 and Table.3 in our revised version. Here are the results:
> | Dataset      | Fid Real | R@1 Real | R@3 Real | MMDist Real |
> |--------------|----------|----------|----------|-------------|
> | HumanML3D    | 0.002    | 0.511    | 0.797    | 2.974       |
> | MotionX      | 0.038    | 0.496    | 0.821    | 2.438       |
> | MotionBase   | 0.011    | 0.290    | 0.563    | 3.480       |
>
> **W4. The paper would benefit from dynamic visualizations within the qualitative analysis of the motions in the proposed datasets, which could provide a clearer and more engaging illustration of the dataset's scope and quality.**
>
> We fully agree that dynamic visualizations are essential for effectively demonstrating the value of our dataset. This concern has also been raised by other reviewers. To address this, we build a visualization platform that allows all reviewers to directly and clearly explore a wide range of motion examples, providing a more intuitive and comprehensive understanding of the richness and high quality of our dataset. We sincerely invite you to visit the platform and experience these dynamic examples. Your comment is important.

---

> > ### Author Response · Authors · 2024-11-22
> > **Official Comment to Reviewer phNv's question.**
> >
> > Here are our responses to questions.
> >
> > **Q2. Regarding the automated evaluation metrics referenced by the authors, it is also noteworthy that the R-precision scores are relatively low on the proposed large-scale MotionBase dataset, potentially weakening the benchmarking results. Implementing text-motion retrieval models like TMR[3] may provide a more accurate evaluation of model performance.**
> >
> > We also notice the unsatisfying results of R-precision. To address this, we have trained a more robust retrieval model on MotionBase and re-evaluated the results based on this model. The new evaluation results are updated in our revised paper, which demonstrate the benchmarking conclusions more clearly. Due to time constraints, we have not yet conducted evaluations using TMR or other more recent retrieval models, but we plan to perform these experiments in future work.
> >
> > We hope the above clarifications address your questions, and we kindly ask if there are additional concerns we can address to further improve your support for the paper. Thank you again for your valuable suggestions.

---

### Official Review · Reviewer_CH9D · 2024-11-05

**Soundness:** 4
**Presentation:** 3
**Contribution:** 3
**Rating:** 8
**Confidence:** 3

**Summary:**

Paper introduce motionbase, a motion generation benchmark trained on large amount of data with focus on motion generation with LLMs.

**Strengths:**

The work is well motivated in terms:
 Showing the gap of prior work and lack of domain generlization
Showing limitation of prior metrics
A new motion codebook


The new dataset is quite large in comparison with prior ones, which is a valuable addition to the community. It comes with a good set of text descriptions.

Evaluation on multiple datasets and multiple models with strong baselines.
Answer to important questions like the need of scale and model size impact on the task
Discussion on OOD behaviour
Ablation of motion quantization

**Weaknesses:**

I do not see much of concerns about the work, more of questions.

**Questions:**

– Questions:
How did the author verify the correctness/accuracy of the pose estimation
What do authors think about properties of a new metric?

---

> ### Author Response · Authors · 2024-11-22
> **Official Comment to Reviewer CH9D**
>
> We appreciate the reviewer’s careful assessment and acknowledgment of our paper's clarity, motivation, and contributions.
>
> Before reading our feedback, we introduce a visualization website to validate the quality of our dataset at http://www.motionbase3d.com. (**Please note: this link may be unstable, and refreshing the page 10~20 times may be required to view the content**). In case the reviewer is unable to access the site, we have also provided an anonymous cloud storage link containing all the examples featured on the website: [click this link](https://www.dropbox.com/scl/fo/6w7yuhun8mpuz9yuaz5ej/ALUo_dOLYIyNgOZ3Ll85JCI?rlkey=yecydfnno43b5o602ae2fmr0w&st=rq97kdcw&dl=0).
> The platform includes visualized samples from our dataset, physically grounded examples of refined motions, rendered results generated by different motion models, and live demonstrations in both simulated environments and real-world settings (using H1 and G1 UNITREE humanoid robots). We hope this visualization platform addresses the data quality concern reviewers may have. If the reviewer requires additional information or examples, we are more than happy to upload further relevant results upon request. Additionally, our dataset is highly scalable due to the robust data collection pipeline. As of the start of the rebuttal period, we have collected over 1.5 million motion trajectories and 4 million motion-text pairs, which are 1.5 times compared to our first submission. These datasets have undergone three rounds of rigorous data filtering to ensure their quality.
>
> Here are our responses to the questions.
>
> **Q1: How did the author verify the correctness/accuracy of the pose estimation?**
>
> We acknowledge that this is a key concern for all reviewers so we provide a visualization platform for all reviewers to easily examine our data. In addition, we outline the strategies used to verify the accuracy of the estimated human poses in our dataset:
>   - Physical Grounding: We train an RL-based policy $\pi_{\rm refine}$, which takes raw poses as input (target poses) and generates refined pose sequences. If the policy successfully tracks the raw poses and produces smooth, balanced motions, we assess the data as high-quality. This approach not only allows the policy $\pi_{\rm refine}$ to serve as a discriminator for verifying pose accuracy and avoiding issues like jittering or sliding, but also acts as an effective method to refine the raw motions.
>   - Rule-based Methods: In the latest version of MotionBase, we incorporate an additional level of text: rule-based descriptions derived from joint positions and angles. These precise descriptions enable us to assign rule-based scores to each motion. Motions with low scores are filtered out to improve overall quality.
>   - Manual Review via Visualization Platform: We conduct random sampling for manual check and leverage a visualization platform to facilitate efficient assessment of motion quality.
>
>
> **Q2. What do authors think about properties of a new metric?**
>
> Regarding this, we believe that an ideal text-to-motion metric should exhibit the following properties:
>   - Fine-grained Quality Assessment: The new metric should capture detailed motion features, such as local hand and leg movements in addition to global postures. The metric should evaluate multiple aspects of quality, including naturalness, smoothness, fidelity, and style. In fact, the fine-grained assessment has long been ignored by previous works, partially because the lack of motions required to provide subtle differences. Our MotionBase provides an opportunity to achieve this.
>   - Human-like Perception: The metric should strongly align with human evaluation, considering factors like human kinematics and biomechanics. It should also account for perceptual similarities, even in the absence of reference motions.
>   - Physical Plausibility: The metric should evaluate adherence to physical laws, such as gravity and inertia, and analyze dynamic properties like joint torques and ground reaction forces.  This is particularly important for applications in robotics.
>   - Strong Generalization: Ideal metrics should be capable of handling the one-to-many nature of text-to-motion mapping. They should generalize well across datasets and various motion types, such as walking, running, and jumping.
>   Due to time and space constraints, we could not comprehensively address or explore all these aspects. We plan to delve deeper into these directions in future work.
>
> We hope our clarifications address your questions, and we kindly ask if there are additional concerns we can address to further improve your support for the paper. Thank you again for your valuable suggestions.

---

> ### Comment · Reviewer_CH9D · 2024-11-30
>
> After checking everything here and the discussion
> I’m keeping my score as is. The works itself is valuable with some minor concerns that can be addressed or progressed in a later work.

---

### Comment · Reviewer_WfZ5 · 2024-11-22
**reply**

Thanks for your discussion. I have carefully checked reviews from other reviewers and checked the latest response. I think we still need some discussions.

Before stating my comments, I would like to clarify that the significant revisions related to contributions are not supported officially, according to the review guidance. The review should be based on the original submission, not the major revision. Although there are these issues, I would also like to discuss more about them. Besides, external links are not permitted because this will not be track by the fair reviewing process. Why not use the supp.?

1. For your motion-capturing process, what is the policy you use? PHC? RFC? ASE? For the statement “If a refined motion maintains balance in the simulated environment for a specific duration, it can be regarded as high-quality with no severe issues …”, what is the duration, and how to compute the success rate? What is your success rate? Besides, how do you deal with videos with shot cuts?
2. I also noticed reviewer `qpUP` has similar concerns on static motions. I think the data is the pose, not motion or animation. Our animation community does not recognize the pose as motion. When the data distribution includes such data, it will be harmful to your generation results. I have also provided the blog by Daniel in my previous review.
3. The video demos on the website are not friendly for usage. Please provide them in supp..
4. For the Gemini-score from 1 to 5, how can the method be fair enough to evaluate whether the texts are aligned with motion? With motion input?
5. For the hierarchical text issue, is your evaluator trained with hierarchical text? Why is the R-P of generated results higher than GT?
6. Why do your results of `Hierarchical`and `Full Param`are not the same in response? I do not see the result of T2M-GPT as a baseline in the response.
7. I am still a bit curious about citing reference [1]. Which part of reference [1] did the authors refer to?

Besides, I did not see any revision in the manuscript. (w/o any highlights)

Up to now, I will temporarily keep my rating and wait for replies from other reviewers. If any point of my review or statement is wrong, please directly point it out. This will help us to clarify issues.

---

> ### Author Response · Authors · 2024-11-23
> **Official Comment to Reviewer WfZ5's questions 1-2**
>
> Thank you for your prompt response! We sincerely appreciate the opportunity to address your concerns.
>
> To begin, note the corresponding revisions in our latest manuscript have been masked in blue, and we would like to highlight two key points:
>
> 1. **Revised manuscript:** We have not made significant modifications to our revised manuscript. All updates reflect responses to the reviewers' comments and questions, which align with our initial conclusions and do not contradict them.
>
> 2. **External links:** The primary reason for not using the OpenReview Supp to provide our demos is due to **OpenReivew's file size limitation of 100MB**, which is insufficient for our larger demo files. Since the anonymous website may be unstable, we provide an anonymous cloud drive link to show the demos, which is a widely accepted practice in current conference submissions. Additionally, we include a smaller zip file in OpenReview Supp, containing a subset of examples that fit with the 100MB restriction. After reviewing ICLR's guidelines (https://iclr.cc/Conferences/2025/CallForPapers), **we confirm that anonymous links are not forbidden during both the paper submission and discussion phase.**
>
> The following sections are our latest replies to the questions:
>
> **Q1. For your motion-capturing process, what is the policy you use? PHC? RFC? ASE? For the statement “If a refined motion maintains balance in the simulated environment for a specific duration, it can be regarded as high-quality with no severe issues …”, what is the duration, and how to compute the success rate? What is your success rate? Besides, how do you deal with videos with shot cuts?**
>
> We train the single primitive policy from PHC, which takes a raw motion sequence as input and generates refined joint positions using the simulator (IsaacGym). To ensure consistency in data format, we then convert the global joint positions into HumanML3D's representation using HumanML3D's conversion scripts. The success is determined by the duration of the simulation process. Following PHC's termination conditions, the policy is tasked with tracking the provided motion sequences. During simulation, we log the step number (corresponding to the frame index of the motion sequence) at which termination is triggered and calculate the duration for each sample. Later for each sample, we set a threshold as 50% of its original length, and if the duration of the refined motion sequence exceeds this threshold, it is considered a successful sample. The success rate is then defined as the ratio of successful samples to the total number of samples, with our conversion achieving a success rate of 51.4%. Regarding short cuts, if you mean "videos that have abrupt transitions between different scenes or camera angles", we apply a human tracking algorithm to ensure consistency in human motions across each video, as detailed in the data construction section.
>
> **Q2. I also noticed reviewer qpUP has similar concerns on static motions. I think the data is the pose, not motion or animation. Our animation community does not recognize the pose as motion. When the data distribution includes such data, it will be harmful to your generation results. I have also provided the blog by Daniel in my previous review.**
>
> - Most importantly, our dataset comprises over 800K dynamic motion sequences, making it at least 10 times larger than existing benchmarks. This scale ensures that even if all static poses were removed from MotionBase, the dataset's overall contribution would remain largely unaffected.
>
> - We argue that static poses provide valuable additional knowledge about human activity, similar to how static images contribute to video understanding or single-frame object bounding boxes aid video tracking. The effectiveness of static data is validated in the ablation study presented in Table 4. In fact, we suppose static poses are particularly beneficial for our LLM-based decoder, as they enhance its understanding of positional and rotational relationships among joints.  This is especially important when motion vocabulary is learned from scratch during pretraining. Furthermore, we ensure the reliability of our LLM-generated motion sequences by using purely dynamic motion data for instruction tuning.
>
> - Of course, we fully agree with both you and Daniel's blog that dynamic data plays a more critical role in motion learning. This is precisely why we prioritize collecting such data from human-related videos.

---

> > ### Author Response · Authors · 2024-11-23
> > **Official Comment to Reviewer WfZ5's questions 3-7**
> >
> > **Q3. The video demos on the website are not friendly for usage. Please provide them in supp.**
> >
> > We provide a smaller zip file in Openreview Supp. It's important to note that this file only contains partial examples, due to OpenReview's file size limitation (**<100MB**).
> >
> > **Q4. For the Gemini-score from 1 to 5, how can the method be fair enough to evaluate whether the texts are aligned with motion? With motion input?**
> >
> > Regarding your question on how the Gemini score (1-5) fairly evaluates the alignment between text and motion, we have outlined two methods of text evaluation in our response to W2.
> >
> > Firstly, we independently assess the quality of the text description. More importantly, we evaluate the alignment between the text and motion by providing both the text description and the corresponding rendered motion video as input to the Gemini-Pro model. This approach allows us to measure how accurately the text reflects the visual content. Specifically, we randomly sampled 500 rendered motion sequences and their corresponding text descriptions from each dataset. These samples were input into the Gemini-Pro model, which evaluated them using our predefined 1-5 point scoring criteria.
> >
> > The results showed that the average Gemini scores for text descriptions were 2.25 for the MotionX dataset and 3.08 for the HumanML3D dataset. Notably, the MotionBase dataset achieved a significantly higher average score of 3.82, demonstrating that its text descriptions align more effectively with the motion compared to the other datasets.
> >
> > **Q5. For the hierarchical text issue, is your evaluator trained with hierarchical text? Why is the R-P of generated results higher than GT?**
> >
> > Yes, our evaluator was also trained using hierarchical text, which consists of both "basic" text and "detailed" text. Regarding the observation that the R-P of generated results exceeds that of the ground truth (GT), **we obtain similar observations in several works like MoMask [1].** This discrepancy could be attributed to distribution differences between the training and testing sets of text-motion data. The evaluator, being a network trained on the training set, is inherently tailored to fit the training distribution. As a result, it may exhibit variance or bias errors when applied to the test set. If the generated text-motion data happens to align more closely with the training distribution, it can lead to evaluation metrics that surpass those of the GT test set. The quantitative evaluation of motion generation performance will be an interesting topic to explore.
> >
> > [1] MoMask: Generative Masked Modeling of 3D Human Motions
> >
> > **Q6. Why do your results of Hierarchical and Full Paramare not the same in response? I do not see the result of T2M-GPT as a baseline in the response.**
> >
> > Firstly, regarding the inconsistency between the results of full-parameter and hierarchical-parameter experiments, it is important to clarify that the comparison between the full-parameter and LoRA-parameter experiments was conducted using models trained solely on "basic" text descriptions.  As a result, these outcomes naturally differ from those of the hierarchical text experiments. However, the results from the full-parameter experiments are consistent with those obtained using "basic" text for training. To make this distinction clearer, we have included both sets of experimental results in Appendix C.5 and C.6 of the paper.
> >
> > Secondly, concerning the baseline results of T2M-GPT, we conducted additional experiments by training the T2M-GPT model on the MotionBase dataset and comparing it to a model based on GPT-2. As shown in the results, the T2M-GPT approach struggles to deliver competitive performance, further highlighting the critical role of pre-trained language models.  Compared to methods that use a frozen CLIP model as the text encoder followed by a decoder, motion generation models based on decoder-only pre-trained language models achieve significantly better results. These findings have also been included in Appendix C.7 of the paper.
> >
> > | Model | R@1 ↑ | R@3 ↑ | FID ↓ | MMDist ↓ |
> > |-------|-------|-------|-------|-----------|
> > | Real | 0.290 | 0.563 | 0.011 | 3.480 |
> > | T2M-GPT | 0.111 | 0.250 | 73.063 | 9.208 |
> > | GPT-2 | 0.264 | 0.542 | 0.516 | 4.007 |
> >
> > **Q7. I am still a bit curious about citing reference [1]. Which part of reference [1] did the authors refer to?**
> >
> > The citation of reference [1] serves to support the argument that recent research has extended instruction tuning to the multimodal domain, specifically when discussing Related Work Section of Large Language Models and Multi-modality in Section 2.1.
> >
> > Once again, we sincerely appreciate the time and effort you have dedicated to reviewing our work. We hope that our responses have effectively addressed your additional questions.

---

> > > ### Comment · Reviewer_WfZ5 · 2024-11-24
> > >
> > > Thanks for the response. Before checking your latest response, I still suggest you revise the website into supp.. There are so many motion generation papers submitted to ICLR/SIGGRAPH using the supp.. I don't know what your barrier is. You can concatenate them into a video demo. Is it very challenging? I would like to state that external links are not appropriate. This is because we cannot track external links without a third-party timestamp to make sure no new material is updated after my previous response to you. Besides, I am not sure whether my visit will be tracked by you. I hope the authors can understand my points. According to the ICLR review pipeline, I have the right to ignore these major changes. However, as I think the review process is to reduce concerns, I choose to check them. If these files on the website are changed, it might make my statement inconsistent with the material you provided. Thanks for your understanding.

---

> > > > ### Author Response · Authors · 2024-11-24
> > > >
> > > > Thank you for your feedback. We understand your concerns and want to clarify that we uploaded a supplementary file to OpenReview during the first-round review. Additionally, we provided a larger version of the demos via an anonymous Dropbox link: https://www.dropbox.com/scl/fo/6w7yuhun8mpuz9yuaz5ej/ALUo_dOLYIyNgOZ3Ll85JCI?rlkey=yecydfnno43b5o602ae2fmr0w&st=rq97kdcw&dl=0 , where the modification timestamp is manifest and your visit is not tracked.
> > > >
> > > > We sincerely hope these efforts adequately address your concerns, and we remain open to further feedback.

---

> ### Author Response · Authors · 2024-11-25
>
> Dear reviewer, we thank you once again for your valuable feedback, which has greatly helped improve the quality of our paper. As the rebuttal deadline approaches, we would like to confirm whether our responses have adequately addressed your concerns.  Specifically, we have:
>  - (1) explained the details of the motion-capturing details.
>  - (2) clarified that the dataset includes massive dynamic motion sequences, which we believe is sufficient large to substantiate our data contribution, while also demonstrating the benefits of static data via our ablation in Table 4 and Appendix C.1 (Tables 7 & 8).
>  - (3) uploaded a supp material to OpenReview.
>  - (4) provided a detailed explanation of the scoring process based on the Gemini-score.
>  - (5) introduce the usage of hierarchical text.
>  - (6) address why the results of Hierarchical and Full settings differ.
>  - (7) clarified the purpose of citing reference [1].
>
> All new additions and modifications are highlighted in blue in the manuscript for your easy reference.
>
> We would greatly appreciate your confirmation on whether these responses have fully resolved your concerns. If not, we welcome any additional feedback or further questions you may have.
>
> Best regards,
>
> The Authors

---

> > ### Comment · Reviewer_WfZ5 · 2024-11-26
> >
> > Thanks for the response from the authors. I acknowledge your efforts in this.
> >
> > - For PHC, do you use the default setting of the PHC? Any modification? Besides, the success rate of PHC is too low. Is there any reason for this? How do you deal with cases like "sit"?
> >
> > - I acknowledge that "static images contribute to video understanding or single-frame object bounding boxes aid video tracking". However, the general motion generation aims to fit a data distribution. Once you introduce the static ones, it will make your motion over-smoothing. Could you please clarify this? Besides, the setting of Table 4 is not clear as to why this can strongly support the claim.
> >
> > - Could you please concatenate the videos of motion and data in `example_zoo_of_dataset/` for checking jointly?
> >
> > - The video of sim2real, like `sim_real_demo/video_real/s0000.mp4`, shows the robot needs a residual force. Because there is a line hanging it in the sky, which is not physics plausible motion. This violates the claim of physics simulation. Besides, one of the main claims in PHC is jumping out of the residual force. This is not aligned with the previous usage of PHC. I am quite confused.
> >
> > - To independently assess the quality of the text description without any visual inputs, the LLM always exists hallucinations without any visual grouds. I suggest authors to improve the evaluation process.
> >
> > - The FID of T2M-GPT is amazingly high than others. Have you ever slcae the size of T2M-GPT? I have strong concern on this.
> >
> > I hope the concerns can be resolved.

---

> > > ### Comment · Reviewer_WfZ5 · 2024-11-26
> > >
> > > @pT95, what is your opinion on these? I hope you can join our discussion as our ratings are diverse.

---

> > > > ### Author Response · Authors · 2024-12-01
> > > >
> > > > Dear reviewers:
> > > >
> > > > We noticed that we haven't received your response to our latest responses, and we're eager to move forward with your feedback. Given the approaching deadline, would it be possible for you to provide your feedback at your earliest convenience? Do our responses answer your questions? We would be grateful for even brief comments. Thank you again for your expertise and consideration.
> > > >
> > > > Sincerely,
> > > >
> > > > Authors

---

> > > > > ### Comment · Reviewer_WfZ5 · 2024-12-02
> > > > >
> > > > > Thanks for the reply from the authors.
> > > > >
> > > > > I found that some of our discussions are not aligned. Thus I would like to specify.
> > > > >
> > > > > + For the PHC part, my main concern is about the retraining of the PHC. If you only used the pre-trained PHC from AMASS, how can it be generalized to the motion you captured? If you assume your data is shifted from AMASS, the tracked motions are not iid. How can you ensure the success rate? If you assume your data is not shifted from AMASS, the dataset diversity and contribution will be weakened. I hope the authors can specify this. If it is the best choice to apply this choice, I will treat it is as an ideal choice.
> > > > >
> > > > > + I do agree that the blog is not peer-reviewed, which cannot serve an important role in the discussion. The motivation I would like to refer to this is to clarify my points on the static data. I note that other reviewers have similar concerns. The authors stated their experiments show that static motion works. I ask for the ablation setting in the previous reply. However, I did not note the detailed setting. Is it trained with static data and the test set is not included with the static data? Did I miss something? Besides, I hope authors do not assume I do not know much about LLM/VLMs. Although I come from the animation community, I have several publications related to LLM/VLMs and even tuned related models by myself. Therefore, I am very clear about these technical details. The mentioned works are most related to VL understanding, which is mainly not related to motion generation. As a result, the evidence seems not convincing enough to me up to now.
> > > > >
> > > > > + For the updated supp., I do not know why it cannot be downloaded now. I will try it later.
> > > > >
> > > > > + I know the robot demos are used for showing the demo. However, the shown results seem to be a straightforward application, without additional technical innovation. If you emphasize this too much, reviewers will treat this as your unique contribution seriously. Otherwise, it will be questioned. I think a better choice is to include it in the appendix. If authors would like to claim the contribution to this, a better choice is to show the key scientific problem you resolved.
> > > > >
> > > > > + I think you missed my meaning regarding the Gemini evaluation. I asked the evaluation protocol previously and got the reply "Firstly, we independently assess the quality of the text description.". According to the word "independently", I treat the evaluation as text only. However, after checking the further response, I noticed that it has visual grounds. As a result, I clarified why I treated it as text only in the last reply. The authors asked me to revisit the response, I do not know what I missed.
> > > > >
> > > > > + Why does T2M-GPT struggle to match the performance of the GPT-2 architecture? The pretrained process of GPT-2 or the setting of CLIP?
> > > > >
> > > > > I do hope the discussion are aligned between authors and reivewers. If any points are not clear, please feel free to discuss more. Good luck!

---

> ### Author Response · Authors · 2024-11-26
>
> Dear Reviewer,
>
> Thank you for sharing your remaining questions. Before attaching our further responses, we are encouraged to see that the discussion is becoming more focused and specific with reduced questions, which suggests we are moving in the right direction!
>
> Here is our summary to the remaining five questions: (1) details issues about PHC; (2) discussion about static data; (3) video concatenation; (4) sim-to-real demo videos; (5) text hallucinations; (6) the FID of T2M-GPT. Hoping that  our understanding of the remaining questions is correct.
>
> **We will attach our responses very soon.** Thanks again for your constructive feedbacks!
>
> Best regards,
>
> The Authors

---

> > ### Author Response · Authors · 2024-11-26
> > **Official Comment by Authors**
> >
> > Dear reviewer, here are our responses to your remaining questions.
> >
> > 1. Our policy builds on the single primitive policy from PHC, with most configurations adhering closely to its design. For motions involving interaction like 'sit', we do not apply any pre-processing. Instead, we rely on the policy to perform directly tracking. Unsurprisingly, this often results in falling down to the ground at the very beginning, which is considered as a failure sample based on the previously defined criteria. This is probably the first reason why the success rate is relatively low. The second reason may stem from the fact that PHC is trained exclusively on the AMASS dataset. While AMASS is a high-quality dataset, its limited size constrains generalization. As the data scales up, the policy may struggle to track motions that fall outside the distribution it was trained on, further contributing to failures during tracking.
> >
> > 2. There are many common strategies to avoid the "over smoothing" you mentioned:
> >     1. During fine-tuning, we design specific prompts to guide the model's generation. For static poses, we include additional language prompts, such as 'keep the action still,' to help the model distinguish between generating a static pose and a dynamic motion.This approach leverages the well-known ability of large generation models based on LLMs [1,2,3,4,5] to produce diverse outputs through carefully crafted prompts. In addition, no negative effects are observed when combining images and videos, or bounding boxes and segmentation masks, or 3D skeleton poses and sequences, or vision and audios during training. We believe this principle should extend to motion as well.
> >     2. Visualization. To better prove our conclusion, in the visualization appendix, we present generation results from both models trained with and without additional language prompts. The clear differences demonstrate that **with additional language prompts, the model can distinguish between different motion distributions, thus not affecting the generation of dynamic motions.**
> >     3. Many common training strategies, like weighted sampling, weighted gradient, progressively increasing the ratio of dynamic motion, all these strategies can improve the stability of training, ensuring the model to avoid "over smoothing".
> >
> > 3. Sure, we have updated this video to the supp file on OpenReview
> >
> > 4. It's worth to note that we claim physical plausibility only within simulated environments, not in the real world.  For real robots, we demonstrate that the generated motion results can be successfully retargeted to the robot's joints, highlighting the potential applications for fields such as robotics. **We believe this has the potential to make a significant impact on the motion understanding community, extending to a wide range of areas.** A suspension system is required, as no one, to our knowledge, has yet been able to deploy physically plausible motions on real robots. Regarding your other question, our approach differs from using PHC to filter physically feasible motions. In our framework, PHC is used primarily during the data filtering stage to ensure the physical plausibility of training data. In the real robot demonstration, our focus is on accurate joint trajectory tracking.
> >
> > 5. Regarding text evaluation, it's important to note that **we did not solely rely on text-only scoring. As mentioned in our initial response**,we also rendered the motions into videos and input them, along with the text descriptions, into Gemini Pro, which has visual understanding capabilities. This approach helps mitigate the hallucination issue that can arise when text is used without visual grounding. The final evaluation results showed that our text annotations achieved higher motion matching accuracy.
> >
> > 6. The performance gap with T2M-GPT, we believe, is not due to differences in model size. The key factor lies in the text encoder used by T2M-GPT, which is a frozen CLIP text encoder. Due to the inherent limitations of CLIP's text encoding capabilities, models trained in this manner may struggle to comprehend a broader range of motion-related language, even if they demonstrate strong motion generation abilities. In contrast, decoder-only LLM-based large motion models, which jointly train text tokens and motion tokens, achieve both superior text-motion semantic alignment and enhanced motion generation capabilities.
> >
> > We are sincerely expecting your further feedback!
> >
> > Best regards,
> >
> > The Authors
> >
> > [1]Conditional Language Learning with Context. ICML 2024
> >
> > [2]Video-LaVIT: Unified Video-Language Pre-training with Decoupled Visual-Motional Tokenization. ICML 2024
> >
> > [3]Emu3: Next-Token Prediction is All You Need
> >
> > [4]VideoPoet: A Large Language Model for Zero-Shot Video Generation. ICML 2024
> >
> > [5]Unified-IO 2: Scaling Autoregressive Multimodal Models with Vision, Language, Audio, and Action. CVPR 2024

---

> > > ### Comment · Reviewer_WfZ5 · 2024-11-27
> > >
> > > Thanks for your answer. After checking your response, I have some concerns that I am not clear about currently.
> > >
> > > - For PHC, why is your design choice of primitives single and not scales the numbers? Besides, if you directly use the PHC trained on AMASS, the model you use to track in-the-wild motions is quite hard. Why not train a new PHC on your dataset? Besides, I still think the success rate is low, which might result from the inappropriate choice.
> > >
> > > - I am a bit sorry that I still cannot understand why static motions enhance the quality. If you read Daniel's blog, you will find that the provided result is not self-standing. These motions will introduce noise into the dataset.
> > >
> > > - Although you concatenate the motions and the original videos, the resolution is still low. Thus, I cannot see the fine-grained motions.
> > >
> > > - For the claim of applying motion on the robot, I am not clear what the core scientific problem you resolve here. From your response, the efforts seem to be engineering.
> > >
> > > - You stated "Firstly, we independently assess the quality of the text description.". I identify the evlauation is text-only according to "independently".
> > >
> > > - Is there any evidence to suppor that a frozen CLIP is not effective? The CLIP mainly contribute the T-M alignment, which is not highly related to the quality itself. I am currently not convinced by this clarification.
> > >
> > > I hope these issues can be resolved. If anything is not objective, you can directly point it out, becauce the review process mainly focuses on the reserach quality itself.
> > >
> > > Good luck!

---

> > > > ### Author Response · Authors · 2024-12-01
> > > >
> > > > Dear reviewer,
> > > >
> > > > With only about 2 days remaining until the rebuttal deadline, we are still eagerly awaiting your response. We sincerely hope that when you have a moment, you could spare a few minutes to check the summary and reply above. Have your previous questions and concerns been addressed? We are very keen to know whether our rebuttal has changed your recommendation regarding our work.
> > > >
> > > > Sincerely,
> > > >
> > > > Authors

---

> ### Author Response · Authors · 2024-11-28
>
> Thank you for your thoughtful reply. While we deeply appreciate your feedback, we would like to respectfully address some points of disagreement and provide clarification.
>
> 1. We believe there may be a misunderstanding in your question and suggestion. **How could we use to-be-refine data (our data) to train PHC for data to be refined (our data), akin to the idea of lifting oneself with their own hands.** It is important to note that no large-scale benchmark can guarantee 100% clean data. Therefore, even with a successful conversion rate of approximately one-third, this demonstrates the effectiveness of our data processing strategy with at least 30% additional improved motion with high quality.
>
> 2. We respectfully note that citing an informal blog as valid evidence in a professional rebuttal may not be appropriate. Additionally, we respectfully disagree with the blog's conclusion as it has not been validated under the context of large-scale generative pretraining. **Considering this, we believe the use of static data are still an open question instead of a conclusion.** We kindly recommend that the reviewer refer to newest well-recognized generative works in the field, such as **UnifiedIO-2, VideoPoet, EMU3**, and multimodal tuning methods like **LLaVA, Qwen-2-vl**, to gain a deeper understanding of the capabilities of current generative models. If you have already read these works, you might observe that concerns about "over-smoothing" are not self-standing when considering large-scale pretraining, well-designed prompts, and effective training strategies. As we understand the reviewer might not specialize in the area of LMM, we have provided a more detailed explanation below to offer further clarity.
>
>     1. From a statistical learning perspective, directly combining data from different distributions without incorporating conditional information may result in learning a biased marginal distribution, which could potentially degrade generation quality. However, by introducing additional conditional information—in our case, specific language prompts—we guide the model to learn the conditional distribution P(motion|text, condition) rather than the marginal distribution P(motion). This allows the model to accurately switch between different distribution modes during generation, effectively avoiding distribution confusion. Moreover, static data can be seen as a special case of dynamic data, functioning as a "snapshot" of motion at a particular moment.
>
>     2. More importantly, static data contributes to improved performance by helping the model better understand spatial constraints between joints. Static poses provide abundant examples of valid joint positions and rotations, enabling the model to learn what constitutes physically plausible human postures. Furthermore, static data enhances the model’s ability to capture correlations between joints, such as motion range limitations and inter-joint dependencies. With the guidance of conditional prompts, the model can not only distinguish between static and dynamic data but also leverage the foundational knowledge learned from static data to enhance dynamic motion generation, thereby improving overall quality.
>
>     3. We carry out experimental results to validate the above points: when specific conditional prompts are added, the model effectively distinguishes between data distributions and generates outputs accordingly.
>
> 3. We provided a new video with higher resolution to the supp.
>
> 4. The robot videos are intended as demos to showcase the potential applications of our work.
>
> 5. We kindly request the reviewer to revisit our response, as we have addressed your question. Key points have been clearly marked with significant symbols, such as (1.; 2.), **directly following the sentence you referenced.** We hope this format makes it easier to locate our responses.

---

> > ### Author Response · Authors · 2024-11-30
> >
> > Dear reviewer,
> >
> > Thank you again for your valuable review. We have responded to your every question and concern. We hope to hear back from you! If you have any unresolved concerns, please feel free to let us know! Or if our responses have addressed your questions, we would be grateful if you could consider adjusting your score accordingly.
> >
> > Best,
> >
> > Authors

---

> ### Author Response · Authors · 2024-11-28
>
> 6. We understand that you may have a different interpretation of the experimental results. However, we respectfully suggest that this difference in interpretation should not overshadow the significance of our contributions. We retrain a T2M-GPT model with a parameter count comparable to GPT-2 medium on MotionBase. Nevertheless, the results indicate that it struggles to match the performance of the GPT-2 architecture. We believe that large motion models based on decoder-only LLMs, which jointly train text tokens and motion tokens, achieve better text-motion semantic alignment and stronger motion generation capabilities.
>
> | Method | #Param. | R@1 ↑ | R@3 ↑ | FID ↓ | MMDist ↓ |
> |--------|---------|--------|--------|---------|-----------|
> | Real | - | 0.290 | 0.563 | 0.011 | 3.480 |
> | T2M-GPT | 380M | 0.243 | 0.504 | 1.909 | 4.593 |
> | GPT-2 Medium | 355M | 0.264 | 0.542 | 0.516 | 4.007 |

---

> > ### Author Response · Authors · 2024-11-29
> >
> > Dear reviewer,
> >
> > As we near the conclusion of the discussion phase, we would like to inquire if our response has effectively addressed your inquiries. In the rebuttal, we have provided explanations for your questions and concerns. Should our explanations have met your expectations, we kindly request your consideration in revising the score accordingly.
> >
> > Should you have any additional comments or concerns, we are more than willing to address them promptly during the response period. We deeply appreciate your constructive and insightful feedback, which has greatly contributed to the refinement of our work.
> >
> > Best regards,
> >
> > Authors

---

### Author Response · Authors · 2024-12-02

**Dear reviewer WfZ5**,

Thank you for your unique and valuable feedback. Inspired by you, we believe it would be beneficial to include our response in the top general response, allowing other reviewers and AC to easily access it due to the lengthy reply history. After this extensive journey of rebuttal, we sincerely hope to reach a final conclusion.

1. To answer your first question. You mention "how can it be generalized to the motion you captured?". However, we do not claim a generalizable PHC with 100% success rate from the beginning. Instead, our goal is to present PHC as an additional technique compared to MotionX [1] and Holistic-Motion2D [2], which also rely on pretrained models to extract motions from videos without any physical-plausible refinement. Compared with them, our approach achieves 30% more physically plausible motions, effectively reducing jittering and sliding artifacts. **We believe this makes our dataset a self-stand contribution, otherwise all other works using pretrained models to extract motion without physical-plausible refinement, would become of no value.** Considering this, we argue that an ideal method you are expecting is another large topic [3] which obviously falls outside the scope of this paper.
2. The details of the static data ablation can be clearly found in L435–L438 and L1049–L1059, with the marked title **"TRAIN SET w/o static data"** clearly indicating the relevant settings. The test sets used are MotionBase-test and MotionBase-test w/o static/syn, as described in L1055. In both test sets, using the complete MotionBase dataset significantly outperforms the "w/o static data" configuration. I do not know why you can not see it. We also note that no other reviewers, apart from you, have raised similar questions, suggesting that the provided setting details should already be sufficiently clear.

In addition to this, for the static pose:
  - We show its effectiveness through our experiments in Table.4 and Table.8
  - We emphasize once again that static data serves as a bonus to our dataset contribution. MotionBase already includes 800K dynamic motion samples—10 times more than previously available datasets.
  - We note that the reviewer has overlooked several important references despite claiming being familiar with motion and LMM community. To assist with the reviewer's evaluation, we kindly provide some references for better judgment. Reference [4], for example, has already demonstrated that static data can smoothly produce whole-body and hand poses in a video. Many well-known works in motion and keypoint [5][6][7] heavily rely on well-pretrained backbones using static pose (e.g., COCO) or directly incorporate static pose during training. **These works consistently validate that static pose contributes positively to smooth generation and poses no harm**. We strongly recommend the reviewer consult these officially published and well-known works and their repositories, rather than relying on unofficial blogs as evidence.  As a specialist of LMM, we believe the reviewer can readily appreciate the paradigm of using static and dynamic data for pretraining and dynamic data for fine-tuning.
  - We kindly point out that the reviewer’s statement is not entirely accurate. No other reviewers except you have raised further concerns about static data following our earlier responses.
3. We believe the supplementary material is accessible, as we successfully downloaded the file from OpenReview using various IP addresses across different locations worldwide. All attempts were successful, except for your instance.
4. We are so puzzled by this feedback, as the robot demo was only included in the supp file and was not presented as a contribution in the main paper ever. **It is concerning when a reviewer misrepresents our stated contributions by suggesting**: "I think a better choice is to include it in the appendix" or suggesting, or "If you emphasize this too much, reviewers will treat this as your unique contribution seriously." **These words definitely go against our statement, which risk misleading the AC or other reviewers, potentially creating misunderstandings about our work.** In fact, we provided the demo to all reviewers, and none of them raised similar concerns.
5. **We are quite puzzled that the reviewer has focused on the word "independently" while overlooking the initial word, "Firstly."** As clearly stated, we have two approaches for evaluation: one is text-only, and the other is vision-text. "Firstly" refers to the first approach, while "More important" introduces the second. In our response (22 Nov 2024, 21:03), we had already clarified this by using clearer numerical markers (1..., 2...) to enhance readability. Notably, this explanation was also shared with reviewers qpUP and pt95, neither of whom expressed any misunderstanding.

---

> ### Author Response · Authors · 2024-12-02
>
> 6. For T2M-GPT, we follow the official training configuration, using the text embedding from the pretrained CLIP ViT-B/32 version, and for GPT-2, we similarly use the official pre-trained model [8]. The experimental results show inferior performance, which we believe is self-evident. T2M-GPT uses a frozen CLIP text encoder, resulting in fixed text representations. In contrast, GPT-2 enables joint training of text and motion tokens, allowing text representations to be optimized along with the task, thus achieving better text-motion semantic alignment.
>
> Sincerely, we hope the discussion can be based on the officially published literature and experimental results in our paper, rather than unofficial blogs. **We also provide our code and checkpoints from the beginning. We are more than happy you can run it by yourself if you are not convinced by our shown results.**
>
> If you have further questions, feel free to tell us, or if, our responses successfully address your concern, would you mind reconsidering your review based on these more reliable references?
>
> Sincerely,
>
> Authors
>
> [1] MotionX: A Large-scale 3D Expressive Whole-body Human Motion Dataset, CVPR 24
>
> [2] Holistic-Motion2D: Scalable Whole-body Human Motion Generation in 2D Space
>
> [3] Morph: A Motion-free Physics Optimization Framework for Human Motion Generation, 2024
>
> [4] https://github.com/IDEA-Research/DINO-X-API
>
> [5] https://github.com/ViTAE-Transformer/ViTPose, 1.4K star
>
> [6] https://github.com/caizhongang/SMPLer-X, 1K star
>
> [7] Reconstructing World-grounded Humans with Accurate 3D Motion, CVPR2024
>
> [8] https://huggingface.co/openai-community/gpt2-medium

---

> > ### Comment · Reviewer_qpUP · 2024-12-02
> > **Official Comment by Reviewer qpUP**
> >
> > I also have concerns regarding the static data used in the study. References [4][5][6][7] all focus on detection algorithms, while in the context of data generation, the distribution of the data itself plays a critical role. To illustrate, consider the case of pose data taken from the middle frames of a “running” sequence. The corresponding caption might be something like "A man is running" or a similar description. This type of caption is very close to the actual running motion data. When sampling, it could influence the generation process. While I agree that joint training on pose and motion data is beneficial, repeatedly using the same pose 64 times is problematic and not ideal.
> >
> > Additionally, several important aspects of dataset construction were either not addressed or mentioned in the original submission, such as policy training, retargeting, and dataset evaluation. These details, I believe, are crucial and should not be overlooked. However, all of these details are supplemented during the rebuttal period. This is unfair to other submissions.
> >
> > I acknowledge and appreciate the authors' efforts, but I feel the paper is not fully prepared for the ICLR. If the authors can include these additional details and resubmit, I believe the work would be much stronger and would be a good paper.
> >
> > Reviewer qpUP

---

> > > ### Author Response · Authors · 2024-12-02
> > >
> > > Dear reviewer,
> > >
> > > It would be unfair to disregard all the details we provided during the ICLR rebuttal, especially after we took the time to respond thoroughly to your questions. ICLR allows authors to revise their papers during the rebuttal period, which is different from other conferences like CVPR and NeurIPS. Most importantly, the provided details are provided to address your questions, which does not go against our contribution in the submitted version. **Here, we really need @AC to clarify this.**
> > >
> > > **If the reviewer only refers to the original submission, then what is the purpose of the rebuttal process? Why ask for all these details, instead of directly telling us that you would not change your score at the beginning?**
> > >
> > > Regarding the static data, we have mentioned several times that we provided 800K dynamic data, which takes a larger proportion of the dataset (56%). Wouldn’t it be inappropriate to overlook this significant portion and focus solely on the static data (we have also verified the benefit of static data experimentally)?  We have provided experimental results and the implemented code. While we understand the reviewers’ concerns, if they are unconvinced by our experimental results, wouldn't it be more appropriate for the reviewers to run the code themselves? If the reviewer overlooks the results and code we provided and relies on subjective experiences alone to draw conclusions, how can we convince a reviewer?
> > >
> > > Best,
> > >
> > > Authors

---

> ### Author Response · Authors · 2024-12-02
>
> Dear reviewer,
>
> We have further questions: Based on your feedback, do you believe the paper would be improved by removing all of our static data and corresponding experimental results? Doesn’t this suggestion seem somewhat unreasonable? Additionally, do you recognize the contribution of the dynamic part of MotionBase? **If you believe the static data is more problematic, then the dynamic data should be of higher quality to compensate, right?** Otherwise, how would the results improve?
>
> Best,
>
> Authors

---

> > ### Comment · Reviewer_WfZ5 · 2024-12-03
> >
> > It seems that the authors' tone is quite unkind. Please stay calm.
> >
> > For the PHC part, the pre-trained PHC is trained on AMASS, not your dataset. This will give you a low success rate for your tracking. It is also a choice to train a new PHC training on your dataset, which can promise generalization ability.
> >
> > I notice that `qpUP` has a similar concern on the static motion, which is not an animation but poses. I think this is not a suggested choice in the animation community.
> >
> > Besides, the MotionBase dataset construction pipeline introduced in Section 3 does not include any introduction of PHC, which plays an important role in the annotation process.
> >
> > According to the review guidance, the major revisions are not worthy of consideration in the review process. However, I tried to improve the quality of the submission with the authors and to provide my feedback on this, which encouraged me to discuss with the authors more about the unclear details.
> >
> > The authors have made major revisions to the dataset. The authors' response:
> > > As of the start of the rebuttal period, we have collected over 1.5 million motion trajectories and 4 million motion-text pairs, **which are 1.5X compared to our first submission.**
> >
> > If no special notes, all contents related to PHC are in the Appendix and highlighted as revision color, which can be recognized as a major revision on the technical pipeline.
> >
> > **The discussion process is for addressing misunderstandings in the reviews, not for providing major revisions. Up to now, the authors have made major revisions to the original submission, on the dataset and the annotation pipeline. Besides, the details of these parts are not clearly clarified during the discussion.**
> >
> > Reviewer WfZ5

---

### Author Response · Authors · 2024-12-03

Dear reviewer WfZ5,

This is the reviewer guidance on ICLR 2025 official website. We sincerely note that ICLR's rebuttal phase is different from most other AL/ML comminuty: https://iclr.cc/Conferences/2025/ReviewerGuide#Reviewer%20tasks. If you are the first time to serve as ICLR's reviewer, you should really  read the guideline carefully first. Otherwise, your conclusion may disobey the traidition and spirit of our ICLR community.

**Engage in discussion: The discussion phase at ICLR is different from most conferences in the AI/ML community. During this phase, reviewers, authors and area chairs engage in asynchronous discussion and authors are allowed to revise their submissions to address concerns that arise. It is crucial that you are actively engaged during this phase. Maintain a spirit of openness to changing your initial recommendation (either to a more positive or more negative) rating.**

In what context does this paragraph suggest that the reviewer should base their evaluation on the "initial paper"?

We are quite puzzled where do you get the information that "a reviewer's decision should base on the original version". As a ICLR reviewer too, I saw many reviewers appreciate the revision of the author and change. If the reviewers do not follow the ICLR's spirit, it would be quite unfair for our paper's rebuttal.

In addition to rebuttal guidance, we have the following concerns that we believe should be clarified first:

  - For PHC and other details: the reason we provide so much details about PHC is because you ask the corresponding questions. The PHC's role is only a sentence in our initial paper, not a major part. **Once again. we concern the reviewer mistakenly represent our contribution.**
  - For static data: Do we have any approach to convince you in addition to experimental results and codes we provided?
  - Most importantly, we raise the **logical questions**: Based on your feedback, do you believe the paper would be improved by removing all of our static data and corresponding experimental results? Doesn’t this suggestion seem somewhat unreasonable? Additionally, do you recognize the contribution of the dynamic part of MotionBase? If you believe the static data is more problematic, then the dynamic data should be of higher quality to compensate, right? Otherwise, how would the results improve?


Finally, we believe it’s not productive to judge the tone of others. This could serve as a subjective hint to the AC and other reviewers. If any of our words stray from the matter at hand, please point it out. Otherwise, it risks being an unfounded accusation.

Best,

Authors

---

### Author Response · Authors · 2024-12-03

Dear AC/PAC/PC,

For the long time during ICLR's history, we believe ICLR allows authors to revise their papers in response to reviewers' concerns. However, we are now concerned about potential misunderstandings regarding the rebuttal phase, as some reviewers may be confusing ICLR’s rebuttal process with that of traditional conferences. **The spirit of ICLR is the only reason we have been willing to provide such detailed responses before the official release of our paper (over 30 ICLR pages in rebuttal), especially given that we initially submitted a content-rich version (over 25 ICLR pages).**

Some reviewers have suggested that "major revisions are not worthy of consideration in the review process" and that "basing a decision on revisions would be unfair to other papers." We would appreciate clarification from the AC, PAC, PA on this matter, as we’ve noticed that one of the main reasons reviewers have not raised their scores is their belief that the decision should be based solely on the original version of the paper.

For reference, here is the reviewer’s guideline on the official website:

**Engage in discussion: The discussion phase at ICLR is different from most conferences in the AI/ML community. During this phase, reviewers, authors and area chairs engage in asynchronous discussion and authors are allowed to revise their submissions to address concerns that arise. It is crucial that you are actively engaged during this phase. Maintain a spirit of openness to changing your initial recommendation (either to a more positive or more negative) rating.**

This raise another question we wonder: if the reviewer suggest "the decision should base on initial version", why do they ask so many questions refering to our data construction details (65 reply so far)? The level of detail we have provided has become so extensive that reviewers might mistakenly consider it the paper's main contribution. To our knowledge, no other works (e.g., MotionX, Smpler-X, EgoBody) have released such extensive details, even after their official publication.

In addition to this, another concern is about the static data. We have provide experimental results to validate our proposal. If the reviewer does not believe the results, we also provide corresponding code, checkpoints. If the reviewer overlooks the results and code we provided and relies on subjective experiences alone to draw conclusions, how can we convince a reviewer?

Best regards,

Authors

---

### Author Response · Authors · 2024-12-03

Dear reviewers,

We sincerely appreciate the time and effort each reviewer has dedicated to this discussion. However, with the rebuttal deadline approaching, there are still some remaining questions that we believe should be addressed.

Before raising these questions, we want to clarify an important point: Why do some reviewers perceive the level of detail we provide as resembling a major revision?  **In fact, we are also puzzled by this. We suggest after several rounds of rebuttal, some reviewers may have mistaken minor aspects of our work for major contributions.**
For example, reviewer WfZ5 thinks we emphasize too much about robot demos. Here is a brief rebuttal summary of this point (**if anything of the following is wrong, please feel free to point it out**):
- Q: We need a recorded video.
- A: Sure, we made a website for you.
- Q: The external link is not allowed, you should provide a supp file to OpenReivew
- A:  OK, we provide the zipfile that contains all demos in the website.
- Q:  Is it challenging for you to concatenate them in a video?
- A:  Sure, we concatenate our demos and update the supp file once again.
- Q:  The resolution is too small, details can not be seen.
- A:   OK, we provide a higher-resolution video for you.
- Q:  Why do you hang the robot in your robot demo?
- A:  The robot videos are only demos.
- Q:  The author should not emphasize too much about the demo. If you emphasize this too much, reviewers will treat this as your unique contribution seriously.

After addressing the reviewer's questions **about demos**, we believe they may have received an overwhelming amount of information from us, potentially leading to the mistaken impression that the robot demo is a significant aspect of our work. **In fact, we do not even discuss it in the main paper.**  This misunderstanding occurred multiple times during the rebuttal process. For example, PHC is a technique proposed by others, and we use it only as one of the methods to refine our data, which is briefly mentioned in just 1–2 sentences in our paper, just like most previous works (e.g., HumanPlus, MotionX). However, during the rebuttal, we provided two pages of detailed explanations about PHC. As a result, the reviewer mistakenly regarded it as a central aspect of MotionBase, **which we have never claimed.**

Finally, we want to go back to our remaining questions, which we believe are important to draw a conclusion:
- Does reviewer suggest the ICLR's rating should at least consider the revised paper according to the spirit of ICLR?
- Do you believe our experimental results, or our code, checkpoints? If you do, why do you still have concerns about static data? If you do not, what else do we need to provide? Is there any possibility to convince you?
- The original dynamic data is near 400K, almost 5 times larger than before. The revised version is 800K, 10 times larger than before. If the reviewer is not convinced by static data, do you recognize the contribution of our MotionBase?
- If the reviewer thinks static data is harmful, do you believe that our dynamic data should have higher quality? This should be a conclusion that can be reached simply through logic.

Best,

Authors

---

### Author Response · Authors · 2024-12-04
**Summary of the rebuttal**

Dear AC and reviewers:

We deeply appreciate your efforts during the rebuttal process. We are encouraged that reviewers CH9D, phNv, and pT95 recognize the value of our work. Reviewer qpUP believe our paper would be a good paper with the inclusion of revised details during rebuttal, though their review, along with WfZ5's, is borderline reject. To address our lengthy and fragmented discussion with WfZ5, we provide a brief summary below.

---
# Our work in brief
We propose to construct a large-scale motion generation dataset, validate the scaling law in motion understanding, and explore motion tokenization. The importance of contributions is ranked in order.

---
# Reviewers' positive recognitions
* **From reviewer CH9D**:
  - well motivated
  - dataset is large
  - Evaluation on multiple datasets and multiple models with strong baselines.
* **From reviewer phNv**:
  - propose a large-scale dataset
  - identifies key factors influencing the effectiveness of large motion models
  - explore effective motion quantization method

* **From reviewer pT95**:
  - the proposed large-scale data has the potential to be a valuable resource for the community

* **From reviewer qpUP**:
  - collect a new large-scale text-motion dataset.
  - scale the tokenizer vocab size and the model size.

* **From reviewer WfZ5**:
  - good writting
  - The statistics of the dataset are clear.

# Remaining points:

**Should the revised details be considered during rebuttal?**

We emphasize that the spirit of ICLR motivates us to share extensive details of our work, including code, checkpoints, and demo —— something few other papers do before official publication. The unique spirit of ICLR, which fosters open discussion, is what sets it apart from other conferences. Since some reviewers may be attending for the first time, we hope the AC, Senior AC, and PC will remind them of this. If reviewers insist on statements like:
> "the discussion process is for addressing misunderstandings in the reviews, not for providing major revisions. "

>  "basing a decision on revisions would be unfair to other papers"

>  "These details are supplemented during the rebuttal period. This is unfair to other submissions."

  Our efforts to share will be in vain. We feel discouraged and disheartened by these comments, especially after receiving appreciation in previous ICLR submissions. We are even restricted from using anonymous links to share our code, checkpoints, and demos. As reviewers ourselves, we’ve seen other papers praised for their revisions. If our revision is disregarded, it would be unfair. For reference, here’s the reviewer guideline from the official website:

  > "Engage in discussion: The discussion phase at ICLR is different from most conferences in the AI/ML community. During this phase, reviewers, authors and area chairs engage in asynchronous discussion and authors are allowed to revise their submissions to address concerns that arise. It is crucial that you are actively engaged during this phase. Maintain a spirit of openness to changing your initial recommendation (either to a more positive or more negative) rating."

**WfZ5 considers some points as major revisions, but we disagree.**

The details during rebuttal do not go against our original paper. For example:
  - WfZ5 labels PHC as a significant revision, but we never claimed that; it's just one step in a broader data refinement process.
  - WfZ5 argues we overemphasize the robot demo, which we did not claim.
  - WfZ5 views increasing dynamic data from 400K to 800K as a major revision, but we question this; the original 400K clips with over 1M motion-text pairs are already  self-standing.
  - A major revision should be defined as something goes against our contribution to the original paper. Asking eight questions about a small step in the pipeline, such as implementation details, reward setup, or success rates, cannot be considered a major revision.

**Is Static data harmful?**
- We provide experiments, code, pretrained models, 100+ demos, and related work, yet the reviewers remain unconvinced. Additionally, we find some feedback from WfZ5 to be subjective:
    - WfZ5 states, "I think this is not a suggested choice in the animation community," which is a biased conclusion. How can motion generation be limited to animation, ignoring applications in vision, robotics, and VR/AR?
    - WfZ5 recommends reading an unofficial blog by Daniel Holden, claiming "you will find it not self-stand if you read this well-known scientist's blog." After reading it, we found only empirical suggestions. Using this as evidence to question our results contradicts scientific principles, which should be based on truth and experimental evidence, not empirical opinions.
- This opinion leads to a logical contradiction: if static data is harmful, dynamic data should be of even higher quality. Otherwise, how could our results improve?

---

> ### Author Response · Authors · 2024-12-04
> **Rebuttal Summary with WfZ5**
>
> # Rebuttal Summary with WfZ5
>
> ---
>
> Given the lengthy and fragmented rebuttal history with WfZ5, we provide a brief summary to facilitate reading.  We hope the AC, Senior AC, and PC will carefully review this summary before making a final decision. In total, WfZ5 raised 51 questions, including 40 detailed ones and 11 of less importance.
>
> ---
>
> ## **2-round questions of "External links are not allowed"**
> - **Q1**: External links are not permitted as they cannot be tracked fairly. Why not use the OPenreview's Supp?
> - **A1**: OpenReview's 100MB file size limit prevents us from including all materials in Supp. We also confirmed ICLR guidelines do not forbid anonymous links.
> - **Q2**: So many motion generation papers submitted to ICLR/SIGGRAPH using Supp. I don't know what your barrier is. You can concatenate them into a video demo. Is it very challenging? External links are not appropriate.
> - **A2**: This is the first time we’ve been informed that external links for codes and resources are not only discouraged but permitted, despite their common use in ICLR 2025 submissions.
> - **Q3**: NO FURTHER REPLY
>
> ---
>
> ## **8-round questions about demos.**
> - **Q1**: The work should include demo videos.
> - **A1**: We provide an anonymous website and cloud link.
> - **Q2**: The videos on the website are not user-friendly. Please add them to the Supp.
> - **A2**: Our videos exceed the 100MB limit, so we offer an anonymous cloud link for easy access.
> - **Q3**: Many motion papers use the Supp. Why can’t you concatenate them into a single video?
> - **A3**:  We provided a zip file with all demos to the Supp.
> - **Q4**:  Could you concatenate the videos?
> - **A4**:  We concatenated the demos and updated the supp file.
> - **Q5**:  The resolution is too low; I can’t see fine-grained motions.
> - **A5**:  We provided higher-resolution videos.
> - **Q6**:  I do not know why it cannot be downloaded now. I will try it later.
> - **A6**:  We verified successful downloads worldwide via OpenReview.
> - **Q7**:  I am not clear what the core scientific problem you resolve for robot demos.
> - **A7**:  The robot videos demonstrate potential applications, not a core contribution.
> - **Q8**: The demos seem straightforward with no technical innovation. If you emphasize them, reviewers might treat them as a major contribution. It’s better to include them in the appendix or show the scientific problem solved.
> - **A8**: The robot demos are in the supp file only. We never claimed them as a core contribution. These comments contradict our statements.
> - **Q9**: NO FURTHER REPLY
>
> ---
>
> ## **6-round Questions about "Text annotations"**
> - **Q1**: Has hierarchical text been used in model training?
> - **A1**: Yes, hierarchical text was used during pretraining, and relevant experiments were provided.
> - **Q2**: Has the text data been verified?
> - **A2**: Yes, using two methods: (1) a text-only method and (2) a vision-text method.
> - **Q3**: How can a text-only method fairly evaluate alignment with motion?
> - **A3**: As stated in A2, we have two evaluation methods: Firstly, we independently use text. More importantly, we use both vision and text.
> - **Q4**: LLMs hallucinate without visual input. Should the method be improved?
> - **A4**: We don’t rely solely on text-only scoring. Both text-only and vision-text methods are used, as explained in A1 and A2.
> - **Q5**: You stated "Firstly, we independently use...". I identify the evaluation is text-only according to "independently".
> - **A5**: We clearly outlined two methods using "1... 2..." in A1 and "Firstly... More importantly..." in A2.
> - **Q6**: I asked the evaluation protocol previously and got the reply "Firstly, we independently assess the quality of the text description.". According to the word "independently", I treat the evaluation as text only. However, after checking the further response, I noticed that it has visual grounds. As a result, I clarified why I treated it as text only in the last reply. The authors asked me to revisit the response, I do not know what I missed.
> - **A6**: We don't know why the reviewer only focused on the word "independently" while overlooking the initial word, "Firstly."
> - **Q7**: NO FURTHER REPLY

---

> ### Author Response · Authors · 2024-12-04
>
> ## **7-Round Questions about "Static data"**
> - **Q1**: How is motion quality ensured? I suggest authors read **Daniel Holden's blog**, who is a well-known graphics scientist.
> - **A1**: We use a 3-step method to ensure motion quality.
> - **Q2**: Our animation community  considers static data harmful. Refer to **Daniel's blog**.
> - **A2**: (1) Our dataset contains 400K motions in the initial version, 800K in the latest version, 5 times and 10 times larger than the previous ones, respectively. (2) Results on Table 4 show effectiveness of static data. (3) Dynamic motion tuning ensures high-quality generation.
> - **Q3**: Once you introduce the static ones, it will make your motion over-smoothing. Could you please clarify this?
> - **A3**: Strategies like (1) prompt design, (2) weighted sampling, and (3) progressive training help prevent over-smoothing. We provided cases to show the effectiveness of prompt design to avoid "over-smoothing" in Supp.
> - **Q4**: I still cannot understand why static motions enhance the quality. **If you read Daniel's blog**, you will find that the provided result is not self-standing. These motions will introduce noise into the dataset.
> - **A4**: Informal blogs are not valid evidence. Static data is an open question, not a conclusion. Ablations comparing training "with" and "without static data" demonstrate its effectiveness.
> - **Q5**: I asked for ablation settings but missed the details.
> - **A5**: Details are clearly stated in L435–L438 and L1049–L1059 under "**TRAIN SET w/o static data.**"
> - **Q6**: I think this is not a suggested choice to use static data in the animation community.
> - **A6**: We’ve provided experiments, codes, and checkpoints proving effectiveness. Subjective conclusions alone are unconvincing.  Besides, our datasets contain 400K motions in the original version and 800K in the latest one.
> - **Q7**:Expanding from 400K to 800K is a major revision and is not worthy of being considered.
> - **A7**: Based on your feedback, do you believe the paper would be improved by removing all static data, experiments and discussion? Doesn’t this suggestion seem somewhat unreasonable?
> - **A7**: Do you recognize the contribution of our dynamic data, no matter 400K or 800K?
> - **A7**: There is a logical contradiction in reviewer's opinion: The more harmful you believe static data is, the higher-quality dynamic data should be. Otherwise, how would our results improve?
> - **Q8**: NO FURTHER REPLY
>
> ---
>
> ## **8-round questions about "Data Quality and PHC"**
> - **Q1**: The data contribution is limited. The videos comes from InternViD and WebVid.
> - **A1**: We don’t use raw videos directly. Instead, we process them through a rigorous framework: (1) 2-step video selection, (2) 4-step boundary detection, (3) 3-step occlusion removal, and (4) 2-step single-frame processing. (5) ...
> - **Q2**: How is motion quality ensured?
> - **A2**: (1) An RL policy refines motions to be physically plausible; (2) empirical quality assessment; (3) a pretrained motion model for further refinement.
> - **Q3**: What RL policy do you use? PHC? RFC? ASE? What is the duration and success rate?
> - **A3**: We use PHC: 50% of the original duration, 51.4% success rate.
> - **Q4**: Why is PHC’s success rate so low, and how do you handle cases like "sit"?
> - **A4**: Low rates stem from (1) interaction motions like "sit" failing and (2) PHC being trained only on AMASS, limiting generalization.
> - **Q5**: Why not train a new PHC on your dataset?
> - **A5**: It’s illogical to use to-be-refined data to train PHC for refining that data to-be-refined.
> - **Q6**: If PHC doesn’t generalize well, how can success be ensured?
> - **A6**: We never claim 100% success. PHC refines 30% of the data, which is still an improvement over prior works with 0%.
> - **Q7**:  **(Repeat of Q5)** Why not train a new PHC on your dataset?
> - **A7**: Same as A5.
> - **Q8**: PHC is highlighted as a revision and should be seen as a major change.
> - **A8**: PHC is a minor step in data refinement, itself a small part of the pipeline. It’s not a core contribution of our paper, and implementation details are not central to our work. Highlighting it for clarity doesn’t make it a major revision.
> - **Q9**: NO FURTHER REPLY

---

> ### Author Response · Authors · 2024-12-04
>
> ## **6-round questions about "LLM Motivation and T2M-GPT Comparison"**
> - **Q1**: Hierarchical text contribution isn’t well discussed. Has it been used in training? What’s the motivation?
> - **A1**: Yes, hierarchical text is used in pretraining. It improves (1) text corpus diversity, (2) finer-grained descriptions, and (3) performance over basic text.
> - **Q2**: The motivation for introducing LLM is unclear. Why no T2M-GPT baseline?  Are pre-trained parameters and fine-tuning (e.g., LoRA) used?
> - **A2**: LLMs enhance language-motion understanding. Training: (1) pre-trained parameters, (2) full fine-tuning (LoRA struggles with motion tokens). HumanML3D results outperform T2M-GPT baseline.
> - **Q3**: Why differ between hierarchical and full-param results? I do not see the result of T2M-GPT as a baseline.
> - **A3**: Full-param used only basic text; hierarchical used both basic and detailed text. T2M-GPT results on MotionBase are in Appendix C.7, showing lower performance than ours.
> - **Q4**: T2M-GPT's FID is surprisingly high. Did you scale its size?
> - **A4**:  The gap isn’t size-related but due to T2M-GPT’s frozen CLIP encoder, which limits text understanding versus our joint training approach.
> - **Q5**: What proves frozen CLIP is ineffective? CLIP mainly helps T-M alignment.
> - **A5**: We retrained T2M-GPT with comparable GPT-2 medium parameters (380M vs. 355M). It still underperforms our approach.
> - **Q6**: Why does T2M-GPT struggle against GPT-2? Pretraining or CLIP settings?
> - **A6**: T2M-GPT’s frozen CLIP encoder provides fixed text representations, while GPT-2 enables joint optimization of text and motion, achieving better semantic alignment.
> - **Q7**: NO FURTHER REPLY
>
> ---
>
> ## **3-round questions of "Should the reviewer base on initial paper?"**
> - **Q1**: I would like to clarify that the significant revisions related to contributions are not supported officially, according to the review guidance.
> - **A1**:We have not made significant changes. All updates address reviewers' comments, align with our initial conclusions, and do not contradict them.
> - **Q2**:  According to the ICLR review pipeline, I have the right to ignore these major changes.
> - **A2**:  Disregarding our responses would be unfair. ICLR allows revisions during rebuttals to address reviewer concerns, unlike conferences like CVPR and NeurIPS. These updates aim to clarify misunderstandings and do not alter our original contributions.
> - **Q3**:  PHC and the dynamic data expansion are major revisions. The discussion process is for addressing misunderstandings in the reviews, not for providing major revisions.
> - **A3**:  The ICLR reviewer guideline states:
> Engage in discussion: The discussion phase at ICLR is different from most conferences in the AI/ML community. During this phase, reviewers, authors and area chairs engage in asynchronous discussion and authors are allowed to revise their submissions to address concerns that arise. It is crucial that you are actively engaged during this phase. Maintain a spirit of openness to changing your initial recommendation (either to a more positive or more negative) rating.
> - **Q4**: NO FURTHER REPLY

---

### Meta-Review · Area_Chair_A5AJ · 2024-12-20

**Metareview:**

The submission proposes MotionBase, a large-scale dataset of human motion, as well as methods that aim to develop a large motion model.  The submission received mixed feedback before and after the rebuttal, with long, extensive, and somewhat intense discussions.  The AC read the submission, reviews, rebuttals, and discussions.

The positive side of the submission is the usefulness of the dataset and the additional evaluation.  The concerns are about the use of static frames, as well as the possibly "major" changes during the rebuttal.  The AC agreed with the authors that links can be used and changes are allowed during rebuttal; meanwhile, the AC would also like to acknowledge that reviewers are indeed allowed to ignore changes if they are considered too major.  Whether the changes are major is mostly a subjective decision, and people may have different opinions.

The AC carefully reviewed the submission and the anonymous link to better understand the dataset and technical contributions.  The AC found the quality of the demos below expectations, with missing documentation and unclear visualizations.  The authors are encouraged to thoroughly revise the submission to significantly improve the presentation of the main paper, supp, and video demos for the next venue.

**Additional Comments On Reviewer Discussion:**

The discussion was intense, and eventually reviewers remained split.

---

### Decision · Program_Chairs · 2025-01-22

Reject